# DeGLIF for Label Noise Robust Node Classification using GNNs

## Abstract

Noisy labelled datasets are generally inexpensive compared to clean labelled datasets, and the same is true for graph data. In this paper, we propose a denoising technique DeGLIF: Denoising Graph Data using Leave-One-Out Influence Function. DeGLIF uses a small set of clean data and the leave-one-out influence function to make label noise robust node-level prediction on graph data. Leave-one-out influence function approximates the change in the model parameters if a training point is removed from the training dataset. Recent advances propose a way to calculate the leave-one-out influence function for Graph Neural Networks (GNNs). We extend that recent work to estimate the change in validation loss, if a training node is removed from the training dataset. We use this estimate and a new theoretically motivated relabelling function to denoise the training dataset. We propose two DeGLIF variants to identify noisy nodes. Both these variants do not require any information about the noise model or the noise level in the dataset; DeGLIF also does not estimate these quantities. For one of these variants, we prove that the noisy points detected can indeed increase risk. We carry out detailed computational experiments on different datasets to investigate the effectiveness of DeGLIF. It achieves better accuracy than other baseline algorithms.

## 1 INTRODUCTION

Data labelling is expensive and requires domain experts. In the absence of expertise or due to human weariness/negligence, data points are often labelled incorrectly, making the dataset noisy (Dai et al., 2021; Tripathi & Hemachandra, 2020; Sastry & Manwani, 2017). Other sources of noise can be erroneous devices, adversaries changing labels, insufficient information to provide labels, etc. The impact of noisy labels and the need to learn with noisy labels become more pronounced for GNNs as they use message passing on graph data. Message passing with noisy labels can propagate noise through network topology, leading to a significant decrease in performance (Dai et al., 2021; Qian et al., 2023). Addressing the degradation of GNNs due to noisily labelled graphs has become a significant challenge, attracting increased attention from researchers (Yu et al., 2019; Dai et al., 2021; Qian et al., 2023; Du et al., 2021; Yuan et al., 2023; Zhu et al., 2024; Li et al., 2024). In high-stakes applications like medical graph annotation or fake news detection, the annotation process is labour-intensive and prohibitively expensive. Consequently, a practical data landscape typically emerges: a scarce/small, high-fidelity 'clean data set' ($< 2\%$ nodes) curated by domain experts, coexisting with a vast corpus of noisily labeled data acquired via scalable but unreliable means like crowd-sourcing.

While existing noise robust approaches often disregard the former to focus exclusively on robust learning from the latter, this work addresses the practical scenario of jointly leveraging both data sources.

In this work, we use an extension of leave-one-out influence function (Kong et al., 2022; Hammoudeh & Lowd, 2022) to denoise noisy graph datasets. We consider a setup where we have a dataset with noisy node labels ($D$) and a small set of clean nodes ($D_c$). We train our model on the $D$ and use it to obtain predictions on $D_c$. Suppose we drop a training node $z$ and retrain our model on new training data. If we get a better prediction accuracy on $D_c$ using new weights, it is reasonable to assume that $z$ is out of distribution

with respect to $D_c$. The goal of the denoising model is to make $D$ close to $D_c$, which can be achieved by changing the label of $z$ (under the assumption that the only difference between the distribution of $D$ and $D_c$ is the noise in $D$). Dropping one training point at a time and retraining the model is computationally very expensive; hence, we approximate this change in empirical risk over $D_C$ using an extension of leave-one-out influence function. Koh & Liang (2017) proposed the use of the influence function to approximate the change in model parameters of MLP (Multi-Layer Perceptron) if a training point is dropped. This approximation can further be extended to approximate change in validation loss (on $D_c$) if a training point is dropped (Koh & Liang, 2017; Kong et al., 2022).

Leave-one-out influence function proposed for i.i.d. data (Koh & Liang, 2017; Kong et al., 2022) is not directly applicable to GNNs as removing a node also removes all edges connected to that node. This further impacts the intermediate representation of other nodes in GNN, as information cannot flow through these edges. Chen et al. (2023) proposed a formulation to calculate the influence function for graph data, which gives the approximate change in GNN's parameter when an edge/node is dropped from training data. Here, we extend that work to approximate the change in validation loss if an edge/node is dropped. We use this estimate to predict noisy nodes and then denoise using a theoretically motivated relabelling function. As per our information, there has been no prior work on the intersection of the influence function and noise-robust node classification, making this the first attempt to use the leave-one-out influence function for graph data denoising.

The main contributions of the paper are as follows: **1**. We propose a novel way to denoise graph data using the leave-one-out influence function. A variant of the influence function is used to identify the impact of training nodes on clean validation nodes (Section 3.1). **2**. We design a theoretically motivated relabelling function to process this information and then decide which nodes are noisy and relabel them (Section 3.2). **3**. We perform detailed experiments to see the effectiveness of the proposed algorithm (including on the dense Amazon Photo dataset). Overall, we observe that the accuracy obtained with denoising is up to 17.9% higher from other baselines (Upto 8% higher iin terms of absolute value, see Section 5). **4**. We also perform experiments to understand different aspects of our algorithm, like the role of the size of $D_c$, role of hyperparameter $\lambda$ and $\mu$, the effectiveness of the relabelling function and the decrease in the fraction of noisy nodes in training data with successive applications of DeGLIF (Section 5)

**Notations:** Let $G = (V, E, X, Y)$ represent the graph with $n$ training nodes. We summarize the key mathematical notations used throughout this paper below:

- **Datasets & Graph Elements:** $D$ denotes the training dataset containing noisy node labels, and $D_c$ is the small, high-fidelity clean validation dataset. $D^*$ represents the final denoised training dataset, and $D_n$ is the subset of training nodes identified as noisy. A generic training node is denoted as $z_i = (x_i, y_i)$ for $i \in [n]$, where $x_i$ is the feature and $y_i$ is the label. Clean validation points are denoted by $v_i \in D_c$.

- **Model Parameters:** $\theta$ represents the GNN model parameters. $\hat{\theta}$ denotes the optimal parameters obtained by training on $D$. The term $\hat{\theta}_{\epsilon,z}$ represents the optimal parameters when a training point $z$ is upweighted by a factor of $\epsilon$, and $\hat{\theta}_{-z}$ denotes the optimal parameters when node $z$ is entirely removed from the graph.

- **Loss & Risk:** $L(z, \theta)$ defines the loss evaluated at point $z$ for the model parameterized by $\theta$. $R(\theta, D_c)$ is the empirical risk evaluated over the clean validation dataset $D_c$, and $H_\theta$ is the Hessian matrix of the loss function.

- **Influence Functions:** $I(-z)$ is the approximated change in model parameters if training point $z$ is removed. $I(z \rightarrow z_\delta)$ denotes the approximated change if $z$ is replaced with a new point $z_\delta$. The term $I_{up}(-z, v_i)$ represents the approximate change in the loss of a validation point $v_i$ when $z$ is removed. $I_{cv}(-z)$ is the summation of $-I_{up}(-z, v_i)$ across all $v_i \in D_c$.

## 2 LEAVE-ONE-OUT INFLUENCE FUNCTION

Many times in different applications (e.g. Explainability (Koh & Liang, 2017), Out of distribution point detection (Kong et al., 2022)), we may want to remove a training node and observe the change in model parameters. Retraining again and again after removing one training point at a time is computationally very expensive, even for moderate-sized graphs. As the name suggests, leave-one-out influence function is used to approximate the impact of removing a training point. It is used to approximate the change in model parameters if the model is retrained, leaving out a training point. Using 1st-order Taylor series approximation, Koh & Liang (2017) derived the influence function for MLP. Based on this work, the change in the model parameters for an MLP, if a training point $z$ is up-weighted by $\epsilon$, is given by

$$I(\epsilon, z) := \hat{\theta}_{\epsilon, z} - \hat{\theta} = -\epsilon H_{\theta}^{-1} \nabla_{\theta} L(z, \hat{\theta})$$

where $\hat{\theta}$ and $\hat{\theta}_{\epsilon, z}$ are the model parameters obtained when the model is trained on complete training data and on training data with $z$ up-weighted by factor $\epsilon$, respectively. $L(z, \theta)$ is the loss at point $z$ for a model with parameter $\theta$, $n$ is the size of training data and the matrix $H_{\theta} = \frac{1}{n} \sum_{i=1}^{n} \nabla_{\theta}^{2} L(z_i, \theta)$. For derivation, see Appendix C.1. Removing a point is equivalent to choosing $\epsilon = -\frac{1}{n}$. The change in model parameter of an MLP, if a training point $z$ is removed is given by

$$I(-z) := \hat{\theta}_{-1/n, z} - \hat{\theta} = \frac{1}{n} H_{\theta}^{-1} \nabla_{\theta} L(z, \hat{\theta}) \tag{1}$$

Further, this can be used to approximate the change in parameters if a training point $z$ is replaced by $z_{\delta}$, it is denoted by $I(z \rightarrow z_{\delta})$ and is given by ($I(+z_{\delta})$ denotes the change in parameters if $z_{\delta}$ is added as a training point; see Koh & Liang (2017) for more details).

$$I(z \rightarrow z_{\delta}) = I(-z) + I(+z_{\delta}) = I(-z) - I(-z_{\delta}) \tag{2}$$

### 2.1 Estimating the Change in Parameters for Graph Data

Our objective is ~~We want~~ to estimate the change in model parameters $(\hat{\theta}_{-1/n, z} - \hat{\theta})$ when we remove an edge or a node from a graph. In graph data, nodes are not i.i.d. Removing a node leads to the removal of edges associated with it, which further leads to a change in the intermediate representations of other nodes in a GNN (as the representations of neighbouring nodes are aggregated to obtain the representation at the next step). So, implicitly a node $z$ is involved in more than one term while calculating the empirical risk($\frac{1}{n} \sum_{i}^{n} L(z_i, \theta)$). This is not the case with i.i.d. data, and hence, Equation (1) is insufficient for graph data. Chen et al. (2023) derived the influence function when removing a node or an edge. From now on, we will use $\hat{\theta}_{-z}$ for $\hat{\theta}_{-1/n, z}$ ($\hat{\theta}_{-x}$ denotes the optimal model parameter if $x$ is removed from graph, $x$ can be an edge or a node).

#### 2.1.1 Impact of Removing an Edge

If an edge $e_{ij}$ is removed, messages can no longer pass through $e_{ij}$. This means the adjacency matrix of the graph gets updated. If $M$ denotes the last layer representation matrix and $\Delta$ denotes the last layer representational change, then $M$ gets updated to $M + \Delta$ because of an update in the adjacency matrix. Using Equation (2), if edge $e_{ij}$ is removed, the update in the parameters is given by ($V_{train}$ : set of training nodes)

$$I(-e_{ij}) := \hat{\theta}_{-e_{ij}} - \hat{\theta} = I(M \rightarrow M + \Delta) = \sum_{k \in V_{train}} I(+(M_k + \Delta_k)) + I(-M_k). \tag{3}$$

#### 2.1.2 Impact of Removing a Node

If a node $z$, having feature $z_i$, is removed, then the loss term $L((z_i, y_i), \theta)$ is not present in risk calculation, and all the edges connected to that node also get removed. If these edges are removed, the matrix last layer representation matrix $M$ gets updated to $M + \Delta$. When a node $z$ is dropped, the update in the parameters $(\hat{\theta}_{-z} - \hat{\theta})$ is be given by (expanded by Eq. (1))

---

**Algorithm 1** DeGLIF

    **Input:** $D, D_c, \lambda$, GNN Model-1, GNN Model-2
    **Output:** $D^*$, Model-2 parameters
1: $f(.) =$ Train Model-1 on $D$
2: Compute $I_{up}(-z_i, v_j); \ \forall z_i \in D, v_j \in D_c$
3: $D_i =$ **Identify**$(\{I_{up}(-z_i, v_j)\}, \lambda)$                          ▷ **Identify** = DeGLIF(mv) or DeGLIF(sum)
4: **If** $D_i == 1$ **then**
5:      $y_i^* = \arg \max_k (\{f(x_i)_1, \ldots, f(x_i)_k\} \setminus \{f(x_i)_{y_i}\})$
6: Retrain Model-2 on $D^*$

---

$$
\begin{aligned}
I(-z) &= I(-z_i) + I(M \rightarrow M + \Delta) \\
&= I(-z_i) + \sum_{k \in V_{train}} I(+(M_k + \Delta_k)) + I(-M_k), \\
&= \frac{1}{n} H_\theta^{-1} \left\{ \mathbb{1}_{v_i \in V_{train}} \nabla_\theta L((z_i, y_i), \hat{\theta}) - \sum_{k \in V_{train}} \nabla_\theta L((M_k + \Delta_k, y_k), \hat{\theta}) + \sum_{k \in V_{train}} \nabla_\theta L((M_k, y_k), \hat{\theta}) \right\}.
\end{aligned}
\tag{4}
$$

When removing an edge, $\Delta$ reflects the representational change due to the absence of information flow through that specific edge. In contrast, when removing a node, $\Delta$ encompasses the change resulting from the elimination of all incident edges. For derivation of Equations (3) and (4) see Chen et al. (2023).

## 3 PROPOSED ARCHITECTURE

The goal of DeGLIF is to learn from noisy data using a small set of clean data ($D_c$). The algorithm is summarized in Algorithm 1. We begin with a noisy dataset $D$. In step 1, a GNN (Model-1) is trained on noisy data (see Fig. 1 for DeGLIF architecture). In step 2, using the trained parameters and $D_c$, $I_{up}(-z, v_i)$ is calculated for every pair $(z_i, v_j) \in D \times D_c$ (see Subsection 3.1). Step 3 uses these values to determine if a node $z$ is noisy (see DeGLIF(mv) and DeGLIF(sum) in Section 3.1). In steps $4-5$ noisy points are then passed to the relabelling function, which updates their labels (see Section 3.2), producing a denoised dataset $D^*$. Finally, in step , we train GNN Model-2 on $D^*$ to get the final output (see Fig. 1). One can choose any model for Model-1 as long as the Hessian obtained is invertible ($H_\theta$, Equation (4)). DeGLIF supports different GNN models and loss functions and can be combined with noise robust loss functions like (Tripathi & Hemachandra, 2020; Wang et al., 2019; Kumar & Sastry, 2018) for improved results. We now elaborate on different components of DeGLIF.

### 3.1 Identifying Noisy Nodes

We have an approximation for change in the parameters if we drop a training node (Equation (4)). We extend the work by Chen et al. (2023) to estimate the change in loss over a clean validation point if we drop a training node and use it to identify noisy nodes. If dropping a training point leads to a decrease in loss of a significant number of validation points, then we may infer that the training point we dropped is out of distribution with respect to the validation set and hence noisy and hence, the training node needs to be relabelled. We calculate the change in risk $(L(v_i, \hat{\theta}_{\epsilon,z}) - L(v_i, \hat{\theta}))$ of a validation point $v_i$ if we remove a node $z$, using Equation (4) (see Appendix C.2 for detailed derivation)

$$
I_{up}(-z, v_i) := \left. \frac{dL(v_i, \hat{\theta}_{\epsilon,z})}{d\epsilon} \right|_{\epsilon=-1/n} = \left. \nabla_\theta L(v_i, \hat{\theta})^\top \frac{d\hat{\theta}_{\epsilon,z}}{d\epsilon} \right|_{\epsilon=-1/n} = \nabla_\theta L(v_i, \hat{\theta})^\top I(-z). \tag{5}
$$

$I_{up}(-z, v_i) < 0$ means, risk of point $v_i$ would decrease on removing the node $z$. Based on how a node is identified as noisy, we have two algorithms:

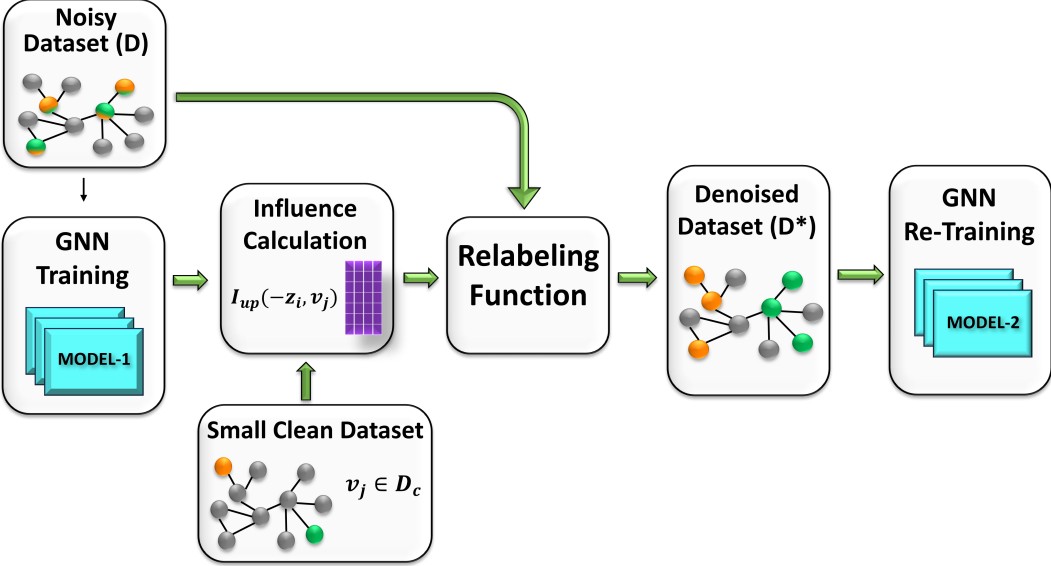

Figure 1: Proposed DeGLIF Architecture: Model-1 is Trained on Noisy Dataset $D$, Trained Weights are Used to Calculate Influence for Every Pair of Training Point $z_i$, and $v_j \in D_c$ (Small Clean Dataset). Influence Values are then Passed to a Relabelling Function, Producing Denoised Data $D^*$. Model-2 is Trained on $D^*$ for Final Output.

• **DeGLIF(mv):** Given a $\lambda \in [0.5, 1)$, we change the label of a training point $z$ if it has a negative influence on more than a $\lambda$ fraction of the validation points. Specifically, we count the validation points $v_i \in D_c$ with negative influence, where $I_{up}(-z, v_i) < 0$. We say $z$ is noisy if this count is at least $\lambda \cdot |D_c|$. In our experiments, we treat $\lambda$ as a hyperparameter and is tuned over the set $\{0.5, 0.52, 0.53, 0.55, 0.56, 0.6\}$ using the validation set. $\lambda \geq 0.5$ acts as a modified majority vote. (Algorithm 2 in Appendix A).

• **DeGLIF(sum):** $I_{cv}(-z) := \sum_{v_i \in D_c} -I_{up}(-z, v_i)$. Given $\mu$, $z$ is classified as noisy if $I_{cv}(-z) > \mu$. We tune $\mu$ over the set $\{0, 0.1, 1, 10, 20\}$. (Algorithm 3 in Appendix A).

As Eq. $(1)-(5)$ are loss dependent, here onwards, we fix the loss function as the cross-entropy loss (in practice, we also use a dataset dependent regularization parameter(weight decay)). Let $R(\theta, D_c) := \frac{1}{n} \sum_{v_i \in D_c} L(v_i, \theta)$, denote the empirical risk when model with optimal parameter $\theta$ is tested on clean dataset $D_c$ of size $n$. Also, let $D_n$ represent the set of points detected as noisy by DeGLIF(sum). Following Kong et al. (2022), Chen et al. (2023) and Wang et al. (2020), we assume additivity of influence function when more than one training node is removed; also see Remark below. We derive the following Theorem.

**Theorem 1.** *Let $\hat{\theta}$ be an optimal parameter for GNN when the model is trained on noisy dataset $(D)$, and $\hat{\theta}_{-D_n}$ be the optimal parameter when the model is trained, removing training nodes in $D_n$. Then,*

$$R(\hat{\theta}, D_c) - R(\hat{\theta}_{-D_n}, D_c) \approx \frac{1}{n} \sum_{z \in D_n} I_{cv}(-z) \geq 0$$

Theorem 1 (proof in Appendix C.3) justifies noisy node identification by DeGLIF(sum), as removing these nodes can reduce test risk.

**Remark 1:** While the additivity of influence functions is a standard approximation, its application to graph data is inherently non-trivial due to the structural dependencies and potential second-order interactions between neighboring nodes. To rigorously validate the practical reliability of this assumption, we provide an empirical evaluation of group node removal (for various group sizes) in Appendix D.1. Crucially, we observe that the influence-predicted changes maintain a remarkably strong Pearson correlation ($\approx 0.76$) with the actual loss changes even for large group sizes, confirming that this approximation remains highly reliable for practical applications.

**Remark 2**: Although the standard derivation of the influence function assumes $\hat{\theta}$ is a global minimum, the derivation (Appendix C.1) and Theorems 1 and 2 remain valid for any stationary point, provided the Hessian ($H_\theta$) is invertible. Furthermore, in many practical scenarios, a GCN may fail to converge to even a local minimum. To evaluate the validity of our influence estimates under these conditions, we compared the predicted change in loss against the actual change (as discussed in Remark 1). Using an identical experimental setup to Appendix D.1 with a group size of 1, we observed a consistently high correlation (ranging from 0.70 to 0.86) across multiple runs. This demonstrates DeGLIF's ability to reliably identify noisy nodes even when the model does not converge to a strict stationary point.

### 3.2 Relabelling Function

Relabelling function for the *binary labelled datasets* ($\{z_i = (x_i, y_i)|y_i \in \{0,1\}\}$) is simple, if a node is predicted noisy, its label is flipped ($y_i^* = 1 - y_i$). Theorem 3 (present in Appendix C.5 with proof), shows relabelling nodes in $D_n$ can lead to even lower risk than removing nodes in $D_n$. We aim for similar property in relabelled multiclass dataset. Let a dataset have $j$ classes. For a training node $z = (x, y) \in D_n$ (predicted noisy) with label $m$, let the final layer GNN probability distribution output be $f(x) = [f(x)_1, \ldots, f(x)_j]$. If we relabel $z$ as $[0, \ldots, \varphi_k(z), \ldots, 0]$ (similar to one hot encoding with nonzero value at position $k$), where $\varphi_k(z) := \log_{f(x)_k}(1 - f(x)_m)$ for $k \neq m$. Theorem 2 gives, training on such relabelled data can achieve even lower risk over $D_C$, than $D_n$. Relabelling is equivalent to removing that node and adding back the node with a new label. $I_{up}(z \to z_\delta, v_i) = \nabla_\theta L(v_i, \hat{\theta})^\top I(z \to z_\delta)$ denotes approximate change in loss of $v_i$, when a node $z$ is relabeled as $z_\delta$. Define $I_{cv}(z \to z_\delta) = \sum_{v_i \in D_c} I_{up}(., v_i)$.

**Theorem 2.** *For a multiclass dataset $\{z_i = (x_i, y_i)\}$, where $y_i \in \{1, \ldots j\}$. Let the relabelling function be $r(z_i) = [0, \ldots, \log_{f(z)_k}(1 - f(z)_m), \ldots, 0]$. Let $\hat{\theta}_r$ denote the optimal parameter when the model is trained on relabelled data then,*

$$R(\hat{\theta}_{-D_n}, D_c) - R(\hat{\theta}_r, D_c) \approx \frac{1}{n} \sum_{z \in D_n} I_{cv}(-z) - I_{cv}(z \to z_\delta) \geq 0$$

Every node in $D$ has a single label; we want the denoised data ($D^*$) to retain this structure and have a single label rather than a probabilistic label. Probabilistic assignment also restricts the choice of loss function after denoising (e.g., $0 - 1$ loss can not be used). As above, $f = [f_1, \ldots, f_j]$ is the prediction probabilities. For a node $z$, using label $y = [0, 0, \ldots, 1, 0, 0, \ldots, 0]$ (1 at $k-$th position), gives cross-entropy loss $-1 \times \log(f_k)$. Whereas, if we using $y = [0, \ldots, \varphi_k(z), \ldots, 0]$, yields $-\varphi_k \log(f_k)$. The model's output $f(x)$ is a valid probability distribution where $f(x)_k + f(x)_m \leq 1$, which implies $1 - f(x)_m \geq f(x)_k$ and consequently $0 \leq \log_{f(x)_k}(1 - f(x)_m) \leq 1$. As, $0 \leq \varphi_k(z) \leq 1, \forall z, k$, Theorem 2 implies that relabeling $z$ to class $k$ and downweighting the loss by $\varphi_k$, can achieve a lower risk on $D_c$ than removing $z$ from training data. To assign a single label to noisy nodes, we relabel $z$ to the class requiring the least downweighting. If $f_{k1} > f_{k2}$ then $\varphi_{k1} > \varphi_{k2}$, hence the new label for $z$ is $y* = \operatorname{argmax}(\{f_1, \ldots, f_j\} \setminus f_m)$. The same relabelling function is used for DeGLIF(mv), and performs well empirically. It is worth mentioning that, because $D_c$ is sampled i.i.d. from the underlying clean distribution, the empirical risk reductions established in Theorems 1 and 2 serve as unbiased estimators for the true expected risk, thereby generalizing to the unseen test risk.

## 4 EXPERIMENTAL SETUP

**Noise Model Used:** We use Symmetric Label Noise (SLN) and Pairwise Noise (Pairwise noise is a type of broader class of noise called Class Conditional Noise) (Tripathi & Hemachandra, 2020; 2019; Ghosh et al., 2015) models to inject noise in training data. We want to mention that our algorithm *does not* use or estimate noise level in the graph, noise level is an unknown quantity. See Appendix B for details about these noise models.

**Datasets Used:** We test DeGLIF for both binary and multiclass classification. For the *multiclass classification* task, we test DeGLIF on Cora (Yang et al., 2016), Citeseer (Yang et al., 2016) and Amazon photo (Shchur et al., 2018) datasets (see Table 1 for dataset statistics). Details about the binary labelled

dataset and its results are in Appendix D.5. The PubMed dataset (Yang et al., 2016) has been widely used by baseline algorithms. However, we observe that its performance does not degrade significantly (especially for SLN) even under high noise levels, leaving limited scope for improvement. For completeness, we include a comparison on the PubMed dataset in the supplementary material (Appendix D.7).

Table 1: Dataset Statistics

| Dataset | # Nodes | # Edges | Feature dim | # Classes |
|---|---|---|---|---|
| Cora | 2,708 | 10,556 | 1,433 | 7 |
| CiteSeer | 3,327 | 9,104 | 3,703 | 6 |
| Amazon Photo | 7,650 | 238,162 | 745 | 8 |

### 4.1 Implementation Details

In DeGLIF, we employ two GNN models (see Fig. 1). DeGLIF supports any combination of GNN architecture, provided the Hessian of Model-1 is invertible. For fair comparison with baselines, we select both Model-1 and Model-2 as GCN (Kipf & Welling, 2017) with a single hidden layer having a dimension of 16 (for the Citeseer dataset, hidden dimension is 10). The output dimension matches the number of classes in the dataset. All experiments are repeated 5 times with random seeds, and mean ± standard deviation are reported. Implementation is done in Python, with training on NVIDIA 24 GB A5000 and 15 GB T4 GPUs.

#### 4.1.1 Baselines

We compare our method with existing state-of-the-art noise-robust models for node classification: **GCN** (Kipf & Welling, 2017), **Coteaching+** (Yu et al., 2019), **NRGNN** (Dai et al., 2021), **RTGNN** (Qian et al., 2023), **CP** (Zhang et al., 2020), **CGNN** (Yuan et al., 2023), **CRGNN** (Li et al., 2024), **RNCGLN** (Zhu et al., 2024), **PIGNN** (Du et al., 2021), **DGNN** (NT et al., 2019), **TSS** (Wu et al., 2024). For details about the methodology adapted by these models, see Section 8. For GCN, we use the implementation from the PyG (Fey & Lenssen, 2019) library. Also, for a fair comparison, we use the implementation of NRGNN, Coteach+ by Dai et al. (2021), the implementation of CP, CGNN, CRGNN, RNCGLN, PIGNN, and DGNN by Wang et al. (2024), and the implementation of RTGNN (Qian et al., 2023) and TSS (Wu et al., 2024).

## 5 COMPUTATIONAL RESULTS

We add noise to original datasets using SLN and Pairwise noise models, varying noise levels from 5% to 50% in steps of 5%. For Cora and Citeseer, we use 1000 nodes for testing, 500 for validation and the rest for training. For the Amazon photo, we use 5% nodes for training, 15% for validation, and the rest for testing. For all datasets, we choose 50 nodes as clean nodes from the validation set. This means we take around 1.85% of the Cora dataset, 1.5% of the Citeseer dataset, and 0.7% of the Amazon Photo dataset as clean nodes that are used for influence calculation.

Results reported in Table 2 show that DeGLIF outperforms existing state-of-the-art methods (by up to 17.8%) in most cases and is comparable otherwise. Our results highlight distinct advantages of DeGLIF over existing baselines across varying dataset characteristics. First, standard GNNs (e.g., GCN) tend to overfit to noisy training data, leading to a significant degradation in performance as noise levels increase. The influence function enables DeGLIF to mitigate this by actively identifying and correcting these noisy nodes. Second, DeGLIF is robust to different dataset sizes and densities. For instance, the training set size for Amazon Photo is small; this hinders semi-supervised methods like Co-teaching+. Additionally, methods like NRGNN and RTGNN rely on edge predictors to connect unlabelled nodes; this strategy offers little benefit on already dense graphs like Amazon Photo. In contrast, DeGLIF remains effective across both sparse and dense topologies. Finally, we observe that DeGLIF and CP perform comparably on Amazon Photo, but DeGLIF performs noticeably better on other datasets. The CP algorithm relies on clear community structures to work well (Zhang et al. 2020). Amazon Photo is a dense graph organized into strong item-based communities; conversely, the sparse citation graphs of Cora and Citeseer often have weak or overlapping

Table 2: Performance Comparison for the Multi-Class Node Classification Task in Presence of Label Noise. For Every Dataset and Every Noise Level, the Highest Value is Marked in **Bold** and the $2^{nd}$ Highest is Underlined.

| Noise Model | Dataset | Noise Robust Algorithm | Noise Level | | | | | | | | | |
|---|---|---|---|---|---|---|---|---|---|---|---|---|
| | | | 5% | 10% | 15% | 20% | 25% | 30% | 35% | 40% | 45% | 50% |
| Symmetric Label Noise | Cora | GCN | 85.5 ± 0.6 | 84.5 ± 0.5 | 83.1 ± 1.7 | 81.5 ± 0.9 | 79.2 ± 1.2 | 76.5 ± 1.5 | 72.8 ± 1.9 | 69.2 ± 1.8 | 65.1 ± 1.9 | 59.1 ± 2.8 |
| | | Co-teaching + | 81.5 ± 0.5 | 81.2 ± 0.5 | 79.9 ± 1.7 | 79.6 ± 0.8 | 79.4 ± 1.2 | 78.8 ± 1.5 | 77.7 ± 1.9 | 76.4 ± 1.8 | 74.2 ± 1.9 | 70.7 ± 2.7 |
| | | NRGNN | 84.1 ± 2.5 | 83.9 ± 2.6 | 82.7 ± 3.8 | 82.0 ± 2.5 | 81.0 ± 2.9 | 79.8 ± 2.5 | 78.7 ± 2.1 | 76.5 ± 3.7 | 75.1 ± 5.1 | 72.5 ± 3 |
| | | RTGNN | 79.2 ± 1 | 79.0 ± 2.7 | 80.7 ± 0.4 | 80.2 ± 3.6 | 83.8 ± 1.9 | 83.0 ± 0.4 | 82.5 ± 0.7 | 81.7 ± 0.9 | **84.0 ± 0.6** | 77.6 ± 0.7 |
| | | CP | 82.9 ± 1.3 | 81.4 ± 2.1 | 82.9 ± 1.8 | 79.0 ± 3 | 80.3 ± 0.9 | 75.8 ± 6.4 | 77.7 ± 3 | 70.1 ± 9.9 | 70.0 ± 7.6 | 65.6 ± 6.3 |
| | | CGNN | 83.4 ± 2.4 | 83.1 ± 2.6 | 82.5 ± 2 | 82.2 ± 3.9 | 82.9 ± 0.8 | 83.2 ± 2.3 | 81.2 ± 2.1 | 78.6 ± 4.2 | 77.0 ± 6.8 | 73.0 ± 8.9 |
| | | CRGNN | 84.5 ± 2.6 | 84.6 ± 1.7 | 82.8 ± 1.7 | 80.7 ± 1.7 | 80.3 ± 2.3 | 78.5 ± 1.6 | 73.8 ± 3.8 | 67.6 ± 7.5 | 66.6 ± 4.9 | 54.4 ± 15.4 |
| | | RNCGLN | 83.0 ± 0.2 | 79.8 ± 0.8 | 78.1 ± 1.1 | 76.7 ± 0.8 | 73.6 ± 2.7 | 70.4 ± 3.4 | 67.0 ± 4.6 | 62.6 ± 3.7 | 54.2 ± 2.3 | 51.7 ± 2.9 |
| | | PIGNN | 81.9 ± 1.5 | 83.3 ± 1.9 | 84.0 ± 1 | 81.2 ± 5.5 | 81.9 ± 2.1 | 78.8 ± 4.2 | 82.2 ± 1.2 | 78.0 ± 2.2 | 71.1 ± 12.3 | 75.9 ± 8.1 |
| | | DGNN | 80.8 ± 2.6 | 79.0 ± 3.2 | 80.9 ± 1.2 | 67.9 ± 11.9 | 77.3 ± 0.8 | 73.2 ± 5.2 | 64.3 ± 11.3 | 61.9 ± 16.5 | 65.0 ± 7.7 | 60.0 ± 7.6 |
| | | TSS | 82.4 ± 5.4 | 82.4 ± 2.3 | 83.4 ± 2 | 81.9 ± 5.3 | 81.1 ± 3.9 | 80.3 ± 4.1 | 76.9 ± 9.2 | 78.4 ± 3.8 | 75.1 ± 3.6 | 74.5 ± 4.4 |
| | | DeGLIF(mv) | 85.1 ± 0.3 | **84.7 ± 0.5** | 84.0 ± 0.2 | 83.2 ± 1.1 | 82.7 ± 0.8 | 82.1 ± 1.6 | 82.0 ± 0.7 | 80.9 ± 1.9 | 79.0 ± 2.3 | 76.8 ± 1.8 |
| | | DeGLIF(sum) | **86.1 ± 0.5** | **84.7 ± 0.8** | **85.8 ± 0.3** | **84.3 ± 1.3** | **84.6 ± 1.1** | **84.3 ± 0.7** | **84.8 ± 0.6** | **84.0 ± 0.6** | 81.1 ± 1.4 | **80.5 ± 1.8** |
| | Citeseer | GCN | 76.6 ± 0.2 | 73.9 ± 1.1 | 71.2 ± 0.9 | 69.0 ± 1.1 | 66.3 ± 0.4 | 63.4 ± 1.5 | 59.3 ± 1.4 | 55.4 ± 1.1 | 51.3 ± 1.3 | 48.0 ± 1.4 |
| | | Co-teaching + | 79.3 ± 1.7 | 77.8 ± 5.8 | 75.8 ± 6.6 | 73.6 ± 5.5 | 69.0 ± 9.1 | 71.8 ± 7.2 | 72.5 ± 6.1 | 65.6 ± 9.2 | 67.5 ± 8.5 | 61.7 ± 8.3 |
| | | NRGNN | 75.2 ± 1.1 | 74.3 ± 1 | 73.3 ± 1.4 | 72.2 ± 1.2 | 71.8 ± 1.3 | 71.4 ± 1.7 | 70.4 ± 1.8 | 69.5 ± 1.2 | 68.7 ± 2 | 67.9 ± 1.5 |
| | | RTGNN | 76.8 ± 0.6 | 76.5 ± 0.5 | 76.7 ± 0.7 | 76.4 ± 1.2 | 76.0 ± 1.7 | 76.3 ± 0.7 | 74.1 ± 0.6 | 73.2 ± 0.9 | 72.7 ± 1.5 | 72.7 ± 0.6 |
| | | CP | 77.6 ± 1.3 | 72.9 ± 4.6 | 76.6 ± 2.3 | 74.5 ± 3.5 | 71.2 ± 7.3 | 69.9 ± 7.7 | 65.5 ± 10.8 | 69.1 ± 12.3 | 67.0 ± 3.6 | 63.4 ± 14.1 |
| | | CGNN | 75.6 ± 4.1 | 73.9 ± 4.6 | 71.5 ± 7.8 | 71.4 ± 7.2 | 75.6 ± 2.4 | 71.6 ± 4.7 | 69.3 ± 6.3 | 68.8 ± 8.1 | 59.8 ± 13.5 | 64.3 ± 2.8 |
| | | CRGNN | 76.3 ± 2.4 | 74.8 ± 1.9 | 72.7 ± 3.8 | 73.2 ± 3.1 | 73.2 ± 1.6 | 72.6 ± 6.6 | 66.6 ± 8.4 | 70.3 ± 2 | 67.0 ± 8 | 58.6 ± 10.7 |
| | | RNCGLN | 72.2 ± 3.1 | 69.4 ± 2.6 | 67.7 ± 3.5 | 66.1 ± 3.8 | 65.1 ± 4.1 | 63.8 ± 4.7 | 58.3 ± 4.1 | 57.8 ± 4.9 | 51.7 ± 5 | 47.4 ± 2.6 |
| | | PIGNN | 76.6 ± 2 | 73.2 ± 5.3 | 72.3 ± 5.2 | 70.8 ± 4.8 | 71.6 ± 3.7 | 71.0 ± 4.7 | 66.6 ± 5.8 | 67.4 ± 4.1 | 60.8 ± 11.1 | 60.5 ± 7.4 |
| | | DGNN | 66.5 ± 2.8 | 62.1 ± 2.9 | 59.9 ± 2.9 | 56.1 ± 3.8 | 53.1 ± 4.9 | 46.9 ± 8.9 | 45.5 ± 6.3 | 38.7 ± 8.9 | 41.9 ± 6.7 | 33.3 ± 8.0 |
| | | TSS | 77.5 ± 1.9 | 76.9 ± 3.2 | 76.6 ± 4.2 | 77.5 ± 1.9 | 75.5 ± 3.1 | 74.1 ± 4.1 | 75.4 ± 4.3 | 73.0 ± 4.8 | 71.4 ± 4.1 | 70.3 ± 2.4 |
| | | DeGLIF(mv) | 77.8 ± 1.2 | 77.2 ± 1.2 | 76.9 ± 1.3 | 76.5 ± 0.5 | 76.5 ± 0.7 | 76.1 ± 0.4 | 75.4 ± 0.8 | 74.1 ± 0.6 | 74.2 ± 0.5 | 72.3 ± 0.7 |
| | | DeGLIF(sum) | **81.5 ± 0.8** | **81.2 ± 0.6** | **80.7 ± 0.8** | **79.5 ± 0.8** | **79.7 ± 0.7** | **78.8 ± 0.7** | **77.4 ± 1.9** | **75.6 ± 1.7** | **76.3 ± 1.9** | **74.3 ± 1.6** |
| | Amazon Photo | GCN | 87.3 ± 0.8 | 87.1 ± 0.3 | 85.5 ± 0.4 | 85.7 ± 1 | 85.7 ± 1 | 84.6 ± 2.1 | 83.7 ± 1.4 | 80.7 ± 2 | 79.1 ± 1.2 | 75.2 ± 5.2 |
| | | Co-teaching + | 84.4 ± 1.3 | 82.2 ± 1.6 | 82.0 ± 1.4 | 80.4 ± 1.3 | 73.2 ± 2.1 | 73.7 ± 1.8 | 61.1 ± 2.3 | 59.2 ± 2.4 | 57.1 ± 2.1 | 47.9 ± 2 |
| | | NRGNN | 69.0 ± 8 | 68.1 ± 6.8 | 72.4 ± 5.5 | 66.5 ± 5.1 | 55.1 ± 5.4 | 60.0 ± 7.1 | 58.3 ± 6 | 60.0 ± 5.1 | 54.5 ± 6.2 | 51.5 ± 5.9 |
| | | RTGNN | 80.8 ± 5.3 | 82.2 ± 5.3 | 83.6 ± 2.7 | 84.0 ± 2.2 | 82.9 ± 4.9 | 78.8 ± 6.4 | 79.3 ± 5.4 | 83.5 ± 4.7 | 86.0 ± 1.3 | 79.9 ± 5 |
| | | CP | 91.4 ± 0.6 | 90.6 ± 0.7 | 90.4 ± 1.3 | 89.9 ± 0.5 | **90.2 ± 1.1** | **89.9 ± 1.1** | **88.9 ± 0.4** | **87.3 ± 2.8** | 85.6 ± 3.5 | **83.0 ± 0.9** |
| | | CGNN | 61.1 ± 23.5 | 66.4 ± 17.6 | 51.4 ± 23.2 | 69.6 ± 22.8 | 61.7 ± 19.3 | 54.2 ± 26.4 | 52.0 ± 34.1 | 43.0 ± 26.4 | 42.3 ± 25.3 | 40.3 ± 26.7 |
| | | CRGNN | 37.5 ± 45.4 | 20.7 ± 36.4 | 33.5 ± 39.8 | 19.2 ± 34.3 | 37.5 ± 45.4 | 30.8 ± 37.2 | 32.2 ± 38.1 | 24.4 ± 30.2 | 50.9 ± 28.9 | 12.3 ± 17.7 |
| | | RNCGLN | 85.9 ± 2.5 | 84.7 ± 1.7 | 80.2 ± 4.1 | 82.7 ± 3.3 | 72.86 ± 4.4 | 74.1 ± 8.3 | 70.9 ± 7.1 | 64.2 ± 5.6 | 62.2 ± 10.1 | 46.6 ± 4.5 |
| | | PIGNN | 88.9 ± 0.4 | 90.1 ± 1.7 | **91.0 ± 1.3** | 88.1 ± 1.8 | 86.8 ± 3.4 | 87.6 ± 0.7 | 86.3 ± 1.8 | 82.2 ± 4.2 | 76.9 ± 4.3 | |
| | | DGNN | 64.9 ± 28.2 | 65.3 ± 23.5 | 55.7 ± 22.8 | 54.4 ± 17.5 | 56.1 ± 20.7 | 57.2 ± 15.4 | 53.0 ± 17.9 | 48.1 ± 14.9 | 46.6 ± 11.2 | 44.1 ± 17.1 |
| | | TSS | 86.9 ± 3.7 | 85.3 ± 3.5 | 84.7 ± 2.9 | 86.8 ± 2.1 | 85.9 ± 2.5 | 86.3 ± 1.8 | 85.3 ± 2.9 | 83.6 ± 2.0 | 82.5 ± 3.3 | 78.9 ± 4.8 |
| | | DeGLIF(mv) | 89.6 ± 0.9 | 89.6 ± 0.3 | 89.9 ± 1.9 | 89.7 ± 1.1 | 89.4 ± 1.4 | 89.1 ± 2.2 | 88.6 ± 0.9 | 86.7 ± 1.6 | 83.5 ± 1.2 | 82.0 ± 4.5 |
| | | DeGLIF(sum) | **91.8 ± 0.6** | **91.6 ± 0.2** | 90.8 ± 0.4 | 90.3 ± 1 | 90.0 ± 0.7 | 89.8 ± 2 | 88.1 ± 1.3 | 87.1 ± 1.6 | **86.4 ± 1.7** | 82.6 ± 4.2 |
| Pairwise Noise | Cora | GCN | 86.1 ± 0.6 | 85.2 ± 0.3 | 83.0 ± 1.1 | 80.0 ± 1.9 | 77.1 ± 2.7 | 72.7 ± 3.5 | 67.2 ± 4 | 60.4 ± 5 | 52.8 ± 4.4 | 43.7 ± 4.2 |
| | | Co-teaching + | 78.6 ± 1.5 | 78.6 ± 1.3 | 77.8 ± 2.2 | 78.3 ± 1.1 | 75.2 ± 4.5 | 74.9 ± 3.1 | 72.4 ± 1.7 | 65.8 ± 4.4 | 56.6 ± 5 | 40.6 ± 8.3 |
| | | NRGNN | 85.3 ± 0.6 | 84.6 ± 0.7 | 84.1 ± 0.7 | 82.0 ± 2.2 | 80.6 ± 2.1 | 78.3 ± 2.6 | 72.3 ± 3 | **69.1 ± 4.3** | 62.3 ± 4.9 | 46.6 ± 6.4 |
| | | RTGNN | 79.2 ± 1.4 | 80.5 ± 2 | 80.4 ± 0.9 | 82.0 ± 2.4 | 77.5 ± 0.6 | 74.3 ± 1.3 | 72.1 ± 2.3 | 59.6 ± 1.5 | 51.1 ± 2.6 | 43.6 ± 2 |
| | | CP | 82.2 ± 0.7 | 81.9 ± 3.1 | 78.5 ± 3.3 | 74.8 ± 5.8 | 71.0 ± 3.3 | 66.4 ± 2.3 | 56.4 ± 7.8 | 56.1 ± 2.3 | 48.9 ± 3.9 | 37.3 ± 6.8 |
| | | CGNN | 84.1 ± 2.6 | 83.6 ± 2 | 82.4 ± 1.4 | 79.3 ± 3.5 | 78.4 ± 4 | 74.5 ± 5.9 | 69.0 ± 6 | 58.2 ± 10.2 | 53.0 ± 4.6 | 47.0 ± 8.6 |
| | | CRGNN | 84.6 ± 2.5 | 80.6 ± 2.6 | 78.0 ± 3.1 | 75.4 ± 2.3 | 58.9 ± 15.7 | 69.9 ± 4.8 | 60.7 ± 3.2 | 47.3 ± 7.8 | 47.3 ± 8.3 | 42.8 ± 6.5 |
| | | RNCGLN | 79.4 ± 1.8 | 78.6 ± 1.3 | 75.8 ± 2.3 | 71.1 ± 2 | 65.7 ± 2.6 | 63.0 ± 3.5 | 56.2 ± 3.6 | 51.5 ± 5.1 | 45.7 ± 5.7 | 40.0 ± 2.5 |
| | | PIGNN | 83.9 ± 1.2 | 82.1 ± 1.8 | 83.1 ± 2.6 | 78.8 ± 2.7 | 76.8 ± 1.3 | 72.8 ± 3.6 | 71.0 ± 3.4 | 54.2 ± 9.7 | 58.3 ± 5.5 | 40.7 ± 6.6 |
| | | DGNN | 82.92 ± 1.6 | 79.4 ± 2.8 | 75.4 ± 1.6 | 74.3 ± 3.2 | 65.3 ± 8.9 | 65.5 ± 6.3 | 58.9 ± 7 | 56.0 ± 5.5 | 49.1 ± 13.3 | 44.5 ± 7.2 |
| | | TSS | 82.5 ± 3.1 | 81.6 ± 5.2 | 81.6 ± 1.8 | 80.6 ± 4.1 | 78.3 ± 2.5 | 74.8 ± 5.3 | 69.0 ± 6.8 | 63.6 ± 3.2 | 52.4 ± 13.4 | 45.0 ± 9.6 |
| | | DeGLIF(mv) | 86.3 ± 0.8 | **86.0 ± 0.7** | **84.8 ± 0.3** | **83.2 ± 0.1** | **81.1 ± 1.6** | 76.7 ± 2 | 72.1 ± 2 | 64.8 ± 3.8 | 57.9 ± 4.1 | 48.6 ± 3.8 |
| | | DeGLIF(sum) | **86.5 ± 0.4** | 85.9 ± 0.4 | 84.1 ± 0.5 | 82.6 ± 0.5 | 80.8 ± 0.5 | **78.5 ± 0.8** | 73.9 ± 3.5 | 66.8 ± 2.5 | **63.0 ± 4.5** | **51.8 ± 4.6** |
| | Citeseer | GCN | 78.6 ± 0.6 | 78.2 ± 0.6 | 76.8 ± 0.9 | 75.5 ± 1.1 | 73.2 ± 1.1 | 70.6 ± 1.6 | 67.0 ± 2.2 | 62.1 ± 3.8 | 47.1 ± 2.7 | 45.3 ± 3.3 |
| | | Co-teaching + | 74.9 ± 1.1 | 75.7 ± 1 | 72.7 ± 0.9 | 71.0 ± 2.1 | 67.0 ± 4.4 | 65.1 ± 5.7 | 63.0 ± 6 | 57.8 ± 12.1 | 51.8 ± 4.7 | 45.4 ± 7.2 |
| | | NRGNN | 77.3 ± 1 | 76.1 ± 1.7 | 75.6 ± 0.9 | 73.0 ± 2.1 | 71.4 ± 2.1 | 68.6 ± 3.4 | 63.7 ± 3.1 | 57.6 ± 5.1 | 55.0 ± 6.8 | 45.4 ± 4.6 |
| | | RTGNN | 76.7 ± 0.2 | 77.3 ± 0.3 | 76.0 ± 1.1 | 75.8 ± 1.2 | 74.3 ± 1.1 | 71.3 ± 1.8 | 71.0 ± 1.6 | 67.9 ± 1 | 57.6 ± 2.5 | 45.8 ± 1.5 |
| | | CP | 78.3 ± 2.5 | 73.9 ± 3.9 | 65.7 ± 2.8 | 66.2 ± 4.5 | 64.3 ± 6 | 59.4 ± 6.5 | 54.9 ± 5.1 | 49.4 ± 6.9 | 43.5 ± 5.3 | 41.1 ± 2.1 |
| | | CGNN | 75.0 ± 2.3 | 77.0 ± 1.3 | 75.3 ± 1.4 | 69.3 ± 3.7 | 69.1 ± 3.6 | 65.8 ± 4.7 | 61.5 ± 3.8 | 55.1 ± 6.1 | 44.5 ± 3.7 | 42.8 ± 3.4 |
| | | CRGNN | 75.1 ± 1.1 | 72.9 ± 2 | 68.5 ± 4.2 | 67.6 ± 4.3 | 66.0 ± 4.3 | 58.3 ± 3.6 | 55.1 ± 4.8 | 52.0 ± 2.7 | 45.2 ± 3.6 | 39.2 ± 3 |
| | | RNCGLN | 69.9 ± 3.2 | 68.3 ± 3.2 | 66.1 ± 2.1 | 62.4 ± 3.8 | 58.2 ± 2.8 | 55.8 ± 3.2 | 53.6 ± 2.8 | 47.6 ± 3.7 | 41.9 ± 3 | 37.9 ± 2.9 |
| | | PIGNN | 74.0 ± 2.2 | 72.9 ± 2.7 | 71.1 ± 4.8 | 68.8 ± 4.4 | 66.2 ± 5.6 | 62.7 ± 5.2 | 57.7 ± 7 | 51.3 ± 6.6 | 44.5 ± 7.2 | 40.7 ± 5.6 |
| | | DGNN | 64.0 ± 2.2 | 60.9 ± 4.7 | 56.6 ± 7.7 | 52.8 ± 4.4 | 49.8 ± 5.9 | 49.3 ± 8.3 | 40.6 ± 9 | 36.8 ± 7.3 | 32.6 ± 6.8 | 29.6 ± 7 |
| | | TSS | 77.7 ± 3.1 | 77.8 ± 2.2 | 77.6 ± 1.8 | 75.9 ± 3.2 | 76.3 ± 2.4 | 75.4 ± 3.2 | 70.1 ± 4.9 | 64.5 ± 4.7 | 60.9 ± 6.0 | 46.1 ± 4.7 |
| | | DeGLIF(mv) | 79.8 ± 0.4 | **79.1 ± 1.1** | 78.4 ± 0.4 | 77.5 ± 1 | 76.7 ± 0.9 | 75.0 ± 0.9 | 71.6 ± 1.4 | 66.6 ± 2.5 | 51.5 ± 3.4 | 49.0 ± 4.1 |
| | | DeGLIF(sum) | **80.2 ± 0.9** | 78.8 ± 0.9 | 78.2 ± 1.1 | **78.0 ± 0.8** | **77.8 ± 0.7** | **75.6 ± 1.2** | **73.4 ± 1.2** | **69.6 ± 2.8** | **63.7 ± 3.7** | **54.0 ± 1.8** |
| | Amazon Photo | GCN | 89.5 ± 1 | 88.6 ± 0.9 | 87.0 ± 1.6 | 83.5 ± 1.7 | 82.4 ± 2.4 | 77.6 ± 3.8 | 69.0 ± 5.4 | 60.8 ± 7.9 | 62.4 ± 6.3 | 50.9 ± 8.3 |
| | | Co-teaching + | 86.3 ± 9.5 | 88.6 ± 3.3 | 82.6 ± 8.3 | 84.1 ± 5.8 | 80.0 ± 6.1 | 78.9 ± 2.7 | 70.4 ± 7.2 | 72.9 ± 3.1 | 61.5 ± 7.5 | 53.9 ± 7.2 |
| | | NRGNN | 65.3 ± 5 | 65.6 ± 2.7 | 67.3 ± 10 | 67.3 ± 5.1 | 60.8 ± 6.8 | 65.7 ± 6 | 61.8 ± 4.8 | 52.6 ± 12.2 | 52.2 ± 12 | 47.3 ± 10.1 |
| | | RTGNN | 88.4 ± 1 | 86.6 ± 0.7 | 88.3 ± 1.1 | 87.1 ± 1.3 | 85.2 ± 2.3 | 77.6 ± 1.1 | 73.3 ± 2.4 | **75.2 ± 6.4** | 64.6 ± 3.1 | 44.0 ± 2.8 |
| | | CP | **90.7 ± 1** | **91.3 ± 0.6** | 89.4 ± 1 | **88.8 ± 2.1** | **86.4 ± 2.9** | 80.8 ± 4.5 | 77.9 ± 2.6 | 68.9 ± 5.9 | 59.6 ± 6.3 | 49.4 ± 7.2 |
| | | CGNN | 64.4 ± 30.6 | 61.5 ± 20.2 | 68.1 ± 17.1 | 58.2 ± 20.1 | 48.3 ± 23.9 | 49.3 ± 25.6 | 46.1 ± 20.6 | 41.4 ± 23.3 | 40.1 ± 20.7 | 30.3 ± 13.9 |
| | | CRGNN | 34.9 ± 41.8 | 21.3 ± 37.8 | 21.3 ± 37.8 | 36.0 ± 43.3 | 20.3 ± 35.5 | 42.7 ± 36.5 | 18.2 ± 30.8 | 15.4 ± 24.5 | 25.3 ± 29.6 | 11.1 ± 15 |
| | | RNCGLN | 86.3 ± 2.4 | 84.6 ± 2.8 | 82.4 ± 3.7 | 80.7 ± 3.1 | 77.7 ± 5.1 | 71.2 ± 4.5 | 67.3 ± 12.5 | 55.7 ± 11.2 | 50.9 ± 11.9 | 43.3 ± 6.6 |
| | | PIGNN | 89.4 ± 0.9 | 89.8 ± 0.8 | 88.1 ± 1 | 86.4 ± 1.6 | 81.8 ± 3.4 | 80.1 ± 5.1 | 76.8 ± 3 | 65.7 ± 8.9 | 62.5 ± 7.3 | 49.9 ± 9.1 |
| | | DGNN | 67.4 ± 21.9 | 63.7 ± 23.5 | 62.7 ± 23.5 | 60.0 ± 16.6 | 54.7 ± 16.5 | 50.4 ± 19.5 | 49.2 ± 16.3 | 48.5 ± 9.5 | 43.9 ± 14.7 | 34.6 ± 8.9 |
| | | TSS | 85.9 ± 3.6 | 83.6 ± 4.1 | 85.6 ± 3.3 | 85.9 ± 3.0 | 82.8 ± 3.0 | 80.6 ± 4.6 | 68.7 ± 7.2 | 67.1 ± 9.7 | 50.0 ± 8.6 | 39.4 ± 10.1 |
| | | DeGLIF(mv) | 87.5 ± 0.5 | 87.8 ± 0.5 | 89.8 ± 3.7 | 88.6 ± 1.2 | 82.8 ± 2.4 | 80.3 ± 2.7 | 73.1 ± 5.9 | 64.0 ± 9.5 | 66.1 ± 7.3 | 54.2 ± 8.1 |
| | | DeGLIF(sum) | 89.6 ± 1 | 89.2 ± 1.3 | 89.2 ± 0.2 | 87.4 ± 0.6 | 85.3 ± 2.6 | **83.2 ± 2.1** | 77.0 ± 6.1 | 74.5 ± 7.6 | **71.8 ± 5.4** | **60.6 ± 10** |

community structures. This lack of clear structure limits the effectiveness of CP. This comparison shows that DeGLIF is a more versatile solution across different graph types. In DeGLIF, the influence function allows

detecting noisy training nodes that would otherwise degrade performance on the clean validation set $D_c$. It also motivates a relabeling function for noisy nodes, where relabeling proves more effective than discarding them. Now, follow the additional computation results to test different components of DeGLIF.

## 5.1 Effectiveness of Relabelling function:

To evaluate the effectiveness of the relabeling function, we measure the percentage of correctly identified noisy nodes correctly reassigned to their true class on the Cora dataset with symmetric label noise. Noisy nodes are identified using DeGLIF(sum) with $\mu = 0$, and results (mean $\pm$ std over five runs) are Results (mean (%age $\pm$ std over five runs) are as follows: $79.1 \pm 1.2$ at 10% noise, $78.2 \pm 2.0$ at 20% noise, $72.8 \pm 2.0$ at 30% noise, $67.0 \pm 3.6$ at 40% noise, and $58.4 \pm 3.5$ at 50% noise. Across noise levels, the relabeling function achieves substantially higher accuracy than random guessing (14.3%). The result shows a decreasing trend as noise increases. But, at higher noise levels, more noisy nodes are identified, so overall noise is still reduced (see Figure 2). Additionally, the relabeling function adds negligible overhead, requiring only an $\arg\max$ operation over $K$ elements, where $K$ is the number of classes. We further tested this for class imbalance and overlapping classes setup; the function remains robust to imbalance but is less effective under overlap, though DeGLIF still outperforms baselines (see Appendix D.2).

## 5.2 Size of $D_c$

We analyzed the impact of varying the size of $D_c$ (small clean dataset) on the accuracy of DeGLIF. To nullify the effect of the relabelling function, we have used a binary version of Cora dataset, details about binary dataset is in Appendix D.5. Results are reported in Table 3. At low noise levels, accuracy closely aligns with that of the clean dataset, hence showing minimal sensitivity to $D_c$ size changes. At higher noise levels, we observe accuracy improvement as $D_c$ size increases.

Table 3: Impact of Size of $D_c$ on DeGLIF

| Noise Level | Size of $D_c$ | | | | | |
|:---:|:---:|:---:|:---:|:---:|:---:|:---:|
| | 0.37% | 0.74% | 1.8% | 3.7% | 7.4% | 18.4% |
| 10% | 91.08 | 91.36 | 91.26 | 91.38 | 91.52 | 90.86 |
| 35% | 74.91 | 76.20 | 78.53 | 80.00 | 82.39 | 82.34 |
| 50% | 55.76 | 56.26 | 57.76 | 58.62 | 60.83 | 63.08 |

## 5.3 Successive Applications of DeGLIF

We apply DeGLIF to the noisy dataset $D$ to produce $D^*$, marking one count. In the second count, we again apply DeGLIF to $D^*$ and obtain $D^{**}$; we repeat this for 5 counts. For every count, we observe the fraction of noisy nodes in the training dataset. The behaviour of DeGLIF under successive applications is analysed for the Cora dataset with different noise levels, and the findings are presented in Fig. 2. It is observed that DeGLIF reduces the fraction of noisy nodes. Remarkably, aside from instances with exceptionally high noise, most of the dataset reaches saturation within 2-3 iterations of DeGLIF. Notably, DeGLIF(sum) outperforms DeGLIF(mv) across various noise levels, except for scenarios with 0% noise. The efficacy of the sum-based algorithm stems from its consideration of the magnitude of $I_{up}(-z, v_i)$, leading to improved results.

## 5.4 New Hybrid Noise Robust Models Using DeGLIF

DeGLIF is a denoising technique that first denoises the data and then, in the second step, trains a GCN on the denoised graph. In contrast, most of the methods compared in our paper are end-to-end noise-robust architectures. From a practical standpoint, DeGLIF was not originally designed to compete with these methods, but rather to complement and assist them. The GCN used in Model-2 (Figure 1) of DeGLIF can be replaced with these noise-robust algorithms to potentially boost performance, as accuracy generally decreases with increasing noise in the graph. We experimentally validated this idea of a hybrid model using

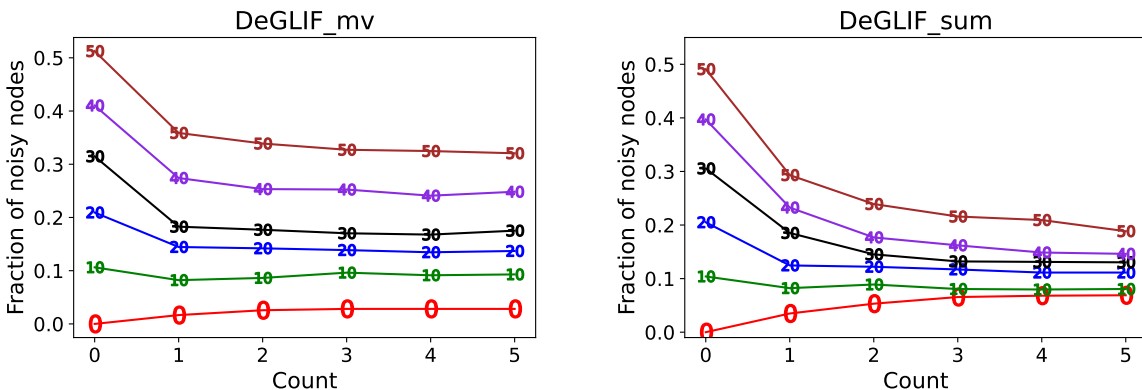

Figure 2: Change in the Fraction of Noisy Training Nodes Plotted Against Increasing Count. Markers Indicate Initial Noise Level. DeGLIF Reduces Noise Levels as Count Increases

TSS. We compare TSS and DeGLIF to a hybrid method (TSS as Model-2 in DeGLIF. We report results on the Cora dataset, using a data split similar to that in the TSS paper.

Table 4: Performance Comparison of the Hybrid Model Against DeGLIF and TSS under Symmetric Label Noise and Pairwise Noise

| Noise Type | Method | 10% | 20% | 30% | 40% | 50% |
|---|---|---|---|---|---|---|
| Symmetric | TSS | 86.0±0.3 | 85.2±0.6 | 83.5±0.7 | 81.4±0.9 | 80.2±0.7 |
| | DeGLIF(sum) | 86.4±0.4 | 85.7±0.5 | 84.3±0.9 | 82.2±0.8 | 79.6±1.7 |
| | Hybrid | 86.1±0.6 | 84.9±0.4 | 84.8±0.8 | 83.7±0.8 | 81.5±1.2 |
| Pairwise | TSS | 85.6±0.6 | 83.0±0.8 | 78.8±0.7 | 68.9±1.3 | 47.0±1.2 |
| | DeGLIF(sum) | 86.4±0.3 | 84.2±1.1 | 78.1±0.8 | 71.4±2.8 | 53.0±4.5 |
| | Hybrid | 85.5±0.5 | 83.4±1.0 | 81.8±0.8 | 74.2±3.3 | 54.7±4.4 |

We observe that this hybrid method improves performance in areas where there is room for enhancement. For example, at 50% pairwise noise, it increases TSS accuracy from 47 to 54.7, which is also higher than the performance achieved by DeGLIF(sum) with GCN.

### 5.5 Role of Hyper-parameters $\lambda$ & $\mu$

When identifying noisy nodes, DeGLIF(mv) uses hyperparameter $\lambda$, and DeGLIF(sum) uses hyperparameter $\mu$ as a threshold. To understand the role of these hyperparameters, we observe accuracy at different threshold levels for a particular noise level, and repeat for all noise levels. The trends for the Cora dataset with GCN at noise levels 5% and 50% are plotted in Figure 3 (5% to 50% on a gap of 5% are reported in Figure 6 and 7 in Appendix D.9). For DeGLIF(sum) we vary $\mu$ over the set $\{0, 0.1, 1, 10, 20\}$ and take $|D_c| = 50$. For DeGLIF(mv) we take $\lambda \in \{0.5, 0.52, 0.53, 0.55, 0.56, 0.6\}$. For a small size of $D_c$, the difference between $\lambda_1 \times |D_c|$ and $\lambda_2 \times |D_c|$ is very small. So, to observe a clear trend with change in $\lambda$, specifically for this experiment, we choose $|D_c| = 400$.

It is evident that, for a lower noise level, the maximum accuracy is achieved with the higher values of $\lambda$ and $\mu$. One explanation for this phenomenon is that there are fewer noisy labels in such cases, making the method (DeGLIF) more susceptible to false positives compared to scenarios with a higher number of noisy data points. As noise increases, flipping labels of more nodes becomes beneficial, as illustrated by the plots. Consequently, as the noise level increases, the optimal value of $\lambda$ and $\mu$ for achieving maximum accuracy tends to decrease. It is worth mentioning that lower values of these thresholds lead to more points being predicted as noisy and, hence, their label being flipped. Also, one can observe that for larger noise levels, the

accuracy is more sensitive to hyperparameters. We can observe that the range of accuracy values is larger for larger noise levels. Similar trends were observed observed for other datasets and architectures.

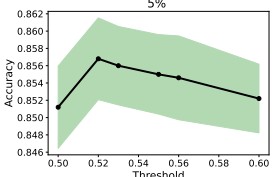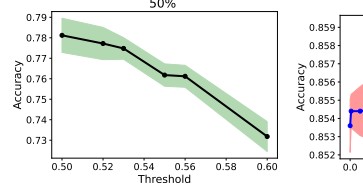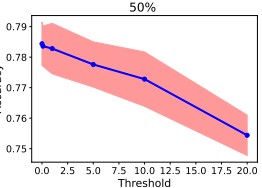

Figure 3: Relation Between DeGLIF's Hyperparameters and Noise Level. First two (green shaded): shows results for $\lambda$ (DeGLIF(mv)). Last two (red shaded): shows results for $\mu$ (DeGLIF(sum)). The Solid Line Denotes Accuracy and the Shaded Region Denotes the 1-Confidence Interval.

## 6 COMPUTATIONAL COMPLEXITY

We acknowledge that the influence function calculation is computationally expensive. The main computational cost of DeGLIF lies in the influence function (Equation 4), which requires computing and inverting a $p \times p$ Hessian matrix ($p = |\theta|$). Complexity for computing hessian matrix is $O(np^2)$, and inverting it, has complexity $O(p^3)$ (Koh & Liang, 2017). In our implementation, we utilize `torch.linalg.inv` for Hessian inversion, which is highly optimized and leverages parallelism to accelerate execution. For graphs with high-dimensional node features, $p$ can be large, making inversion memory-intensive on GPU and slow on CPU. While efficient approximations of influence function, such as Hessian-Vector Products (HVPs), are an active area of research for i.i.d. data (Koh et al., 2024; Lyu et al., 2023), these methods are not directly useful for GNNs. Influence functions on graphs involve additional graph-specific terms arising from message passing and neighborhood dependencies, which existing i.i.d. approximations fail to capture. To our knowledge, no well-established or efficient Hessian approximation currently exists for graph-structured models, and developing one remains an open research problem. Any future progress in this domain can be directly integrated into the DeGLIF framework to further reduce costs.

After influence values are obtained, noisy node detection (Equation 5) requires computing gradients for a small clean set $D_c$ (fixed at 50 in our experiments, complexity of $O(p)$) and performing $|D_c| \times n$ dot products with training nodes. This results in $O(n \times p)$ complexity, but the operations are easily parallelizable on GPU. Relabeling adds only an $\arg\max$ over $K$ classes and thus negligible overhead. Hence, using faster approximations to influence the function can lead to faster DeGLIF.

Table 5: Execution Time comparison of Noise-Robust Algorithms for Cora Dataset

| Algorithm | Time (s) | Algorithm | Time (s) | Algorithm | Time (s) |
|-----------|----------|-----------|----------|-----------|----------|
| NRGNN | 15.75 | CRGNN | 5.40 | RTGNN | 25.24 |
| RNCGLN | 14.69 | CP | 12.57 | PIGNN | 7.53 |
| CGNN | 11.50 | DGNN | 4.53 | DeGLIF | 68.03 |

Despite these theoretical costs, DeGLIF remains practical for graphs with a large number of nodes. DeGLIF is designed as a *one-time preprocessing framework.* The overhead of influence calculation and noise identification occurs only once; subsequently, the final GNN is trained normally on the denoised graph without any additional cost. We tested the scalability of DeGLIF on a large-scale dataset: the OGBN-Arxiv dataset (Hu et al., 2020), with default splits. None of the baselines that we compared with have reported results on the OGBN-Arxiv dataset; so, we performed a run-time comparison. Experiments were conducted on an NVIDIA A5000 (24 GB) GPU. For OGBN-Arxiv, we utilized a GCN with a single hidden layer of dimension 64 for both Model-1 and Model-2 in the DeGLIF pipeline. The results in Table 6 reveal a critical finding: while DeGLIF incurs a higher runtime than the baseline GCN (due to influence calculation), it successfully completes execution on a 24GB GPU. In contrast, many competing state-of-the-art algorithms, including

PIGNN, RNCGLN, CGNN, CRGNN, and DGNN, failed to scale to this dataset, resulting in Out-Of-Memory (OOM) errors. Similarly, RNCGLN fails to scale to the PubMed dataset on a 24 GB GPU (see section D.7).

Table 6: Runtime comparison on OGBN-Arxiv (169k nodes). OOM indicates Out-Of-Memory.

| Algorithm | Time (s) | Algorithm | Time (s) | Algorithm | Time (s) |
|-----------|----------|-----------|----------|-----------|----------|
| NRGNN | 939 | CRGNN | OOM | RTGNN | 1917 |
| RNCGLN | OOM | CP | 1066 | PIGNN | OOM |
| CGNN | OMM | DGNN | OOM | DeGLIF | 1389 |

Thus, while DeGLIF incurs a higher initial cost than simple baselines, the substantial gains in robustness and the ability to scale to large graphs justify this one-time investment.

## 7 RELATED WORKS

**Label Noise Problem** is an important problem (Tripathi & Hemachandra, 2020), and common methods to tackle it involve: **1.** Identifying and eliminating noisy points (Tripathi & Hemachandra, 2020; Malach & Shalev-Shwartz, 2017); this method is not helpful for small-size datasets as we may end up eliminating a lot of important training data. **2.** Using noise tolerant algorithm (Tripathi & Hemachandra, 2019; Kumar & Sastry, 2018): researchers have focused on finding noise robust loss functions that are able to learn to predict well on clean test data (Manwani & Sastry, 2013). For example, hinge loss and exponential loss are not noise robust loss functions for binary classification under the SLN noise model, whereas 0-1 loss and squared error loss with linear classifier are noise robust losses (Tripathi & Hemachandra, 2019; Sastry & Manwani, 2017). Results of these kinds were extended for class conditional noise for binary datasets and then to multiclass datasets. Although for neural networks, commonly used mean squared error and cross-entropy loss are not noise-robust (Tripathi & Hemachandra, 2020), many modifications have been proposed (with empirical and theoretical evidence), which modify the cross-entropy loss to noise-robust loss function. Examples include Robust log loss (Kumar & Sastry, 2018), Symmetric cross entropy (Wang et al., 2019), Generalised cross entropy (Ghosh et al., 2015), etc. **3.** Denoising data: it involves identifying noisy data point and try to provide them with correct labels (Dai et al., 2021; Qian et al., 2023). DeGLIF is based on third approach.

**GNN with Noisy Labels:** GNN has gained recent attention because of their wide application and effectiveness on relational data (Kipf & Welling, 2017; Defferrard et al., 2016; Fey & Lenssen, 2019; Hamilton et al., 2017), with GCN (Kipf & Welling, 2017) being one the most common message passing algorithms. Prior research have explored learning in the presence of noise for graph data. Among them **D-GNN** (NT et al., 2019) uses backward loss correction, NRGNN (Dai et al., 2021) connects unlabelled nodes to labelled nodes with high feature similarity, facilitating the acquisition of accurate pseudo labels for enhanced supervision and reduction of label noise effects. **Coteaching+** (Yu et al., 2019) improves model resilience to noisy labels by training two networks concurrently and dynamically updating the training set based on each network's prediction confidence. **RTGNN** (Qian et al., 2023) adapts three key steps: creating bridges between labeled and unlabeled nodes to enhance information flow, employing dual graph convolutional networks to identify and mitigate noisy labels, and utilizing deep learning's memory for self-correction and consistency enforcement across various data perspectives. **CP** (Zhang et al., 2020) addresses label noise in GNNs by proposing a defense mechanism against adversarial label-flipping attacks. CP leverage a label smoothness assumption to detect and mitigate noisy labels, ensuring consistency between node labels and the graph structure while training the GNN. **RNCGLN** (Zhu et al., 2024) uses a pseudo-labeling technique within a self-training framework to identify and correct noisy labels. This is achieved by constructing a classifier that predicts labels for all nodes (labeled and unlabeled), and then replacing original labels with low predictive confidence as these are considered to be noise. **PIGNN** (Du et al., 2021) leverages pairwise interactions (PI) between nodes, which are less susceptible to noise than individual node labels. It incorporates a confidence-aware PI estimation model that dynamically determines PI labels from graph structure. These PI labels are then used to regularize a separate node classification model, ensuring that nodes with strong PI connections have similar embeddings. **CGNN** (Yuan et al., 2023) utilizes two main strategies to handle noisy labels in graph data. First, it uses graph contrastive learning as a regularization technique. This encourages the model to

learn consistent node representations even when trained on augmented versions of the graph, thus enhancing its robustness against label noise. Second, CGNN employs a sample selection method that leverages the homophily assumption, which states that connected nodes tend to have similar labels. By identifying nodes whose labels are inconsistent with their neighbours, CGNN pinpoints and corrects potentially noisy labels. **CRGNN** (Li et al., 2024) utilizes a combination of contrastive learning and a dynamic cross-entropy loss. Unsupervised and neighborhood contrastive losses, informed by graph homophily, encourage robust feature representations. A dynamic cross-entropy loss, which focuses on nodes with consistent predictions across augmented views, further mitigates the negative impacts of noise. **TSS** (Wu et al., 2024)addresses label noise in graphs by introducing Class-conditional Betweenness Centrality (CBC), a topology-aware measure that identifies node positions relative to class boundaries. TSS applies an easy-to-hard curriculum, initially selecting clean nodes far from boundaries and gradually incorporating harder boundary-near nodes. This process ensures a noise-tolerant training, tailored to graph structure. Our work is different as none of these methods uses the influence function (which helps identify noisy points) for denoising.

**Influence Function:** The idea of the influence function dates back to the 70s (Hampel, 1974; Jaeckel, 1972); it started with the idea of removing training points from linear statistical models. For deep learning models, it was first proposed by Koh & Liang (2017). This was further extended to capturing group impact (Koh et al., 2019) and resolving training bias for i.i.d. dataset (Kong et al., 2022). For graph data, Chen et al. (2023) used influence to approximate change in model parameters of GNNs. Recently influence idea in graph have been used for rectifying harmful edges (Song et al., 2023), graph unlearning (Wu et al., 2023), but not used to solve the label noise problem for graph data. As far as we know, this is the first work on the intersection of graph data, the label noise problem, and the influence function.

## 8 DISCUSSION

In this paper, we address the problem of node classification for graph data with noisy node labels using the leave-one-out influence function. The idea of DeGLIF is to identify noisy nodes (removing which leads to a lower loss on a small clean dataset $D_c$). As retraining the model by removing each node and repeating this for every node is computationally infeasible, we approximate the change in loss on $D_c$ using the influence function. We acknowledge that the influence function involves a computational cost due to Hessian operations; however, as this inversion is a one-time step, DeGLIF remains scalable to large graphs (e.g., OGBN-Arxiv) where other baselines encounter memory limitations. This work serves as a first attempt to leverage influence functions for label noise robustness for graph data, yielding positive results. Furthermore, the computational cost can be mitigated by future research into faster influence approximations for graph data, which currently remains an open problem After identifying noisy nodes, DeGLIF uses a theoretically motivated relabeling function to denoise noisy nodes. Through extensive experimentation, we demonstrate the effectiveness of our method. DeGLIF requires no prior information about the noise level or noise model, nor does it estimate noise level. DeGLIF performs well on graphs with varying properties, including variation in node degree and training set sizes. DeGLIF is also indifferent to the choice of the loss function, allowing it to complement the area of noise-robust loss functions. Another highlight is that DeGLIF can be used in conjunction with any GNN model with a Hessian of regularised loss function and can be applied to a variety of datasets. Code is available in the supplementary material

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

# DeGLIF for Label Noise Robust Node Classification using GNNs: Supplementary Materials

## A ALGORITHMS FOR DeGLIF(mv) and DeGLIF(sum)

---

**Algorithm 2** DeGLIF(mv)

---

**Input:** $I_{\text{up}}(-z_i, v_j)$, $\lambda$
**Output:** $D_i$
**foreach** $v_j \in D_c$ **do**
  **if** $I_{up}(-z_i, v_j) < 0$ **then**
  | $I_{i,j} \leftarrow 1$
  **else**
  └ $I_{i,j} \leftarrow 0$
$C_i \leftarrow (\sum_j I_{i,j})/\text{size}(D_c)$ **if** $C_i > \lambda$ **then**
| $D_i \leftarrow 1$
**else**
└ $D_i \leftarrow 0$

---

---

**Algorithm 3** DeGLIF(sum)

---

**Input:** $I_{\text{up}}(-z_i, v_j)$, $\mu$
**Output:** $D_i$
**if** $\sum_{v_j \in D_c} -I_{up}(-z_i, v_j) > \mu$ **then**
└ $D_i \leftarrow 1$
**else**
└ $D_i \leftarrow 0$

---

## B NOISE MODELS USED

Let $G = (V, E)$ be a graph where each node belongs to one of $K$ classes. Noise models used to add noise to node labels are described as follows:

### B.1 Symmetric Label Noise (SLN)

SLN (Tripathi & Hemachandra, 2020; 2019; Ghosh et al., 2015) refers to a type of label noise where the mislabelled samples are equally likely to be assigned any of the possible labels. Mathematically, if the true label of a sample is denoted by $y$ and its observed (noisy) label is denoted by $y'$, then the probability of mislabelling a sample as any of the possible labels is the same. This can be represented as $P(y' = l|y = k) = c$, where $l$ and $k$ represent the possible labels and $c$ is a constant probability value. Transition probability matrix for SLN is given by

$$Q_{sln} = \begin{bmatrix} 1-(K-1)c & c & c & \ldots & c \\ c & 1-(K-1)c & c & \ldots & c \\ \vdots & \ddots & \ddots & \ddots & \vdots \\ c & \ldots & \ddots & 1-(K-1)c & c \\ c & c & \ldots & c & 1-(K-1)c \end{bmatrix}$$

## B.2 Class Conditional Noise (CCN)

In Class Conditional Noise (CCN) (Tripathi & Hemachandra, 2020; 2019; Ghosh et al., 2015), the probability with which the label is changed depends on both $y$ and $y'$. The probability of a node of class $l$ being reassigned to class $k$ is given by $c_{lk}$ ($P(y' = l|y = k) = c_{lk}$), where $l \neq k$. So, a node with label $l$ is flipped with probability $c_l = \sum_{i=1}^{K} c_{li}$; $i \neq l$ and the label is retained with the probability $1 - c_l$. This is also referred to as random noise and asymmetric noise. The transition probability matrix is given by

$$
Q = \begin{bmatrix}
1 - c_1 & c_{12} & c_{13} & \dots & c_{1K} \\
c_{21} & 1 - c_2 & c_{23} & & c_{2K} \\
\vdots & & \ddots & \ddots & \vdots \\
c_{(K-1)1} & & & 1 - c_{K-1} & c_{(K-1)K} \\
c_{K1} & c_{K2} & \dots & c_{K(K-1)} & 1 - c_K
\end{bmatrix}
$$

### B.2.1 Pairwise Noise

CCN is a very broad class of noise model, and it is difficult to compare noise robust algorithms on CCN because of too many possible combinations of $c_{ij}$. We similar to other works on noise-robust learning ((Yu et al., 2019),(Dai et al., 2021),(Qian et al., 2023), (Zhang et al., 2020), (Yuan et al., 2023), (Li et al., 2024), (Zhu et al., 2024), (Du et al., 2021), (NT et al., 2019)), use a special class of CCN known as Pairwise Noise. . The motivation behind Pairwise Noise is that one is more likely to mislabel two similar classes. For Pairwise Noise $c_1 = c_2 = \dots = c_K = c$, and the label is flipped to the next label (with probability $c$). The transition probability matrix is given by

$$
Q_{pw} = \begin{bmatrix}
1 - c & c & 0 & \dots & 0 \\
0 & 1 - c & c & & 0 \\
\vdots & & \ddots & \ddots & \vdots \\
0 & & & 1 - c & c \\
c & 0 & \dots & 0 & 1 - c
\end{bmatrix}
\tag{6}
$$

## C   DERIVATIONS AND PROOFS

### C.1   Derivation of Leave-One-Out Influence Function

Define $R(\theta) := \frac{1}{n} \sum_{i=1}^{n} L(z_i, \theta)$, where $z_i = (x_i, y_i)$ is a training point and $\theta$ is model parameter. Then $H_\theta = \frac{1}{n} \sum_{i=1}^{n} \nabla_\theta^2 L(z_i, \theta)$.

Now let us define

$$\hat{\theta} := \mathrm{argmin}_{\theta \in \Theta} R(\theta)$$

also define

$$\hat{\theta}_{\epsilon,z} := \mathrm{argmin}_{\theta \in \Theta}(R(\theta) + \epsilon L(z, \theta)) \tag{7}$$

We want to estimate $\Delta_\epsilon = \hat{\theta}_{\epsilon,z} - \hat{\theta}$. Using the first-order optimality condition on 7,

$$\nabla_\theta R(\hat{\theta}_{\epsilon,z}) + \epsilon \nabla_\theta L(z, \hat{\theta}_{\epsilon,z}) = 0$$

Using Taylor series expansion on L.H.S. gives (for this proof, from here on we use $\nabla$ for $\nabla_\theta$)

$$0 \approx (\nabla R(\hat{\theta}) + \epsilon \nabla L(z, \hat{\theta})) + \Delta_\epsilon(\nabla^2 R(\hat{\theta}) + \epsilon \nabla^2 L(z, \hat{\theta}))$$

As $\hat{\theta}$ is optimal for $R(\theta)$, $\nabla_\theta R(\hat{\theta})$ becomes zero, then solving for $\Delta_\epsilon$ gives

$$
\begin{aligned}
\Delta_\epsilon &\approx (\nabla^2 R(\hat{\theta}) + \epsilon \nabla^2 L(z, \hat{\theta}))^{-1}(-\epsilon \nabla L(z, \hat{\theta})) \\
&= (1 + \epsilon \nabla^2 L(z, \hat{\theta}) \nabla^2 R(\hat{\theta})^{-1})^{-1} \nabla^2 R(\hat{\theta})^{-1}(-\epsilon \nabla L(z, \hat{\theta})) \\
&= (1 - \epsilon \nabla^2 L(z, \hat{\theta}) \nabla^2 R(\hat{\theta})^{-1} + (\epsilon \nabla^2 L(z, \hat{\theta}) \nabla^2 R(\hat{\theta})^{-1})^2 \\
&\quad - (\epsilon \nabla^2 L(z, \hat{\theta}) \nabla^2 R(\hat{\theta})^{-1})^3 + \ldots) \nabla^2 R(\hat{\theta})^{-1}(-\epsilon \nabla L(z, \hat{\theta}))
\end{aligned}
$$

As $\epsilon \to 0$, keeping only $O(\epsilon)$ term we obtain.

$$
\Delta_\epsilon \approx \nabla^2 R(\hat{\theta})^{-1}(-\epsilon \nabla L(z, \hat{\theta})) \tag{8}
$$

If we now choose $\epsilon$ to be $-\frac{1}{n}$ in Equation (7), then it is equivalent to removing a data point from the training set. So, if $\hat{\theta}_{-z}$ represents the optimal parameter obtained when the model is trained after removing $z$ and using Equation (8), then we have that

$$
I(-z) := \hat{\theta}_{-z} - \hat{\theta} = \Delta_{-\frac{1}{n}} \approx \frac{1}{n} \nabla^2 R(\hat{\theta})^{-1} \nabla L(z, \hat{\theta}) = \frac{1}{n} H_{\hat{\theta}}^{-1} \nabla_\theta L(z, \hat{\theta})
$$

## C.2 Derivation of Equation 5

As per the notation used in the paper, the graph under consideration has $n$ nodes and $\hat{\theta}$ represents the optimal parameters of a GNN model trained on the noisy training dataset. Let us assume that we want to up-weight a node $z$ by factor $\epsilon$; which means the loss function gets modified from $\frac{1}{n} \sum_{i=1}^n L(z_i, \theta)$ to $(\sum_{i=1}^n L(z_i, \theta)) + \epsilon \times L(z, \theta)$ and all the edges connected to $z$ gets up-weighted from 1 to $1 + n \times \epsilon$. Observe that removing a node is equivalent to choosing $\epsilon = -\frac{1}{n}$, which removes $z$ from the loss term and makes all edge weights connected to $z$ as 0.

$I_{up}(-z, v_i)$ represents the change in loss of validation node $v_i$ (we call it validation node as its label is not observed during training), when node $z$ is removed.

$$
I_{up}(-z, v_i) := \frac{dL(v_i, \theta_{\epsilon, z})}{d\epsilon}\Big|_{\epsilon = -1/n}
$$

that is change is loss of $v_i$ with respect in change in $\epsilon$ calculated at $\epsilon = -1/n$. $L$ is a function of $v_i$ and $\theta_{\epsilon, z}$. $\theta_{\epsilon, z}$ (optimal parameter) is a function of $\epsilon$ and $z$. Also observe that $\theta_{\epsilon, z}$ is list of all parameters and hence is a vector ($\theta = \theta_1, \ldots, \theta_k$).

$$
\frac{dL(v_i, \hat{\theta}_{\epsilon, z})}{d\epsilon}\Big|_{\epsilon = -1/n} = \left(\sum_{i=1}^k \frac{dL}{d\theta_i} \frac{d\theta_i}{d\epsilon}\right)\Big|_{\epsilon = -1/n} = \nabla_\theta L(v_i, \hat{\theta})^\top \frac{d\theta_{\epsilon, z}}{d\epsilon}\Big|_{\epsilon = -1/n}
$$

$\frac{d\theta_{\epsilon, z}}{d\epsilon}\Big|_{\epsilon = -1/n}$ is the change in the model parameter if a training node is dropped, is influence function $I(-z)$ which we have already derived in Equation (4). So, Equation (5) is

$$
\nabla_\theta L(v_i, \hat{\theta})^\top \frac{d\theta_{\epsilon, z}}{d\epsilon}\Big|_{\epsilon = -1/n} = \nabla_\theta L(v_i, \theta)^\top I(-z).
$$

### C.3 Proof of Theorem 1

*Proof.* Using Equation (5) and assuming each training node independently influences test risk

$$
\begin{aligned}
R(\hat{\theta}, D_c) - R(\hat{\theta}_{-D_n}, D_c) &= \frac{1}{n} \sum_{z \in D_n} \sum_{v_i \in D_c} L(v_i, \hat{\theta}) - L(v_i, \hat{\theta}_{-z}) \\
&\approx \frac{1}{n} \sum_{z \in D_n} \sum_{v_i \in D_c} (-I_{up}(-z, v_i)) \\
&\geq \frac{1}{n} \sum_{z \in D_n} \mu \\
&\geq 0
\end{aligned}
$$

□

### C.4 Approximating the Impact of Relabelling on Loss of a Clean Data Point

Let us assume that the node $z = (x, y)$ is relabelled as $z_\delta = (x, y_\delta)$. This can be viewed as removing $z$ and adding $z_\delta$ in training data. Now using Equation (2)

$$
I(z \to z_\delta) = I(-z, +z_\delta) = I(-z) - I(-z_\delta) \tag{9}
$$

Now using Equations (4) and (9)

$$
\begin{aligned}
I(z \to z_\delta) &= \frac{1}{n} H_\theta^{-1} (\nabla_\theta L(z, \theta) - \sum_{k \in V_{train}} (\nabla_\theta L((M_k + \Delta_k, y_k), \theta) - \nabla_\theta L((M_k, y_k), \theta))) \\
&\quad - \frac{1}{n} H_\theta^{-1} (\nabla_\theta L(z_\delta, \theta) + \sum_{k \in V_{train}} (\nabla_\theta L((M_k + \Delta_k, y_k), \theta) - \nabla_\theta L((M_k, y_k), \theta))) \\
&= \frac{1}{n} H_\theta^{-1} (\nabla_\theta L(z, \theta) - \nabla_\theta L(z_\delta, \theta))
\end{aligned} \tag{10}
$$

Now using Equation (5), the change in test loss on $D_c$ is given by

$$
\begin{aligned}
I_{up}(z \to z_\delta, v_i) &= \nabla_\theta L(v_i, \hat{\theta})^\top I(z \to z_\delta) \\
&= \frac{1}{n} \nabla_\theta L(v_i, \hat{\theta})^\top H_\theta^{-1} (\nabla_\theta L(z, \theta) - \nabla_\theta L(z_\delta, \theta)).
\end{aligned} \tag{11}
$$

This is useful in proving Theorem 3 and 2

### C.5 Theorem 3 : Relabeling can Lead to a Lower Test Risk

**Theorem 3.** *For binary labelled dataset $\{z_i = (x_i, y_i)\}$, where $y_i \in \{0, 1\}$. Let the relabelling function be $r(z_i) = 1 - y_i$, and let $\hat{\theta}_r$ denote the optimal parameter when the model is trained on relabelled data then, $R(\hat{\theta}_{-D_n}, D_c) - R(\hat{\theta}_r, D_c) \geq 0$*

*Proof.* The last layer in GNN has nodes equal to the number of classes; for binary labelled dataset, let it be given by the vector $[\phi(x_i), 1 - \phi(x_i)]^\top$. Here $\phi(x_i)$ denotes probability that label is class 1. For a node $z_i = (x_i, y_i)$ loss function used (cross-entropy), in binary setup, reduces to

$$
L(z_i) := L(z_i, \theta) = -y_i \log(\phi(x_i)) - (1 - y_i) \log(1 - \phi(x_i)) \tag{12}
$$

If initially $y_i = 1$ then $L(z_i, \theta) = -\log(\phi(x_i))$; after relabelling the loss changes to $L(z_\delta, \theta) = -\log(1 - \phi(x_i))$. Then the Equation (11) in this case becomes

$$I_{up}(z \to z_\delta, v_i) = \frac{1}{n} \nabla_\theta L(v_i, \hat{\theta})^\top H_\theta^{-1} (\nabla_\theta L(z, \theta) - \nabla_\theta L(z_\delta, \theta))$$

$$= \frac{1}{n} \nabla_\theta L(v_i, \hat{\theta})^\top H_\theta^{-1} (-\nabla_\theta \log(\phi) + \nabla_\theta \log(1 - \phi))$$

$$= \frac{1}{n} \nabla_\theta L(v_i, \hat{\theta})^\top H_\theta^{-1} \left( -\frac{\nabla_\theta \phi}{\phi} - \frac{\nabla_\theta \phi}{1 - \phi} \right)$$

$$= \frac{1}{n} \nabla_\theta L(v_i, \hat{\theta})^\top H_\theta^{-1} \left( -\frac{\nabla_\theta \phi}{\phi} \right) \left( 1 + \frac{\phi}{1 - \phi} \right) \qquad (13)$$

$$= \frac{1}{n} \nabla_\theta L(v_i, \hat{\theta})^\top H_\theta^{-1} \left( -\frac{\nabla_\theta \phi}{\phi} \right) \left( \frac{1}{1 - \phi} \right)$$

$$= \frac{1}{n} \nabla_\theta L(v_i, \hat{\theta})^\top H_\theta^{-1} \frac{\nabla_\theta L(z, \theta)}{1 - \phi}$$

$$= \frac{I_{up}(-z, v_i)}{1 - \phi}$$

Now, let us consider the case when $y_i = 0$ initially and was relabelled as 1. Then the loss changes from $L(z, \theta) = -\log(1 - \phi(x_i))$ to $L(z_\delta, \theta) = -\log(\phi(x_i))$ and the Equation (11) in this case becomes

$$I_{up}(z \to z_\delta, v_i) = \frac{1}{n} \nabla_\theta L(v_i, \hat{\theta})^\top H_\theta^{-1} (\nabla_\theta L(z, \theta) - \nabla_\theta L(z_\delta, \theta))$$

$$= \frac{1}{n} \nabla_\theta L(v_i, \hat{\theta})^\top H_\theta^{-1} (-\nabla_\theta \log(1 - \phi) + \nabla_\theta \log(\phi))$$

$$= \frac{1}{n} \nabla_\theta L(v_i, \hat{\theta})^\top H_\theta^{-1} \left( \frac{\nabla_\theta \phi}{1 - \phi} + \frac{\nabla_\theta \phi}{\phi} \right)$$

$$= \frac{1}{n} \nabla_\theta L(v_i, \hat{\theta})^\top H_\theta^{-1} \left( \frac{\nabla_\theta \phi}{1 - \phi} \right) \left( 1 + \frac{1 - \phi}{\phi} \right) \qquad (14)$$

$$= \frac{1}{n} \nabla_\theta L(v_i, \hat{\theta})^\top H_\theta^{-1} \left( \frac{\nabla_\theta \phi}{1 - \phi} \right) \left( \frac{1}{\phi} \right)$$

$$= \frac{1}{n} \nabla_\theta L(v_i, \hat{\theta})^\top H_\theta^{-1} \frac{\nabla_\theta L(z, \theta)}{\phi}$$

$$= \frac{I_{up}(-z, v_i)}{\phi}$$

Now,

$$R(\hat{\theta}_{-D_n}, D_c) - R(\hat{\theta}_r, D_c) = R(\hat{\theta}_{-D_n}, D_c) - R(\hat{\theta}, D_c) + R(\hat{\theta}, D_c) - R(\hat{\theta}_r, D_c)$$

$$= \frac{1}{n} \sum_{z \in D_n} \sum_{v_i \in D_c} L(v_i, \hat{\theta}_{-z}) - L(v_i, \hat{\theta}) + L(v_i, \hat{\theta}) - L(v_i, \hat{\theta}_r)$$

$$\approx \frac{1}{n} \sum_{z \in D_n} \sum_{v_i \in D_c} I_{up}(-z, v_i) - I_{up}(z \to z_\delta, v_i) \qquad (15)$$

$$= \frac{1}{n} \sum_{z \in D_n} \sum_{v_i \in D_c} (-I_{up}(-z, v_i)) \left( \frac{I_{up}(z \to z_\delta, v_i)}{I_{up}(-z, v_i)} - 1 \right)$$

Using Equation (17) and (14);

$$\frac{I_{up}(z \to z_\delta)}{I_{up}(-z, v_i)} - 1 = \begin{cases} \frac{\phi}{1 - \phi} & \text{if } y_i = 1 \\ \frac{1 - \phi}{\phi} & \text{if } y_i = 0 \end{cases}$$

As in both cases the value is positive take $c_z = \min\{\frac{\phi}{1-\phi}, \frac{1-\phi}{\phi}\}$ (see that $c_z \geq 0$), then the Equation (18) becomes

$$R(\hat{\theta}_{-D_n}, D_c) - R(\hat{\theta}_r, D_c) \approx \frac{1}{n} \sum_{z \in D_n} \sum_{v_i \in D_c} (-I_{up}(-z, v_i)) \left( \frac{I_{up}(z \to z_\delta)}{I_{up}(-z, v_i)} - 1 \right)$$

$$\geq \frac{1}{n} \sum_{z \in D_n} c_z \sum_{v_i \in D_c} (-I_{up}(-z, vi)) \tag{16}$$

$$\geq \frac{1}{n} \sum_{z \in D_n} c_z \times \mu$$

$$\geq 0$$

$\square$

## C.6 Proof of Theorem 2

*Proof.* Let us assume $z = (x_i, y_i = m)$ is predicted noisy via influence calculation. Let $f(x_i) = [f(x_i)_1, \ldots, f(x_i)_j]$ be prediction made by GNN. If we relabel $z$ to $[0, \ldots, \varphi_k, \ldots, 0]$, where $\varphi_k$ is at $k-$th position and is given by $\log_{f(x_i)_k}(1 - f(x_i)_m)$. Then the cross entropy loss changes from $L(z_i, \theta) = -\log(f_m)$ to $L(z_\delta, \theta) = -\log_{f_k}(1 - f_m) \log f_k = -\frac{\log(1-f_m)}{\log f_k} \times \log f_k = -\log(1 - f_m)$. Then using similar approach as in proof of Theorem 3, Equation (11) for this case becomes

$$\begin{aligned} I_{up}(z \to z_\delta, v_i) &= \frac{1}{n} \nabla_\theta L(v_i, \hat{\theta})^\top H_\theta^{-1} (\nabla_\theta L(z, \theta) - \nabla_\theta L(z_\delta, \theta)) \\ &= \frac{1}{n} \nabla_\theta L(v_i, \hat{\theta})^\top H_\theta^{-1} (-\nabla_\theta \log(f_m) + \nabla_\theta \log(1 - f_m)) \\ &= \frac{1}{n} \nabla_\theta L(v_i, \hat{\theta})^\top H_\theta^{-1} \left( -\frac{\nabla_\theta f_m}{f_m} - \frac{\nabla_\theta f_m}{1 - f_m} \right) \\ &= \frac{1}{n} \nabla_\theta L(v_i, \hat{\theta})^\top H_\theta^{-1} \left( -\frac{\nabla_\theta f_m}{f_m} \right) \left( 1 + \frac{f_m}{1 - f_m} \right) \\ &= \frac{1}{n} \nabla_\theta L(v_i, \hat{\theta})^\top H_\theta^{-1} \left( -\frac{\nabla_\theta f_m}{f_m} \right) \left( \frac{1}{1 - f_m} \right) \\ &= \frac{1}{n} \nabla_\theta L(v_i, \hat{\theta})^\top H_\theta^{-1} \frac{\nabla_\theta L(z, \theta)}{1 - f_m} \\ &= \frac{I_{up}(-z, v_i)}{1 - f_m} \end{aligned} \tag{17}$$

Now,

$$
\begin{aligned}
R(\hat{\theta}_{-D_n}, D_c) - R(\hat{\theta}_r, D_c) &= R(\hat{\theta}_{-D_n}, D_c) - R(\hat{\theta}, D_c) + R(\hat{\theta}, D_c) - R(\hat{\theta}_r, D_c) \\
&= \frac{1}{n} \sum_{z \in D_n} \sum_{v_i \in D_c} L(v_i, \hat{\theta}_{-z}) - L(v_i, \hat{\theta}) + L(v_i, \hat{\theta}) - L(v_i, \hat{\theta}_r) \\
&\approx \frac{1}{n} \sum_{z \in D_n} \sum_{v_i \in D_c} I_{up}(-z, v_i) - I_{up}(z \to z_\delta) \\
&= \frac{1}{n} \sum_{z \in D_n} \sum_{v_i \in D_c} (-I_{up}(-z, v_i)) \left( \frac{I_{up}(z \to z_\delta)}{I_{up}(-z, v_i)} - 1 \right) \\
&= \frac{1}{n} \sum_{z \in D_n} \sum_{v_i \in D_c} (-I_{up}(-z, v_i)) \frac{f_{m_z}}{1 - f_{m_z}} - 1 \\
&= \frac{1}{n} \sum_{z \in D_n} \frac{f_{m_z}}{1 - f_{m_z}} \sum_{v_i \in D_c} (-I_{up}(-z, v_i)) \\
&\geq \frac{1}{n} \sum_{z \in D_n} \frac{f_{m_z}}{1 - f_{m_z}} \mu \\
&\geq 0
\end{aligned}
\tag{18}
$$

$\square$

# D ADDITIONAL EXPERIMENTS

## D.1 Empirical Validation of the Additivity Assumption of Influence Function

The theoretical frameworks presented in Theorems 1 and 2 rely on the additivity assumption, which posits that the influence of a group of nodes can be approximated by summing their individual influences. In the context of graph data, this assumption requires careful consideration. Especially when removed nodes share neighborhood structures, the difference $I(-z_i, z_j) - I(-z_i) - I(z_j)$ may not be zero. To quantify the impact of ignoring these interactions and to validate the practical utility of our method, we designed an experiment comparing the influence-predicted (with the additivity assumption) change in loss against the actual change observed after retraining.

### Experimental Setup and Result

To evaluate the impact of removing groups of size $k$, we randomly sampled 100 distinct subsets of training nodes, each containing $k$ nodes. For each subset, the predicted change in validation loss was computed as the sum of the individual node influences: $I(-z_1, \ldots, -z_k) = I(-z_1) + \cdots + I(-z_k)$. To determine the true, ground-truth change in validation loss, we removed the designated nodes and their associated edges in each subset and retrained the network entirely from scratch. Finally, we calculated the Pearson correlation between our predicted changes and the actual changes in loss. We repeated this process across varying group sizes, specifically for $k \in \{1, 2, 25, 50, 200\}$.

Table 7: Pearson correlation between predicted and actual validation loss changes across varying group sizes.

| Group Size ($k$) | Pearson Correlation |
|---|---|
| 1 | 0.8285 |
| 2 | 0.7707 |
| 25 | 0.7500 |
| 50 | 0.7670 |
| 200 | 0.7642 |

The results are in the Table 7. As anticipated, moving from individual node removal ($k = 1$) to group removal ($k > 1$) results in a slight initial drop in correlation. This confirms the presence of unmodeled second-order

interactions between graph nodes. However, the correlation stabilizes rapidly and remains robustly high (consistently above 0.75) even as the group size scales up to 200 nodes. This empirical evidence demonstrates that while the strict additivity assumption is an approximation, the error introduced by ignoring interaction terms is not prohibitive for practical applications. Ultimately, the aggregated individual influences serve as a highly reliable, scalable, and computationally efficient directional proxy for group influence.

### D.2 How Sensitive is the Relabelling Heuristic to Class Imbalance or Overlapping Classes?

To evaluate the effectiveness of the relabeling function on class-imbalanced and overlapping-class datasets, we measure the fraction of correctly relabeled nodes among the correctly identified points—similar to the setup discussed in Section 5.1.

**Class-imbalanced setup** For the class-imbalanced scenario, we modify the Cora dataset, which originally contains 7 classes. The first 3 classes are retained as they are, while the last 4 classes are merged into a single class. This results in a data set with the following class distribution: class 1 has 351 nodes, class 2 has 217 nodes, class 3 has 418 nodes, and class 4 (the merged class) has 1722 nodes, creating a clear class imbalance.

We assess the relabeling function's performance under both symmetric label noise and pairwise label noise. The results are as follows:

Table 8: Effectiveness of Relabelling Function Under Class Imbalance

| Noise level | 10 | 20 | 30 | 40 | 50 |
|---|---|---|---|---|---|
| SLN | 87.9±4.3 | 85.2±1.6 | 78.7±3.1 | 66.4±3.6 | 52.7±2.5 |
| Pairwise | 85.6±3.2 | 78.8±3 | 62.6±3 | 44.2±3.6 | 27.4±3.2 |

We also evaluated the effectiveness of DeGLIF on this modified variant of the Cora dataset, comparing it with GCN and RTGNN (the latter being the second-best performing algorithm after DeGLIF in our experiments on the original Cora dataset, see Section 5.1). The results are as follows:

Table 9: Comparison of Models Under Varying Noise Levels for SLN and Pairwise Settings in the Presence of Class Imbalance

| | Noise level | 10 | 20 | 30 | 40 | 50 |
|---|---|---|---|---|---|---|
| | GCN | 90.3±0.8 | 89.1±0.8 | 86.3±0.7 | 79.8±2.3 | 70.9±3.2 |
| SLN | RTGNN | 79.6±0.6 | 80.6±0.6 | 85.3±0.8 | 83.4±0.3 | 79.3±0.4 |
| | DeGLIF | 90.2±0.7 | 89.4±0.7 | 88.8±0.7 | 86.5±1 | 81.2±1.7 |
| | GCN | 89.7±0.3 | 86.7±0.9 | 78.2±2.7 | 66.0±3.8 | 47.9±4.4 |
| Pairwise | RTGNN | 78.2±0.1 | 80±0.4 | 76.1±1.8 | 73.6±1.9 | 50.1±5.9 |
| | DeGLIF | 89.7±0.3 | 89±0.3 | 85.4±0.8 | 75.2±3.3 | 55.8±6 |

Given the class imbalance in the dataset, we also computed the Macro F1-score, which exhibited a trend similar to that of accuracy. Both tables exhibit a trend similar to that observed with the original Cora dataset, indicating that class imbalance does not significantly affect the relabeling function's performance.

**Overlapping Classes** Overlapping class data refers to a situation in classification tasks where data points from different classes have very similar or identical features..To simulate an overlapping class scenario, we modify the original Cora dataset. Specifically, we take Class 3, which contains 818 nodes, and randomly split it into two halves. One half retains the original label, while the other half is assigned a new class label, effectively creating an overlapping class. The effectiveness of the relabeling function on this modified dataset is as follows:

We observe that the effectiveness of the relabeling function decreases in the presence of overlapping classes, though it remains reasonably effective. Additionally, there is a decline in the overall performance of DeGLIF, which can be attributed to two factors: (i) the reduced effectiveness of the relabeling function, and (ii)

Table 10: Effectiveness of Relabelling Function in Presence of Overlapping Classes

| Noise level | 10 | 20 | 30 | 40 | 50 |
|---|---|---|---|---|---|
| SLN | 87.9±4.3 | 85.2±1.6 | 78.7±3.1 | 66.4±3.6 | 52.7±2.5 |
| Pairwise | 85.6±3.2 | 78.8±3 | 62.6±3 | 44.2±3.6 | 27.4±3.2 |

Table 11: Comparison of Models Under Varying Noise Levels for SLN and Pairwise Settings in the Presence of Overlapping Classes

| | Noise level | 10 | 20 | 30 | 40 | 50 |
|---|---|---|---|---|---|---|
| | GCN | 71.7±0.8 | 70.6±2.1 | 69.4±1.3 | 66.2±1.2 | 61.0±2.2 |
| SLN | RTGNN | 69.1±0.7 | 68.7±1.2 | 67.5±1.7 | 67.1±0.6 | 64.9±3.2 |
| | DeGLIF | 72.0±0.8 | 71.0±1.7 | 69.8±1.1 | 68.7±1.3 | 66.1±2.2 |
| | GCN | 70.9±1.0 | 68.0±1.6 | 62.8±1.8 | 52.7±2.3 | 37.3±2.2 |
| Pairwise | RTGNN | 67.9±2.3 | 65.7±1.4 | 63.7±1.2 | 56.2±2.6 | 38.2±4.4 |
| | DeGLIF | 70.9±1.6 | 69.5±1.5 | 66.0±1.3 | 57.4±2.8 | 42.6±2.4 |

performance degradation of the backbone GNN architecture when dealing with overlapping classes—even in the absence of label noise. DeGLIF improves GCN performance and outperforms RTGNN even in the presence of overlapping classes.

Based on our empirical evaluation, we find that the relabelling function is not sensitive to class imbalance, but in the presence of overlapping classes, we find the relabelling function to be comparatively less effective.

## D.3 Comparison of DeGLIF and TSS on the Dataset Split Used by TSS

For the Cora and Citeseer datasets, both TSS and DeGLIF use similar proportions for training, validation, and test splits. TSS adopts the "full" split, where 500 nodes are sampled for validation, 1,000 for testing, and the remaining nodes are used for training. DeGLIF uses a "random" split. Specifically, a fixed number of nodes per class (e.g., 172 nodes × 7 classes for Cora) are sampled from the entire dataset for training, leaving exactly 1,500 nodes, which are then split into 500 for validation and 1,000 for testing.

DeGLIF demonstrates stable performance across both types of splits, producing comparable results. However, in our experiments, TSS shows noticeable variation depending on the splitting strategy. Since TSS reports results on the "full" split and its performance is sensitive to this choice, we also compare both methods under the full split for consistency. The results are presented in Tables 12,13. We observe that in the case of a "full" split, both these algorithms perform comparably.

Table 12: Comparison of TSS and DeGLIF(sum) Under Symmetric Noise

| Dataset | Method | 5% | 10% | 15% | 20% | 25% | 30% | 35% | 40% | 45% | 50% |
|---|---|---|---|---|---|---|---|---|---|---|---|
| Cora | TSS | 86.3±0.5 | 86±0.3 | 85.9±0.4 | 85.2±0.6 | 84.5±0.8 | 83.5±0.7 | 83.9±0.5 | 81.4±0.9 | 81.4±0.4 | 80.2±0.7 |
| | DeGLIF(sum) | 86.6±0.6 | 86.4±0.4 | 85.7±0.7 | 85.7±0.5 | 85.6±0.6 | 84.3±0.9 | 83.3±0.5 | 82.2±0.8 | 81.7±0.6 | 79.6±1.7 |
| Citeseer | TSS | 77.5±0.4 | 77.1±0.5 | 76.6±0.5 | 76.7±0.4 | 76.2±0.5 | 75.2±0.9 | 75.4±0.5 | 73.9±1.2 | 73.5±0.8 | 71.1±1 |
| | DeGLIF(sum) | 77.4±0.8 | 77±1.3 | 76.9±1 | 76.7±0.9 | 76.8±0.8 | 75.8±0.8 | 75.4±0.3 | 74.5±0.5 | 73.8±0.5 | 72.8±0.4 |

Table 13: Comparison of TSS and DeGLIF(sum) Under Pairwise Noise

| Dataset | Method | 5% | 10% | 15% | 20% | 25% | 30% | 35% | 40% | 45% | 50% |
|---|---|---|---|---|---|---|---|---|---|---|---|
| Cora | TSS | 84.6±0.4 | 85.6±0.6 | 85.1±0.7 | 83±0.8 | 83.1±0.4 | 78.8±0.7 | 74.6±1 | 68.9±1.3 | 60.85±1.3 | 47±1.2 |
| | DeGLIF(sum) | 86.7±0.2 | 86.4±0.3 | 85.1±0.6 | 84.2±1.1 | 82.8±1.2 | 78.1±0.8 | 75.6±2.4 | 71.4±2.8 | 63.9±4.5 | 53±4.5 |
| Citeseer | TSS | 77.4±0.7 | 76.8±0.5 | 77.4±0.3 | 76.7±0.6 | 75.8±0.7 | 74±1 | 71.1±0.9 | 64.7±1 | 55.4±2.4 | 40±2.7 |
| | DeGLIF(sum) | 77.5±1.2 | 77.5±0.8 | 77.1±0.4 | 76.4±1.3 | 75.9±0.6 | 73.6±1.1 | 71.7±1.7 | 68.9±1.8 | 64.8±3.9 | 56.1±6 |

### D.4 Different Model-1 and Model-2

Computation of Hessian inverse can be computationally expensive for complex architectures. In this experiment, we check for the possibility of training Model-1 on simple (fewer parameters) architecture and Model-2 with more complex architecture. For model-1, we use GCN with 1 hidden layer of dimension 8. For Model-2, we experiment on with two choices, GraphSage with 1 hidden layer of dimension 16 and ChebConv with hidden dimension 16 and k=3. We were able to complete training for this setup on a 6GB Nvidia RTX 3060 GPU. Even with different GNN architectures for both the models DeGLIF has improved accuracy under all noise conditions (see Fig. 4).

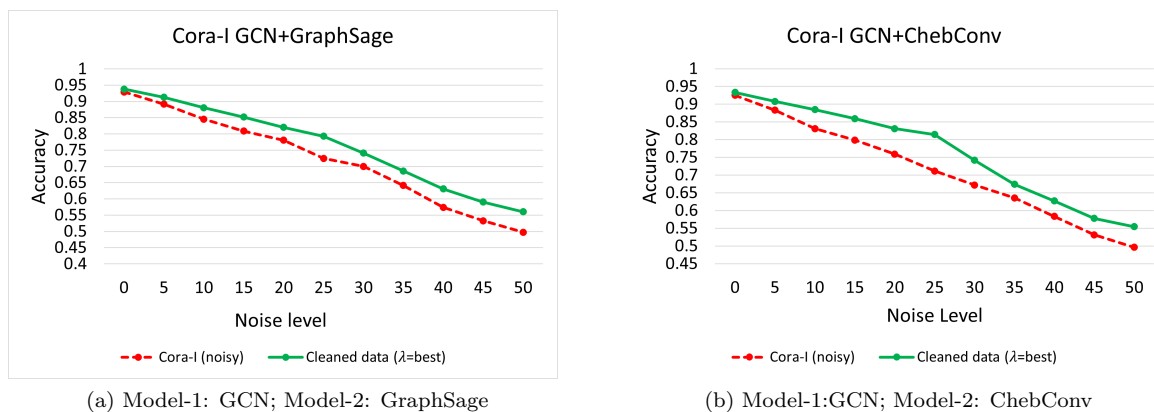

(a) Model-1: GCN; Model-2: GraphSage          (b) Model-1:GCN; Model-2: ChebConv

Figure 4: Here, the experiment is performed on the binary Cora-I with different choices for Model-1 and Model-2. In both cases, Model-1 is GCN with hidden dimension 8. In a) Model-2 is a GraphSage (Hamilton et al., 2017) with hidden dimension 16. b) Model-2 is Chebyshev Convolution (Defferrard et al., 2016)with hidden dimension 16 and k=3

### D.5 Results on Binary Labelled Datasets

In the area of label noise robust learning, binary labelled datasets are generally easier to manage. Working with such datasets can also provide meaningful insights for developing methods applicable to multiclass setups. In case of DEGLIf, relabeling noisy nodes in binary class setup, once a node is identified as corrupted, is straightforward. Building on this, we propose relabeling for the harder problem multi-class case. For binary data, we theoretically observed that flipping labels for noisy nodes is better than discarding them. Following this motivation, we designed a relabeling function with similar effectiveness for multiclass scenarios. As we couldn't find a binary labelled graph data being used by the community working on Label noise robust node classification, we converted commonly used multiclass data. A side benefit of these experiments on binary-labeled data is that they validate the effectiveness of the leave-one-out influence function for noisy node detection.

For binary setup, Cora (Yang et al., 2016), citeseer (Yang et al., 2016) and Amazon photo (Shchur et al., 2018) datasets have been converted into a noisy binary dataset. For **Cora-b**, Classes 0,1,2 were relabelled as class-0, whereas classes 3,4,5,6 were relabelled as class-1. This results in 986 nodes with label 0 and 1722 nodes with label 1. For **Citeseer-b**, classes 0,1,2 were relabelled as class-0, whereas classes 3,4,5 were relabelled as class-1. This results in 1522 nodes with label 0 and 1805 nodes with label 1. For **Amazon Photo-b**, we merge class 0,1,2,3 to form class 0 and class 4,5,6,7 to form class 1. This results in 3673 nodes with label 0 and 3977 nodes with label 1. Results for these datasets are reported in Fig. 5. We observe a similar trend to what we obtain for multiclass classification dataset

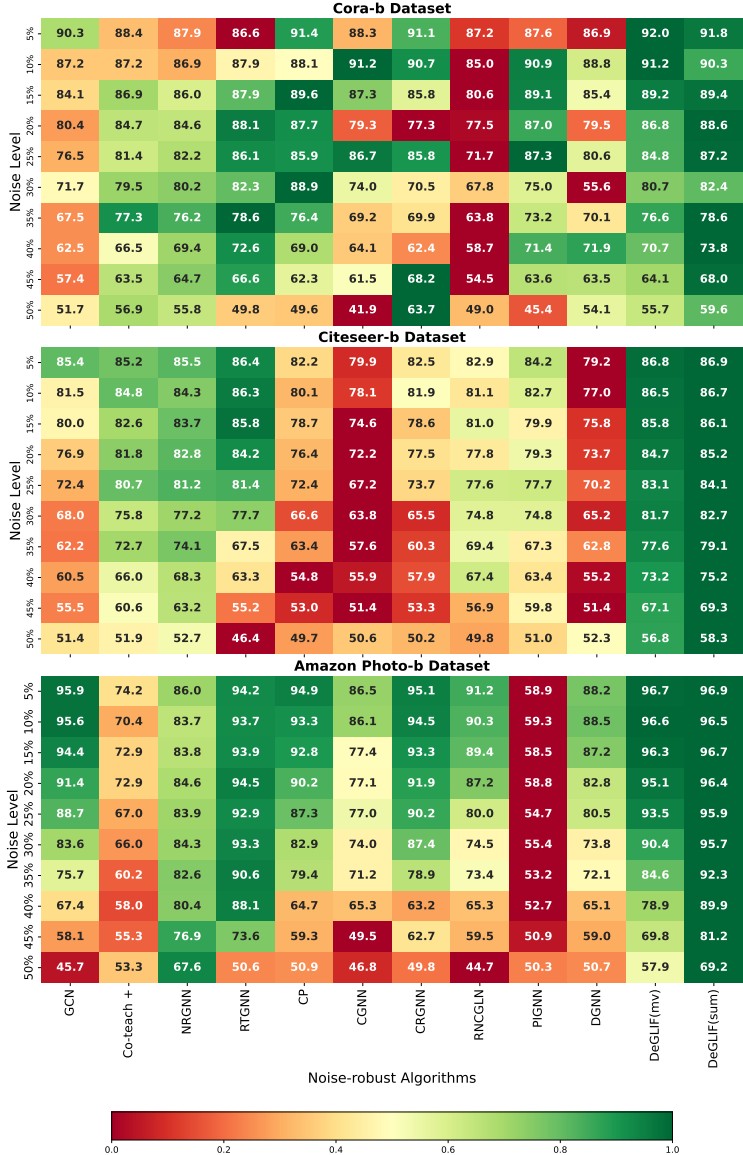

Figure 5: Comparision of DeGLIF with other baselines on binary datasets. For every row, the highest value is mapped to 1 and the lowest to 0; other values are mapped accordingly. Green means goof accuracy, red means low accuracy.

### D.6 Result on Large-Scale Dataset: OGBN-Arxiv

We tested the effectiveness of DeGLIF on a large scale dataset: ogbn-arxiv dataset (Hu et al., 2020), with default splits. None of the baselines we compared with reported results on the OGBN-Arxiv dataset, so we report DeGLIF's comparison with GCN. Accuracy obtained by the underlying GCN architecture on the clean dataset is 67.4%. 400 nodes were sampled as clean nodes for DeGLIF (0.23% of total nodes). Results are reported in Table 14.

We observe that DeGLIF also helps improve accuracy in the case of a large-scale dataset. We also recorded the macro-F1 score and observed a similar trend.

Table 14: Performance Comparison of DeGLIF on OGBN-Arxiv Under Different Noise Types

| Noise type | Model | 5 | 15 | 25 | 35 | 45 | 50 |
|---|---|---|---|---|---|---|---|
| Symmetric | GCN | 67.5±0.2 | 66.9±0.1 | 66.2±0.1 | 65.5±0.4 | 64.7±0.1 | 64.1±0.1 |
| | DeGLIF | 67.4±0.3 | 67.0±0.2 | 66.4±0.3 | 66.1±0.8 | 65.1±0.2 | 65.1±0.3 |
| Pairwise | GCN | 67.5±0.2 | 66.9±0.2 | 66.3±0.2 | 64.8±0.2 | 56.6±1.3 | 31.6±2.7 |
| | DeGLIF | 67.7±0.1 | 67.1±0.1 | 66.7±0.2 | 65.0±0.4 | 57.2±0.5 | 36.7±3.6 |

## D.7 Result on Pubmed Dataset

For the Pubmed dataset, we hold 500 nodes for validation (out of which 50 are selected to be clean), 1000 nodes for testing, and the rest of the nodes are used for training. We use the same architecture for Model-1 and Model-2 of DeGLIF as we used for other datasets in our paper. Results obtained are in Table 15 (DeGIF in result below means DeGLIF(sum)).

Table 15: Performance Comparison on Pubmed Dataset Under Different Noise Types

| Noise type | Model | 10 | 20 | 30 | 40 | 50 |
|---|---|---|---|---|---|---|
| Symmetric | GCN | 85.0±0.3 | 84.9±0.3 | 84.5±0.5 | 83.9±0.1 | 81.8±0.4 |
| | Coteach+ | 85.1±0.5 | 84.9±0.4 | 84.8±0.2 | 84.0±0.2 | 82.7±0.6 |
| | CP | 86.9±0.5 | 86.0±0.5 | 85.4±0.9 | 84.3±0.4 | 80.9±1.2 |
| | RTGNN | 78.9±0.7 | 79.9±0.3 | 80.1±0.6 | 79.2±1.3 | 74.5±0.7 |
| | NRGNN | 80.6±1.0 | 80.8±0.6 | 80.5±1.4 | 80.8±0.5 | 73.6±6.1 |
| | CGNN | 84.3±1.5 | 83.6±2.3 | 81.6±4.6 | 74.6±8.7 | 62.2±14.5 |
| | CRGNN | 87.3±0.5 | 87.2±0.6 | 85.9±0.4 | 85.0±0.9 | 45.5±13.4 |
| | DGNN | 85.0±2.4 | 81.4±3.9 | 68.4±11.3 | 76.5±2.7 | 51.5±20.1 |
| | RNCGLN | OOM | OOM | OOM | OOM | OOM |
| | PIGNN | 85.6±0.6 | 85.9±0.4 | 84.9±0.7 | 84.2±0.4 | 78.0±3.8 |
| | DeGLIF | 86.6±0.4 | 86.1±0.5 | 85.6±0.5 | 85.2±0.9 | 84.5±0.9 |
| Pairwise | GCN | 85.6±0.1 | 86.1±0.0 | 83.9±0.4 | 72.9±1.3 | 46.8±0.8 |
| | Coteach+ | 85.3±0.5 | 85.3±1.9 | 84.5±0.5 | 77.3±0.6 | 51.7±0.4 |
| | CP | 86.4±0.6 | 86.9±0.4 | 82.9±2.0 | 76.2±3.3 | 43.8±3.9 |
| | RTGNN | 79.5±0.3 | 78.9±1.0 | 72.6±1.5 | 66.2±1.8 | 51.7±1.8 |
| | NRGNN | 80.0±0.5 | 76.8±1.1 | 75.0±1.1 | 67.1±1.9 | 53.9±0.7 |
| | CGNN | 84.7±1.5 | 83.4±1.9 | 79.8±1.6 | 71.2±6.3 | 51.0±3.4 |
| | CRGNN | 87.1±0.6 | 85.9±0.4 | 83.1±0.9 | 76.6±2.1 | 44.3±6.2 |
| | DGNN | 79.5±8.3 | 78.4±8.2 | 65.2±14.6 | 60.9±9.4 | 48.3±7.5 |
| | RNCGLN | OOM | OOM | OOM | OOM | OOM |
| | PIGNN | 85.4±1.1 | 83.6±0.3 | 79.5±1.2 | 71.2±3.2 | 50.8±4.3 |
| | DeGLIF | 87.3±0.4 | 86.1±0.4 | 84.9±0.6 | 81.2±1.3 | 64.9±1.4 |

RNCGLN couldn't be trained on 24GB GPU because of more memory requirement similar issue is also reported by Wang et al. (2024). We observe that DeGLIF outperforms other algorithms on the PubMed dataset as well. DeGLIF's superior performance is more prominent in the presence of pairwise noise.

## D.8 Result on Heterophilic Data

To further effectiveness of DeGLIF on heterophilic GNNs, we employ DirGNNConv (Rossi et al., 2023) for both stages. The details are as follows: we tested the performance of DeGLIF (sum) on the Roman Empire dataset (edge homophily 0.05) (Platonov et al., 2023). As the backbone architecture, we used a 2-layer GNN, where each layer is a DirGNNConv from the PyG library.The DirGNNConv layer incorporates transformed root node features into the output. We set the hidden dimension to 16.

The data was split as follows: 50% for training, 25% for validation (from which 50 nodes were selected as clean nodes), and 25% for testing. This architecture outperformed a similarly sized GCN. Notably, the performance of GCN on clean data was lower than that of DirGNNConv on data with 50% uniform label noise. This suggests that DirGNNConv is more suitable than GCN for heterophilic graphs.

Since most of the baseline algorithms we compared DeGLIF against use GCN as their backbone, adapting each to work with DirGNNConv was not feasible. Therefore, we focused on comparing DeGLIF with DirGNNConv alone. The obtained result is in Table 16.

Table 16: Comparison of DirGNNConv and DeGLIF on Roman-Empire (Heterophilic) under Different Noise Types

| Noise type | Model | 10 | 20 | 30 | 40 | 50 |
|---|---|---|---|---|---|---|
| Symmetric | DirGNNConv | 69.1±0.5 | 68.1±0.4 | 67.5±0.3 | 65.7±0.5 | 64.8±0.4 |
| | DeGLIF | 71.2±0.6 | 70.6±0.2 | 69.8±0.6 | 68.6±0.7 | 66.7±0.2 |
| Pairwise | DirGNNConv | 69.4±0.8 | 68.6±0.8 | 66.9±0.4 | 59.9±0.6 | 39.0±1.8 |
| | DeGLIF | 71.1±0.5 | 69.8±0.5 | 67.4±0.2 | 62.2±0.5 | 40.8±2.4 |

We observe that DeGLIF is effective on heterophilic graphs and is compatible with GNN architectures specifically designed for such settings (like DirGNNConv).

### D.9 Role of Hyperparameters $\lambda$ and $\mu$

The trends for the Cora dataset with GCN at noise levels 5% to 50% on a gap of 5% are reported in Figure 6 and 7. For DeGLIF(sum) we vary $\mu$ over the set $\{0, 0.1, 1, 10, 20\}$ and take $|D_c| = 50$.

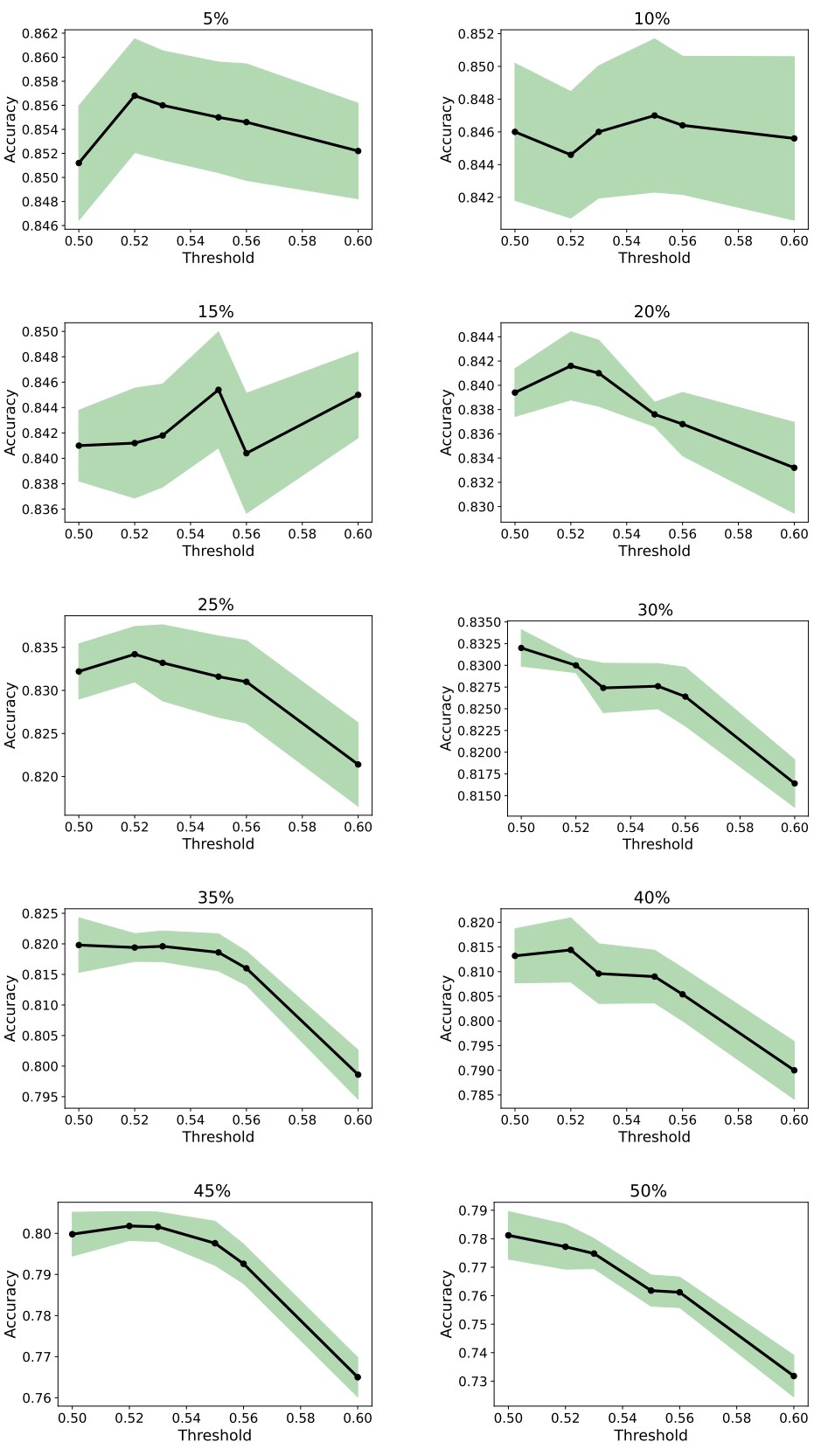

Figure 6: Relation Between $\lambda$ (Hyper Parameter for DeGLIF(mv)) and Noise Level: X-Axis Denote the Value of $\lambda$. The Black Line Denotes Accuracy, and the Shaded Region Denotes 1-Confidence Interval When the Model is Trained on Denoised Data.

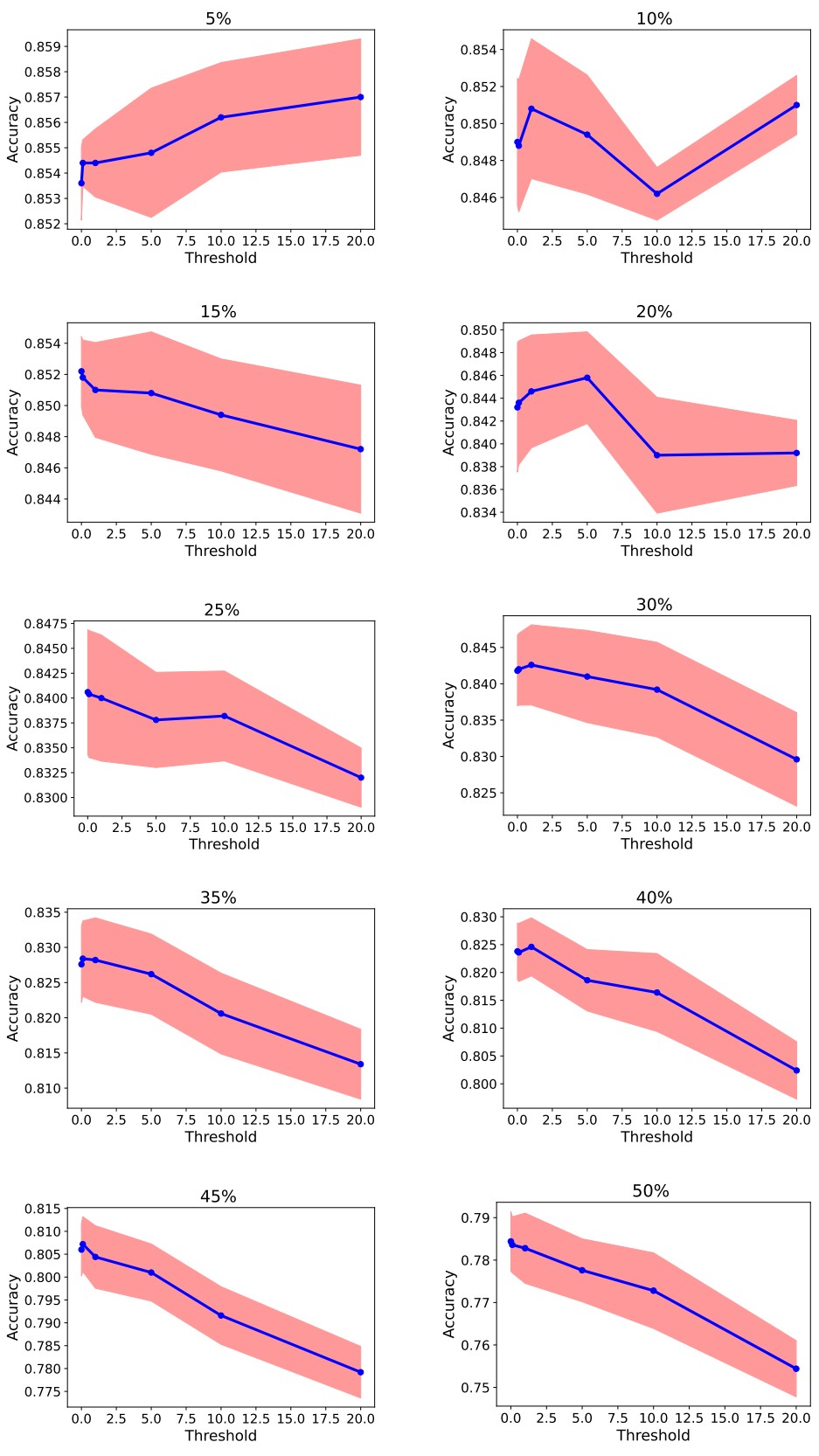

Figure 7: Relation Between $\mu$ (Hyper Parameter for DeGLIF(sum)) and Noise Level: X-axis Denote the Value of $\mu$. The Blue Line Denotes Accuracy When the Model is Trained on Denoised Data, and the Shaded Region Denotes 1-Confidence Interval.

