# OpenReview forum: "DeGLIF for Label Noise Robust Node Classification using GNNs"
_TMLR — Decision pending for TMLR_

### Review · Reviewer_4cXi · 2026-03-15

**Summary Of Contributions:**

This paper proposes DeGLIF, a denoising framework for graph node classification under label noise. The key idea is to leverage leave-one-out influence functions to estimate how removing a training node would affect the loss on a small clean validation set. Training nodes that negatively impact validation performance are identified as potentially noisy and are relabeled using a theoretically motivated relabeling rule.

The main contributions include:

1. Extending influence functions to estimate validation loss changes for graph neural networks, based on prior work that derived influence functions for graph elements.

2. A noise detection mechanism (DeGLIF(mv) and DeGLIF(sum)) that identifies potentially corrupted training nodes using influence signals computed with respect to a small clean dataset.

3. A theoretically motivated relabeling function that reassigns labels to detected noisy nodes rather than discarding them.

Overall, the paper presents a novel perspective on combining influence-function-based analysis with graph denoising for robust node classification.

**Additional Comments:**

No.

**Audience:**

Yes

**Audience Explanation:**

This paper addresses an important and widely studied problem: learning graph neural networks in the presence of label noise. Robust training under noisy labels is relevant in many graph-based applications such as social networks, recommender systems, and biomedical networks.

The work may interest several communities within the TMLR audience:

1.Graph machine learning researchers, especially those working on robustness and noisy-label learning.

2.Researchers studying influence functions and data attribution, as the paper extends influence-function analysis to graph denoising.

3.Practitioners working with weakly labeled graph data, where a small clean dataset may be available alongside large noisy datasets.

The proposed method introduces an interesting combination of influence-based analysis and graph denoising, which could inspire future work on data-centric approaches for GNN robustness.

**Broader Impact Concerns:**

No.

**Claims And Evidence:**

Yes

**Claims Explanation:**

The empirical results generally support the claims that DeGLIF improves robustness to label noise in node classification tasks. The authors evaluate the method on several commonly used datasets (Cora, Citeseer, Amazon Photo) under different noise models and noise levels. The reported results show consistent improvements over multiple baselines across a wide range of noise settings.

The paper also includes several ablation studies examining:

1.the effect of the clean dataset size,

2.the effectiveness of the relabeling function,

3.repeated applications of the denoising procedure,

4.sensitivity to hyperparameters.

These analyses help clarify the behavior of the proposed method and strengthen the empirical validation.

**Requested Changes:**

I suggest the following improvements before publication:


1.Better discussion of computational scalability
The Hessian inversion required for influence computation is expensive.
The paper would benefit from more discussion or experiments with approximate influence methods (e.g., Hessian-vector products).
2. Stronger comparison with recent robust GNN methods
While many baselines are included, it would be helpful to highlight the most recent state-of-the-art approaches and clarify how DeGLIF differs from them conceptually.

---

> ### Author Response · Authors · 2026-05-12
> **Response to Requested changes by Reviewer 4cXi**
>
> We thank the reviewer for their thoughtful evaluation and constructive feedback. We are grateful for the positive remarks regarding our work and have incorporated the suggested improvements to further strengthen the manuscript.
> ___
>
> ### **Response to 1: Better discussion on computational scalability**
>
> Before discussing approximate influence methods, it is worth noting that Hessian inversion is performed *only once for a dataset and not at every iteration.*
>
> Influence calculation (Section 5) involves calculating and inverting the Hessian matrix. The associated computational cost is $O(np^2+p^3)$ ($n$: is the number of data points; $p$: number of model parameters); for modern GNN architectures, this may be computationally challenging. One can use a cheaper approximation by directly approximating the inverse Hessian-Vector Product (iHVP),  $H^{-1}v$.
>
> Equations 4 and 5 give:
>
> $I_{up}(-z,v_i)=\\nabla_{\\theta}L(v_i,\hat{\theta})^\top \frac{1}{n} H_{\theta}^{-1} \left \[ \nabla_{\theta} L((z_i,y_i),\hat{\theta}) -\sum_{k\in V_{train}} \nabla_{\theta} L((M_k+\Delta_k,y_k),\hat{\theta}) + \sum_{k\in V_{train}} \nabla_{\theta} L((M_k,y_k),\hat{\theta}) \right \]$
>
> This can be rewritten as:
>
> $$I_{up}(-z,v_i)= (H_{\theta}^{-1}\nabla_{\theta}L(v_i,\hat{\theta})) ^\top x =s(v_i)\cdot x$$
>
> where:
> * $s(v_i)=H_{\theta}^{-1}\nabla_{\theta}L(v_i,\hat{\theta})$
> * $x=\frac{1}{n} \left[ \nabla_{\theta} L((z_i,y_i),\hat{\theta}) -\sum_{k \in V_{train}} \nabla_{\theta} L((M_k+\Delta_k,y_k),\hat{\theta}) + \sum_{k\in V_{train}} \nabla_{\theta} L((M_k,y_k),\hat{\theta}) \right]$
>
> As compiled by Koh et al. (2017), we may use the following methods to approximate $s(v_i)$:
>
> * **Conjugate Gradient Method:** Under the assumption that $H_{\theta}$ is positive definite, the iHVP $s(v_i)$ can be expressed as the solution to the quadratic optimization problem $\arg \min\_{t}\lbrace t^\top \hat{H}\_{\theta}t-v^{\top} t\rbrace$<span>. This is efficiently addressed using Conjugate Gradient (CG) methods, which avoid explicitly constructing the full $\hat{H}\_{\theta}$ matrix by requiring only the evaluation of the Hessian-vector product matrix by requiring only the evaluation of the Hessian-vector product $\hat{H}\_{\theta}t$. Such evaluations necessitate only $O(np)$ time. While an exact solution theoretically requires $p$ iterations, high-quality approximations are typically attainable with significantly fewer iterations ([Reb_1]).
>
> * **Stochastic Approach (LiSSA):** For problems with a large number of data points, CG can be slow (Koh & Liang, 2017). Using a Taylor series expansion of $H^{-1}$, we get the following relationship:
> $$
> H\_j^{-1}v= v+(1-H)H\_{j-1}^{-1}v
> $$
> where $H_j^{-1}\to H^{-1}$ as $j\to \infty$ (details in Koh & Liang, 2017).
>
> For a model with a large number of parameters, computing and storing $H$ can be expensive too. To tackle the computational challenge, Koh & Liang (2017) proposed using an unbiased estimator for $H$. In particular, they suggested using $\nabla_{\theta}^2L(z_i,\theta)$, for any $z_i$, as an unbiased estimator of $H$. Then the process can be simplified into the following steps: first, randomly select $t$ individual data points $(\{z_1,\ldots,z_t\})$ from the training set. We begin with initial estimate $\hat{H}\_0^{-1}v=v$, and the estimate is updated by $\hat{H}\_j^{-1}v= v+(1-\nabla\_{\theta}^2L(z\_j,\theta))\hat{H}\_{j-1}^{-1}v$. Empirically, stochastic method is faster than conjugate gradient method.
>
> To avoid storing of $p\times p$ Hessian estimate,  Pearlmutter trick [Reb\_2] is used; which is given by:
> $$Hv = \nabla_x (\nabla_x f(x)) \cdot v= \nabla_x (\nabla_x f(x)^\top v).$$
>
> *(Continued below.)*

---

> > ### Author Response · Authors · 2026-05-12
> > **Response to 1: Better discussion on computational scalability (Continued)**
> >
> > **Current Limitations and Graph-Specific Challenges:**
> >
> > Above was the first approach suggested for effectively approximating Hessian inverse. Since then many works have highlighted that the approximation obtained by LiSSA is not accurate. Koh et. al 2024 mentions that reason for inaccurate approximation is due to random sampling, which suffers due to high variance. Basu et.al. 2020 [Reb\_3]   mention that inverse hessian vector product are erroneous especially when the network is deep.  Feldman \& Zhang (2020) [Reb\_4],  confirmed that estimation errors can occur even in simple single layer networks. [Reb\_5] shows that the iterative process may fail to converge in some realistic scenarios.
> >
> > What we would like to highlight here is that even though there has been some work on approximating the influence function via iHVP, it's an ongoing research even for i.i.d. data.
> >
> > Recall that $I_{up}(-z,v_i)=s(v_i)\cdot x$; When it comes to graph data and GNN, computing $x$ is much more expensive than in i.i.d, setup. (For i.i.d., the last two terms in the summation are not present). Hence, an appropriate approximation for $x$ is also required. Additionally, recent more accuracte approximations like Koh et. al. 2024 uses tools that do not translate directly to GNNs or graph data. Koh. et. al. 2024 suggested that rather than using random samples, one should organize data within a latent feature space and then selecting data points based on space topology. To do so, they extract features using a Pretrained ViT, a suitable model is required for Graphs. For the graph, it's ongoing work.
> >
> > **References:**
> >
> > * [Reb_1] Martens, J. Deep learning via Hessian-free optimization. ICML, 2010.
> > * [Reb_2] Pearlmutter, B. A. Fast exact multiplication by the Hessian. Neural Computation, 1994.
> > * [Reb_3] Basu, S., You, X., and Feizi, S. On second-order group influence functions for black-box predictions. ICML, 2020.
> > * [Reb_4] Feldman, V. and Zhang, C. What neural networks memorize and why. NeurIPS, 2020.
> > * [Reb_5] Lyu, Hyeonsu, et al. "Deeper understanding of black-box predictions via generalized influence functions." arXiv, 2023.

---

> ### Author Response · Authors · 2026-05-12
> **Response to 2: Stronger comparison with recent robust GNN methods**
>
> ### **Response to 2: Stronger comparison with recent robust GNN methods**
>
> The current state-of-the-art noise-robust algorithms for graph data include TSS, GD$^2$, and ERASE. Our manuscript already includes a comprehensive comparison between DeGLIF and TSS. Regarding GD$^2$, the official code is not currently publicly available. We had previously contacted the authors via email to request access to their implementation for a related project, but we have not yet been able to obtain it. To further strengthen our empirical evaluation, we have added a new comparison of DeGLIF against ERASE. We report the results below:
>
> **Comparison follows for Cora:**
>
> | Noise Model | Algorithm | 5% | 10% | 15% | 20% | 25% | 30% | 35% | 40% | 45% | 50% |
> | :--- | :--- | :--- | :--- | :--- | :--- | :--- | :--- | :--- | :--- | :--- | :--- |
> | SLN | ERASE | 80.7±1 | 80.6±1 | 80.7±0.9 | 81.8±1.2 | 80.1±1.6 | 81±1.2 | 79.7±1.1 | 81.2±1.1 | **81.5±1.2** | **80.9±1.4** |
> |  | DeGLIF(mv) | 85.1±0.3 | **84.7±0.5** | 84.0±0.2 | 83.2±1.1 | 82.7±0.8 | 82.1±1.6 | 82.0±0.7 | 80.9±1.9 | 79.0±2.3 | 76.8±1.8 |
> |  | DeGLIF(sum) | **86.1±0.5** | **84.7±0.8** | **85.8±0.3** | **84.3±1.3** | **84.6±1.1** | **84.3±0.7** | **84.8±0.6** | **84.0±0.6** | 81.1±1.4 | 80.5±1.8 |
> | PW| ERASE | 80.3±0.9 | 79.9±0.3 | 78.9±1.3 | 78.7±0.9 | 78.8±1.2 | 77.7±1 | **76.8±1.9** | **71.6±1.5** | 55.9±5 | 37.1±0.6 |
> |  | DeGLIF(mv) | 86.3±0.8 | **86.0±0.7** | **84.8±0.3** | **83.2±0.1** | **81.1±1.6** | 76.7±2 | 72.1±2 | 64.8±3.8 | 57.9±4.1 | 48.6±3.8 |
> |  | DeGLIF(sum) | **86.5±0.4** | 85.9±0.4 | 84.1±0.5 | 82.6±0.5 | 80.8±0.5 | **78.5±0.8** | 73.9±3.5 | 66.8±2.5 | **63.0±4.5** | **51.8±4.6** |
>
> **For Citeseer data:**
> | Noise Model | Algorithm | 5% | 10% | 15% | 20% | 25% | 30% | 35% | 40% | 45% | 50% |
> | :--- | :--- | :--- | :--- | :--- | :--- | :--- | :--- | :--- | :--- | :--- | :--- |
> | SLN | ERASE | 80.2±0.5 | 80.3±0.3 | 79.6±0.7 | 79±0.6 | 79.1±0.7 | 78.3±1.3 | 76±2 | **76.2±2.3** | 76.2±0.4 | **75.9±0.8** |
> |  | DeGLIF(mv) | 77.8±1.2 | 77.2±1.2 | 76.9±1.3 | 76.5±0.5 | 76.5±0.7 | 76.1±0.4 | 75.4±0.8 | 74.1±0.6 | 74.2±0.5 | 72.3±0.7 |
> | | DeGLIF(sum) | **81.5±0.8** | **81.2±0.6** | **80.7±0.8** | **79.5±0.8** | **79.7±0.7** | **78.8±0.7** | **77.4±1.9** | 75.6±1.7 | **76.3±1.9** | 74.3±1.6 |
> | PW | ERASE | 79.2±2.4 | **79.9±0.4** | **78.9±0.4** | 77.4±0.4 | 77.1±0.5 | **76.1±0.8** | **73.5±3.4** | **72.5±2.9** | **64.2±1.3** | **61.2±6.8** |
> | | DeGLIF(mv) | 79.8±0.4 | 79.1±1.1 | 78.4±0.4 | 77.5±1 | 76.7±0.9 | 75.0±0.9 | 71.6±1.4 | 66.6±2.5 | 51.5±3.4 | 49.0±4.1 |
> |  | DeGLIF(sum) | **80.2±0.9** | 78.8±0.9 | 78.2±1.1 | **78.0±0.8** | **77.8±0.7** | 75.6±1.2 | 73.4±1.2 | 69.6±2.8 | 63.7±3.7 | 54.0±1.8 |
>
> **For PubMed data:**
> | Noise Model | Algorithm | 10% | 20% | 30% | 40% | 50% |
> | :--- | :--- | :--- | :--- | :--- | :--- | :--- |
> | SLN | ERASE | 84.3±0.4 | 84±0.4 | 83.5±0.4 | 82.9±0.3 | 81.7±1 |
> | | DeGLIF(sum) | **86.6±0.4** | **86.1±0.5** | **85.6±0.5** | **85.2±0.9** | **84.5±0.9** |
> | PW | ERASE | 84.3±0.3 | 84.4±0.3 | 83.7±0.3 | 79.6±1 | 44.4±1.4 |
> | | DeGLIF(sum) | **87.3±0.4** | **86.1±0.4** | **84.9±0.6** | **81.2±1.3** | **64.9±1.4** |
>
> **For Amazon Photo data:**
>
> For the Cora, Citeseer, and PubMed datasets, we use the same hyperparameter settings as the authors. Authors in their implementation have used one set of hyperparameters for one dataset, using a similar approach for Amazon Photo, we tune hyperparameters in the presence of 10\% pairwise noise. Hypermemeters are tuned over the following set:
> gam2 $\in \lbrace1,2,3\rbrace$; eps $\in \lbrace0.01,0.05,0.1\rbrace$; $\alpha \in \lbrace0.1,0.5,0.7\rbrace$; $\beta \in \lbrace0.6,0.7\rbrace$;
> T $\in \lbrace2,4,6\rbrace$. We obtain the following set of hyperparameters: gam2=2, eps=0.05, $\alpha=0.7$, $\beta=0.7$, T=2. We obtain the following results for the Amazon photo dataset:
>
> | Noise Model | Algorithm | 5% | 10% | 15% | 20% | 25% | 30% | 35% | 40% | 45% | 50% |
> | :--- | :--- | :--- | :--- | :--- | :--- | :--- | :--- | :--- | :--- | :--- | :--- |
> | SLN | ERASE | 89.5±0.6 | 88.4±0.6 | 88.9±0.5 | 89.5±0.8 | 89.8±1.3 | 89.0±1.3 | 88.0±0.2 | **88.1±1.6** | **88.9±0.8** | **86.8±0.9** |
> |  | DeGLIF(mv) | 89.6±0.9 | 89.6±0.3 | 89.9±0.7 | 89.7±1.1 | 88.9±1.4 | 89.1±2.2 | **88.6±0.9** | 86.7±1.6 | 83.5±1.2 | 82.0±4.5 |
> |  | DeGLIF(sum) | **91.8±0.6** | **91.6±0.2** | **90.8±0.4** | **90.3±1** | **90.0±0.7** | **89.8±2** | 88.1±1.3 | 87.1±1.6 | 86.4±1.7 | 82.6±4.2 |
> | PW | ERASE | **90±1** | **89.2±0.4** | 88.5±0.7 | 86.7±0.9 | **86.2±1.3** | **85.8±0.9** | **83.9±1.3** | **80.8±0.8** | **73.4±2.7** | 55.8±1.6 |
> | | DeGLIF(mv) | 87.5±0.5 | 87.8±0.5 | **89.8±3.7** | **88.6±1.2** | 82.8±2.4 | 80.3±2.7 | 73.1±5.9 | 64.0±9.5 | 66.1±7.3 | 54.2±8.1 |
> |  | DeGLIF(sum) | 89.6±1 | **89.2±1.3** | 89.2±0.2 | 87.4±0.6 | 85.3±2.6 | 83.2±2.1 | 77.0±6.1 | 74.5±7.6 | 71.8±5.4 | **60.6±10** |
>
> *(Discussion about these computations is continued below.)*

---

> ### Author Response · Authors · 2026-05-12
> **Response to 2: Stronger comparison with recent robust GNN methods (Continued)**
>
> DeGLIF consistently outperforms ERASE on the Cora, Citeseer, and PubMed datasets, particularly at lower noise levels (excluding a few specific instances under the pairwise noise model), while ERASE demonstrates a slight performance edge on the Amazon Photo dataset. We detail the conceptual differences between existing algorithms and DeGLIF.
>
> Conceptually, the majority of recent robust GNN methods, such as RTGNN, CGNN, CP, and NRGNN, etc., can be broadly grouped into two categories based on their mechanisms. The first category includes methods like NRGNN and RTGNN, which attempt to mitigate noise by adding edges between unlabelled and labeled nodes based on feature similarity. While effective on sparse networks, this edge-adding strategy offers little benefit on already dense graphs like Amazon Photos (in contrast, it degrades performance). The second category, which includes methods like CP and CGNN, relies heavily on label smoothness and strong community structures. Consequently, these algorithms tend to underperform on sparse citation networks like Cora and Citeseer, where community boundaries are weak or overlapping.
>
>
> *Because DeGLIF does not rely on these topological modifications or smoothness assumptions, it maintains robust performance across both sparse and dense graph types.* Instead, DeGLIF uses a leave-one-out influence function to estimate the impact of removing a training node on a small clean validation set.
>
> Compared with approaches like TSS, DeGLIF fundamentally differs in how it handles corrupted data. TSS relies on topological sample selection, progressively selecting clean nodes for training while discarding noisy ones. DeGLIF, conversely, actively identifies noisy data and uses a theoretically motivated relabeling function to correct it. This capability allows DeGLIF to work exceptionally well even when the training set is small (such as in the Amazon Photo dataset with 5\% nodes as the training set), a scenario where discarding data via sample selection severely degrades performance.
>
> Furthermore, as highlighted in Section 5.4 of our manuscript, DeGLIF is not solely designed to compete with these algorithms, but rather to assist them. Because DeGLIF acts as a denoising framework, the resulting cleaned dataset can be fed into any other end-to-end noise-robust algorithm. Our experiments ( in Section 5.4) demonstrate that a hybrid model (combining DeGLIF's denoising with TSS) performs better than either method alone, particularly under high noise conditions.
>
> All changes will be incorporated in the camera-ready submission.

---

### Review · Reviewer_6D3i · 2026-03-16

**Summary Of Contributions:**

The authors propose a new method for noise robust label prediction in node classification tasks called DeGLIF. By using a clean subset of the provided data, noisy underlying data is relabeled using a leave-one-out influence function to provide improved node labels.  Empirical results indicate improved performance on high noise levels over baseline methods, while retaining similar performance on lower noise levels. However, most improvements over existing noisy label methods are marginal. Furthermore, ablation studies concerning hyperparameters of DeGLIF and sample ratio of clean node labels are provided.

**Additional Comments:**

Questions:

1. How does the method transfer to the inductive node classification setting, by having multiple graphs where node classification is supposed to be done?

2. It is not completely clear to me on why Theorem 1 and 2 can be directly translated to test risk as outlined in the last line of section 3.

3. Is there a specific reason for the GCN method to only use a single hidden layer? Usual GNN approaches use multiple hidden layers with this setting not considered in the paper.

**Audience:**

Yes

**Audience Explanation:**

There is definitely interest by audience working on node classification problems, especially with real world data on how to improve label noise with a subset of clean labeled data available. Furthermore, the proposed framework extends previous works in a similar fashion while retaining the idea of relabeling on node classification tasks.
With this work being the first to apply the leave-one-out influence function, already known from general machine learning methods, this work combines classic approaches with the domain specific requirements of node classification tasks.
Furthermore, previous works indicate improved performance on noisy data by relabeling methods. However, the audience is probably a small subset of the TMLR audience due to limited applications of noisy node classification models.

**Broader Impact Concerns:**

No specific impact concerns for this work.

**Claims And Evidence:**

Yes

**Claims Explanation:**

The proposed claims by the authors are supported by empirical evidence, specifically for higher noise levels. Regarding results for lower noise levels such as 5%-20% the proposed method does not perform significantly better than other methods for relabeling or baseline methods. Furthermore, most improvements are rather marginal over either baseline methods or other noisy label methods.
Moreover, a detailed comparison to other baseline methods and SOTA methods on noisy node labels is missing for current SOTA methods. Nonetheless, previous methods are outlined in the empirical section with detailed performance results highlighted.
Especially since DeGLIF has a rather high computational overhead (although only once computed) a detailed comparison to other existing baselines would provide additional insights on the application of DeGLIF.

 However, evaluation remains limited to a few datasets, while no major novel theoretical insights are given in the paper except for application of existing concepts from leave-one-out influence functions. Therefore, the contribution is rather limited with the proposed method.
Overall, the contribution is sound in the sense that the application of the leave-one-out influence function improves performance on noisy labels compared to baseline results, however empirical evaluation is lacking results that indicate significant advantages over previous methods.

**Requested Changes:**

Critical: None

Major:
1. Evaluation so far is only done on small scale datasets except ogbn-arxiv in the appendix. It would be beneficial to the empirical part of the work to provide additional results on node classification datasets. Also dedicated benchmarks such as [1] could be useful to evaluate performance of the proposed approach beyond the datasets mentioned in the paper.

 2. Missing comparison to current SOTA methods on label noise methods for GNNs such as GD^2 [2]. Since DeGLIF is supposed to complement existing methods it would be useful to either compare DeGLIF to the SOTA methods or evaluate a combination of both methods.

Minor:
1. Baseline comparison is only given by a GCN architecture, how does the proposed method compare to other baseline methods? An inclusion of additional baselines would be great.

2. Figure 1 could be remade to align better with the remaining paper design, however this is a choice for the authors to make.

3. There Is a missing reference in section 4.1.1.

4. Writing is sometimes inconsistent in terms of grammar used (e.g in Section 2.1 and 2.1.1).

5. Informal writing (e.g in Section 2, first sentence, Section 2.1) occurs in the paper.

[1] NoisyGL: A Comprehensive Benchmark for Graph Neural Networks under Label Noise, Wang et al., 2024

[2] GD^2: Robust Graph Learning under Label Noise via Dual-View Prediction Discrepancy, Li et al., 2025

---

> ### Author Response · Authors · 2026-05-14
> **Response to Reviewer 6D3i**
>
> We sincerely thank all the reviewer for their thorough evaluation and valuable suggestions. Please find our detailed responses below. We first address the requested changes, followed by our responses to the additional comments.
>
> ---
>
> ## **Response to Requested changes: Major**
>
> ### **Response to 1. (regarding testing DeGLIF on additional datasets)**
>
> We thank the reviewer for this constructive suggestion. We would like to mention that compared to node classification task, evaluating models under label noise is computationally demanding. This is because, the experiments must be repeated across multiple noise models (SLN and PW) and noise levels (typically 5 to 10 variations). Consequently, standard noise-robust algorithms for graphs (such as NRGNN and RTGNN,) generally evaluate on Cora, CiteSeer, and PubMed datasets.
>
> To provide a more robust evaluation than the existing literature, our original manuscript expanded upon this standard set by including the Amazon Photo dataset to introduce diversity in average degree and training size. Furthermore, as the reviewer kindly noted, we also demonstrated the scalability of DeGLIF on the large-scale ogbn-arxiv dataset.
>
> However, we agree that further empirical validation strengthens the paper. Following your suggestion, we have now included new experiments on the Amazon Computers dataset, which contains approximately twice as many nodes as Amazon Photo. Using a 20 nodes per class training split, we compared DeGLIF against the baselines using the benchmark code provided in [1] (we only compare against baselines that perform at least as well as GCN on this dataset, based on the results in [1]). These results are as follows:
>
> | Noise Type | Method | 0% | 10% | 20% | 30% | 40% | 50% |
> | :--- | :--- | :--- | :--- | :--- | :--- | :--- | :--- |
> | SLN | GCN | 82.28 ± 1.04 | 80.80 ± 2.31 | 79.52 ± 1.89 | 75.72 ± 1.96 | 72.82 ± 5.25 | 65.82 ± 7.88 |
> | | PiGNN | 80.78 ± 3.04 | **81.90 ± 1.15** | 79.44 ± 1.87 | 78.94 ± 2.54 | 73.64 ± 2.43 | **69.86 ± 4.50** |
> | | CP | 81.76 ± 0.71 | 80.32 ± 2.47 | 77.52 ± 0.89 | 76.20 ± 2.20 | 72.32 ± 3.09 | 66.04 ± 7.99 |
> | | DeGLIF(sum) | **83.56 ± 0.33** | 81.82 ± 0.77 | **80.75 ± 0.94** | **79.16 ± 1.49** | **75.76 ± 3.06** | 68.86 ± 1.63 |
> | PW | GCN | 82.28 ± 1.04 | 80.58 ± 0.99 | 76.78 ± 3.45 | 64.68 ± 5.78 | 57.94 ± 10.58 | 36.70 ± 9.56 |
> | | PiGNN | 80.78 ± 3.04 | 80.46 ± 1.90 | 78.12 ± 3.23 | 67.36 ± 6.03 | 60.40 ± 10.76 | 39.88 ± 16.63 |
> | | CP | 81.76 ± 0.71 | 79.38 ± 1.54 | 75.00 ± 4.60 | 67.24 ± 5.45 | 55.34 ± 6.14 | 37.44 ± 12.98 |
> | | DeGLIF(sum) | **83.56 ± 0.33** | **80.86 ± 0.97** | **78.23 ± 1.76** | **69.66 ± 3.07** | **62.34 ± 4.49** | **41.09 ± 8.12** |
>
> We observe that on the Amazon Computer dataset DeGLIF performs comparable to CP and PIGNN (the phenomenon is similar to one seen on a similar dataset, Amazon Photo). It is worth reiterating that all other algorithms (such as NRGNN, RTGNN, etc) perform worse on this dataset, compared to methods included here.
>
> ---
>
> ### **Response to 2. (regarding comparison with recent SOTA)**
>
>
> The current state-of-the-art noise-robust algorithms for graph data include TSS, GD$^2$, and ERASE. Our manuscript already includes a comprehensive comparison between DeGLIF and TSS. Regarding GD$^2$, the official code is not currently publicly available. We had previously contacted the authors via email to request access to their implementation for a related project, but we have not yet been able to obtain it. To further strengthen our empirical evaluation, we have added a new comparison of DeGLIF against ERASE. We report the results below:
>
> **Comparison follows for Cora:**
>
> | Noise Model | Algorithm | 5% | 10% | 15% | 20% | 25% | 30% | 35% | 40% | 45% | 50% |
> | :--- | :--- | :--- | :--- | :--- | :--- | :--- | :--- | :--- | :--- | :--- | :--- |
> | SLN | ERASE | 80.7±1 | 80.6±1 | 80.7±0.9 | 81.8±1.2 | 80.1±1.6 | 81±1.2 | 79.7±1.1 | 81.2±1.1 | **81.5±1.2** | **80.9±1.4** |
> |  | DeGLIF(mv) | 85.1±0.3 | **84.7±0.5** | 84.0±0.2 | 83.2±1.1 | 82.7±0.8 | 82.1±1.6 | 82.0±0.7 | 80.9±1.9 | 79.0±2.3 | 76.8±1.8 |
> |  | DeGLIF(sum) | **86.1±0.5** | **84.7±0.8** | **85.8±0.3** | **84.3±1.3** | **84.6±1.1** | **84.3±0.7** | **84.8±0.6** | **84.0±0.6** | 81.1±1.4 | 80.5±1.8 |
> | PW| ERASE | 80.3±0.9 | 79.9±0.3 | 78.9±1.3 | 78.7±0.9 | 78.8±1.2 | 77.7±1 | **76.8±1.9** | **71.6±1.5** | 55.9±5 | 37.1±0.6 |
> |  | DeGLIF(mv) | 86.3±0.8 | **86.0±0.7** | **84.8±0.3** | **83.2±0.1** | **81.1±1.6** | 76.7±2 | 72.1±2 | 64.8±3.8 | 57.9±4.1 | 48.6±3.8 |
> |  | DeGLIF(sum) | **86.5±0.4** | 85.9±0.4 | 84.1±0.5 | 82.6±0.5 | 80.8±0.5 | **78.5±0.8** | 73.9±3.5 | 66.8±2.5 | **63.0±4.5** | **51.8±4.6** |
>
> *Response to this comment is continued in the next block.*

---

> > ### Author Response · Authors · 2026-05-14
> > **Response to 2. (regarding comparison with recent SOTA) (continued)**
> >
> > ### **Response to 2. (regarding comparison with recent SOTA) (continued)**
> >
> > **For Citeseer data:**
> > | Noise Model | Algorithm | 5% | 10% | 15% | 20% | 25% | 30% | 35% | 40% | 45% | 50% |
> > | :--- | :--- | :--- | :--- | :--- | :--- | :--- | :--- | :--- | :--- | :--- | :--- |
> > | SLN | ERASE | 80.2±0.5 | 80.3±0.3 | 79.6±0.7 | 79±0.6 | 79.1±0.7 | 78.3±1.3 | 76±2 | **76.2±2.3** | 76.2±0.4 | **75.9±0.8** |
> > |  | DeGLIF(mv) | 77.8±1.2 | 77.2±1.2 | 76.9±1.3 | 76.5±0.5 | 76.5±0.7 | 76.1±0.4 | 75.4±0.8 | 74.1±0.6 | 74.2±0.5 | 72.3±0.7 |
> > | | DeGLIF(sum) | **81.5±0.8** | **81.2±0.6** | **80.7±0.8** | **79.5±0.8** | **79.7±0.7** | **78.8±0.7** | **77.4±1.9** | 75.6±1.7 | **76.3±1.9** | 74.3±1.6 |
> > | PW | ERASE | 79.2±2.4 | **79.9±0.4** | **78.9±0.4** | 77.4±0.4 | 77.1±0.5 | **76.1±0.8** | **73.5±3.4** | **72.5±2.9** | **64.2±1.3** | **61.2±6.8** |
> > | | DeGLIF(mv) | 79.8±0.4 | 79.1±1.1 | 78.4±0.4 | 77.5±1 | 76.7±0.9 | 75.0±0.9 | 71.6±1.4 | 66.6±2.5 | 51.5±3.4 | 49.0±4.1 |
> > |  | DeGLIF(sum) | **80.2±0.9** | 78.8±0.9 | 78.2±1.1 | **78.0±0.8** | **77.8±0.7** | 75.6±1.2 | 73.4±1.2 | 69.6±2.8 | 63.7±3.7 | 54.0±1.8 |
> >
> > **For PubMed data:**
> > | Noise Model | Algorithm | 10% | 20% | 30% | 40% | 50% |
> > | :--- | :--- | :--- | :--- | :--- | :--- | :--- |
> > | SLN | ERASE | 84.3±0.4 | 84±0.4 | 83.5±0.4 | 82.9±0.3 | 81.7±1 |
> > | | DeGLIF(sum) | **86.6±0.4** | **86.1±0.5** | **85.6±0.5** | **85.2±0.9** | **84.5±0.9** |
> > | PW | ERASE | 84.3±0.3 | 84.4±0.3 | 83.7±0.3 | 79.6±1 | 44.4±1.4 |
> > | | DeGLIF(sum) | **87.3±0.4** | **86.1±0.4** | **84.9±0.6** | **81.2±1.3** | **64.9±1.4** |
> >
> > **For Amazon Photo data:**
> >
> > For the Cora, Citeseer, and PubMed datasets, we use the same hyperparameter settings as the authors. Authors in their implementation have used one set of hyperparameters for one dataset, using a similar approach for Amazon Photo, we tune hyperparameters in the presence of 10\% pairwise noise. Hypermemeters are tuned over the following set:
> > gam2 $\in \lbrace1,2,3\rbrace$; eps $\in \lbrace0.01,0.05,0.1\rbrace$; $\alpha \in \lbrace0.1,0.5,0.7\rbrace$; $\beta \in \lbrace0.6,0.7\rbrace$;
> > T $\in \lbrace2,4,6\rbrace$. We obtain the following set of hyperparameters: gam2=2, eps=0.05, $\alpha=0.7$, $\beta=0.7$, T=2. We obtain the following results for the Amazon photo dataset:
> >
> > | Noise Model | Algorithm | 5% | 10% | 15% | 20% | 25% | 30% | 35% | 40% | 45% | 50% |
> > | :--- | :--- | :--- | :--- | :--- | :--- | :--- | :--- | :--- | :--- | :--- | :--- |
> > | SLN | ERASE | 89.5±0.6 | 88.4±0.6 | 88.9±0.5 | 89.5±0.8 | 89.8±1.3 | 89.0±1.3 | 88.0±0.2 | **88.1±1.6** | **88.9±0.8** | **86.8±0.9** |
> > |  | DeGLIF(mv) | 89.6±0.9 | 89.6±0.3 | 89.9±0.7 | 89.7±1.1 | 88.9±1.4 | 89.1±2.2 | **88.6±0.9** | 86.7±1.6 | 83.5±1.2 | 82.0±4.5 |
> > |  | DeGLIF(sum) | **91.8±0.6** | **91.6±0.2** | **90.8±0.4** | **90.3±1** | **90.0±0.7** | **89.8±2** | 88.1±1.3 | 87.1±1.6 | 86.4±1.7 | 82.6±4.2 |
> > | PW | ERASE | **90±1** | **89.2±0.4** | 88.5±0.7 | 86.7±0.9 | **86.2±1.3** | **85.8±0.9** | **83.9±1.3** | **80.8±0.8** | **73.4±2.7** | 55.8±1.6 |
> > | | DeGLIF(mv) | 87.5±0.5 | 87.8±0.5 | **89.8±3.7** | **88.6±1.2** | 82.8±2.4 | 80.3±2.7 | 73.1±5.9 | 64.0±9.5 | 66.1±7.3 | 54.2±8.1 |
> > |  | DeGLIF(sum) | 89.6±1 | **89.2±1.3** | 89.2±0.2 | 87.4±0.6 | 85.3±2.6 | 83.2±2.1 | 77.0±6.1 | 74.5±7.6 | 71.8±5.4 | **60.6±10** |
> >
> > *(Discussion about these computations is continued in next block)*

---

> > > ### Author Response · Authors · 2026-05-14
> > > **Response to 2. (regarding comparison with recent SOTA) (continued)**
> > >
> > > ### **Response to 2. (regarding comparison with recent SOTA) (continued)**
> > >
> > > DeGLIF consistently outperforms ERASE on the Cora, Citeseer, and PubMed datasets, particularly at lower noise levels (excluding a few specific instances under the pairwise noise model), while ERASE demonstrates a slight performance edge on the Amazon Photo dataset. We detail the conceptual differences between existing algorithms and DeGLIF.
> > >
> > > Conceptually, the majority of recent robust GNN methods, such as RTGNN, CGNN, CP, and NRGNN, etc., can be broadly grouped into two categories based on their mechanisms. The first category includes methods like NRGNN and RTGNN, which attempt to mitigate noise by adding edges between unlabelled and labeled nodes based on feature similarity. While effective on sparse networks, this edge-adding strategy offers little benefit on already dense graphs like Amazon Photos (in contrast, it degrades performance). The second category, which includes methods like CP and CGNN, relies heavily on label smoothness and strong community structures. Consequently, these algorithms tend to underperform on sparse citation networks like Cora and Citeseer, where community boundaries are weak or overlapping.
> > >
> > >
> > > *Because DeGLIF does not rely on these topological modifications or smoothness assumptions, it maintains robust performance across both sparse and dense graph types.* Instead, DeGLIF uses a leave-one-out influence function to estimate the impact of removing a training node on a small clean validation set.
> > >
> > > Compared with approaches like TSS, DeGLIF fundamentally differs in how it handles corrupted data. TSS relies on topological sample selection, progressively selecting clean nodes for training while discarding noisy ones. DeGLIF, conversely, actively identifies noisy data and uses a theoretically motivated relabeling function to correct it. This capability allows DeGLIF to work exceptionally well even when the training set is small (such as in the Amazon Photo dataset with 5\% nodes as the training set), a scenario where discarding data via sample selection severely degrades performance.
> > >
> > > Furthermore, as highlighted in Section 5.4 of our manuscript, DeGLIF is not solely designed to compete with these algorithms, but rather to assist them. Because DeGLIF acts as a denoising framework, the resulting cleaned dataset can be fed into any other end-to-end noise-robust algorithm. Our experiments ( in Section 5.4) demonstrate that a hybrid model (combining DeGLIF's denoising with TSS) performs better than either method alone, particularly under high noise conditions.
> > >
> > > These  changes will be incorporated in camera ready submission.
> > >
> > > ---
> > >
> > > ## **Response to requested changes: Minor**
> > >
> > > ### **Response to 1. (comparison with other baselines)**
> > >
> > > We thank the reviewer for this insightful suggestion. In the research area of label noise robust graph learning, most studies stick to a standard GCN architecture as the baseline. The reason for this is to clearly isolate the source of the improvement. If a paper introduces a new denoising technique and simultaneously pairs it with a more advanced architecture like GAT or GraphSAGE, it becomes difficult to tell why the accuracy improved. The reviewer might wonder if the gain was due to the new denoising algorithm itself, or simply because a more powerful GNN model was used. By standardizing on a simple GCN backbone across all methods, the community ensures that any boost in performance is strictly and provably due to the noise-handling mechanism.
> > >
> > > That being said, we completely agree that demonstrating the effectiveness of DeGLIF alongside other standard architectural baselines significantly strengthens our claims of versatility. Following your suggestion, we have included evaluations comparing the standard baselines like GAT, GraphTransformer, and GraphSage. GraphSage uses a similar architectural configuration to that used by GCN, GAT anf Graphtransformer uses a single hiddne layer with 8 heads of size 8, GAT also uses a dropout with $p=0.6$. *The results are continued in the next block.*

---

> > > > ### Author Response · Authors · 2026-05-14
> > > > **Response to 1. (comparison with other baselines) (Continued)**
> > > >
> > > > ### **Response to 1. (comparison with other baselines) (Continued)**
> > > >
> > > > Results are as follows:
> > > >
> > > > For Cora dataset :
> > > >
> > > > | Noise Model | Algorithm | 0% | 5% | 10% | 15% | 20% | 25% | 30% | 35% | 40% | 45% | 50% |
> > > > | :--- | :--- | :--- | :--- | :--- | :--- | :--- | :--- | :--- | :--- | :--- | :--- | :--- |
> > > > | SLN | GCN | **86.4±0.2** | 85.5±0.6 | 84.5±0.5 | 83.1±1.7 | 81.5±0.9 | 79.2±1.2 | 76.5±1.5 | 72.8±1.9 | 69.2±1.8 | 65.1±1.9 | 59.1±2.8 |
> > > > | | GraphSAGE | 84.5±0.2 | 83.10±1.19 | 80.39±1.51 | 77.81±1.74 | 74.34±1.74 | 70.72±1.85 | 67.71±2.36 | 63.46±1.99 | 59.70±2.91 | 55.27±2.50 | 49.93±2.09 |
> > > > | | GAT | 84.4±1.0 | 79.50±1.80 | 77.63±2.14 | 75.26±1.76 | 72.70±2.76 | 70.33±2.16 | 67.73±2.38 | 64.98±2.73 | 61.17±2.76 | 57.46±3.68 | 51.73±2.34 |
> > > > | | Graph Transformer | 86.0±0.1 | 84.42±0.84 | 82.86±1.11 | 81.30±1.20 | 77.56±0.69 | 75.30±0.91 | 73.08±1.92 | 69.66±0.48 | 66.36±1.57 | 61.16±1.75 | 55.70±2.32 |
> > > > | | DeGLIF(sum) | 86.2±0.4 | **86.1±0.5** | **84.7±0.8** | **85.8±0.3** | **84.3±1.3** | **84.6±1.1** | **84.3±0.7** | **84.8±0.6** | **84.0±0.6** | **81.1±1.4** | **80.5±1.8** |
> > > > | PW | GCN | **86.4±0.2** | 86.1±0.6 | 85.2±0.3 | 83.0±1.1 | 80.0±1.9 | 77.1±2.7 | 72.7±3.5 | 67.2±4.0 | 60.4±5.0 | 52.8±4.4 | 43.7±4.2 |
> > > > | | GraphSAGE | 84.5±0.2 | 82.84±1.31 | 79.54±1.72 | 76.22±2.20 | 72.25±2.13 | 67.69±2.50 | 63.08±2.71 | 57.97±2.99 | 52.71±3.32 | 47.20±3.15 | 41.97±3.32 |
> > > > | | GAT | 84.4±1.0 | 79.35±1.67 | 76.38±1.95 | 73.47±2.42 | 68.99±3.07 | 65.87±3.48 | 61.14±2.79 | 55.89±2.96 | 50.89±3.38 | 44.76±3.42 | 40.33±3.41 |
> > > > | | Graph Transformer | 86.0±0.1 | 84.06±0.71 | 82.52±1.04 | 80.10±0.97 | 76.18±0.79 | 72.12±1.05 | 67.56±1.87 | 61.80±1.84 | 56.54±1.84 | 50.08±2.15 | 43.44±2.76 |
> > > > | | DeGLIF(sum) | 86.2±0.4 | **86.5±0.4** | **85.9±0.4** | **84.1±0.5** | **82.6±0.5** | **80.8±0.5** | **78.5±0.8** | **73.9±3.5** | **66.8±2.5** | **63.0±4.5** | **51.8±4.6** |
> > > >
> > > > For Citeseer data:
> > > >
> > > > | Noise Model | Algorithm | 0% | 5% | 10% | 15% | 20% | 25% | 30% | 35% | 40% | 45% | 50% |
> > > > | :--- | :--- | :--- | :--- | :--- | :--- | :--- | :--- | :--- | :--- | :--- | :--- | :--- |
> > > > | SLN | GCN | **77.6±0.5** | 76.6±0.2 | 73.9±1.1 | 71.2±0.9 | 69.0±1.1 | 66.3±0.4 | 63.4±1.5 | 59.3±1.4 | 55.4±1.1 | 51.3±1.3 | 48.0±1.4 |
> > > > | | GraphSAGE | 77.3±0.1 | 75.95±0.96 | 74.75±0.99 | 73.87±1.10 | 72.78±1.21 | 70.96±1.49 | 68.82±1.64 | 66.26±2.10 | 63.58±1.99 | 59.61±1.99 | 55.60±2.01 |
> > > > | | GAT | 76.8±0.4 | 76.46±1.53 | 75.65±1.56 | 75.38±1.66 | 75.10±1.46 | 74.59±1.61 | 73.71±1.60 | 73.15±1.51 | 72.47±1.46 | 70.88±1.64 | 70.25±2.67 |
> > > > | | Graph Transformer | 76.7±0.2 | 76.28±0.86 | 74.68±0.74 | 73.78±1.29 | 72.48±1.80 | 70.24±1.63 | 67.68±1.06 | 65.40±1.00 | 61.62±0.64 | 58.10±1.24 | 52.88±0.35 |
> > > > | | DeGLIF(sum) | 77.2±0.5 | **81.5±0.8** | **81.2±0.6** | **80.7±0.8** | **79.5±0.8** | **79.7±0.7** | **78.8±0.7** | **77.4±1.9** | **75.6±1.7** | **76.3±1.9** | **74.3±1.6** |
> > > > | PW | GCN | **77.6±0.5** | 78.6±0.6 | 78.2±0.6 | 76.8±0.9 | 75.5±1.1 | 73.2±1.1 | 70.6±1.6 | 67.0±2.2 | 62.1±3.8 | 47.1±2.7 | 45.3±3.3 |
> > > > | | GraphSAGE | 77.3±0.1 | 75.97±0.88 | 74.82±0.99 | 73.35±1.36 | 71.45±1.44 | 68.93±1.92 | 65.25±2.39 | 60.78±3.35 | 55.92±3.13 | 49.71±2.85 | 43.99±2.57 |
> > > > | | GAT | 76.8±0.4 | 76.62±1.49 | 76.22±1.46 | 75.77±1.45 | 74.72±1.39 | 73.31±1.84 | 71.04±2.33 | 67.48±2.75 | 62.03±3.60 | 55.00±3.75 | 45.96±3.47 |
> > > > | | Graph Transformer | 76.7±0.2 | 76.10±1.40 | 75.18±0.34 | 73.60±0.86 | 70.92±0.90 | 68.86±1.03 | 64.62±2.29 | 58.90±1.81 | 54.64±2.01 | 49.04±2.25 | 43.32±0.77 |
> > > > | | DeGLIF(sum) | 77.2±0.5 | **80.2±0.9** | **78.8±0.9** | **78.2±1.1** | **78.0±0.8** | **77.8±0.7** | **75.6±1.2** | **73.4±1.2** | **69.6±2.8** | **63.7±3.7** | **54.0±1.8** |
> > > >
> > > > *(Results continued in the next block)*

---

> ### Author Response · Authors · 2026-05-14
> **Response to 1. (comparison with other baselines) (Continued)**
>
> ### **Response to 1. (comparison with other baselines) (Continued)**
>
> For Amazon Photo Dataset:
>
> | Noise Model | Algorithm | 0% | 5% | 10% | 15% | 20% | 25% | 30% | 35% | 40% | 45% | 50% |
> |-|-|-|-|-|-|-|-|-|-|-|-|-|
> |SLN|GCN|**91.1±0.2**|87.3±0.8|87.1±0.3|85.5±0.4|85.7±1|85.7±1|84.6±2.1|83.7±1.4|80.7±2|79.1±1.2|75.2±5.2|
> | |GraphSAGE|88.7±0.2|90.56±0.86|88.49±0.84|85.86±1.99|84.10±1.48|81.73±2.29|77.97±2.17|75.72±1.59|70.79±2.14 | 65.88±3.47|60.79±3.71|
> | |GAT|90.9±0.4|90.6±1.2|90.1±1.2|89.2±1.5|88.3±2.1|86.8±1.9|85.7±2.8|83.8±4.2|81.5±3.7|79.8±3.4|75.4±4.4|
> | | Graph Transformer |89.2±0.3|85.57±0.83|83.17±1.89|79.73±1.42|77.52±2.20|74.34±3.35|69.06±2.85|64.24±2.99| 62.37±0.45|58.77±2.13|52.70±1.47|
> | |DeGLIF(sum)|91±0.6|**91.8±0.6**|**91.6±0.2**|**90.8±0.4**|**90.3±1**|**90.0±0.7**|**89.8±2**|**88.1±1.3**|**87.1±1.6**|**86.4±1.7**|**82.6±4.2**|
> |PW|GCN|**91.1±0.2**|89.5±1|88.6±0.9|87.0±1.6|83.5±1.7|82.4±2.4|77.6±3.8|69.0±5.4|60.8±7.9|62.4±6.3| 50.9±8.3 |
> | |GraphSAGE|88.7±0.2|90.39±1.01|87.76±1.17|84.68±2.27|81.63±2.54|77.96±2.87|72.25±2.84|65.55±3.68|58.99±2.38|50.78±4.01|44.42±4.67|
> | |GAT|90.9±0.4|**90.9±1.2**|**89.4±1.6**|87.5±3.2|85.4±3.8|82.9±3.3|78.4±4.7|72.8±5.8|65.5±8.5|56.5±8.3|47.2±7.7|
> | | Graph Transformer | 89.2±0.3 | 85.48±0.95 | 82.45±1.06 | 78.93±1.54 | 74.06±3.01 | 71.27±4.41 | 67.01±4.39 | 62.16±2.99 |58.16±3.18| 54.38±2.85 | 45.96±2.73 |
> | | DeGLIF(sum) | 91±0.6 | 89.6±1 | 89.2±1.3 | **89.2±0.2** | **87.4±0.6** | **85.3±2.6** | **83.2±2.1** | **77.0±6.1** | **74.5±7.6** | **71.8±5.4** | **60.6±10** |
>
> Notably, DEGLIF demonstrates superior performance compared to all evaluated baselines.
>
> ---
>
> ### **Response to 2. (regarding redesigning Figure 1)**
>
> We thank the reviewer for this observation and will update the visual style of Figure 1 to align with the rest of the paper in the camera-ready version.
>
> ---
>
> ### **Response to 3,4,5**
>
> We thank the reviewer for pointing these, we have fixed these.
>
> ----
>
> ## **Response to additional comments**
>
> ### **Response to 1. (regarding transfer to inductive setting)**
>
> While our current work focuses on the transductive learning setup, and to the best of our knowledge, the intersection of label-noise robust learning and inductive graph settings remains underexplored in the current literature.
>
> Intuitive answer:
> The DeGLIF framework would remain the same and would process the influence values in the same way. The change would be observed in the way the influence value is calculated.
>
> Equation 4 used to calculate the change in model parameter
>
> If the training node belongs to graph $j$, then
>
> $I(-z)=I(-z_i)+I(M\to M+\Delta) $ would become $I(-z)=I(-z_i)+I(M_j\to M_j+\Delta) $
>
> and in summation $k\in V_{train}$ would become $k \in V_{train}\cap V_j.$
>
> Furthermore, inductive settings typically employ architectures like GraphSAGE rather than standard GCNs. This architectural shift would further require a change in the influence calculation. For instance, because GraphSAGE relies on neighborhood sampling, we would only need to consider the impact on nodes whose sampled computation graphs specifically include the removed point.
>
> While transferring this to an inductive setting naturally requires future empirical validation, we believe DeGLIF is inherently better suited for this transition than current baselines. Many existing robust algorithms, such as NRGNN and RTGNN, rely heavily on transductive structural modifications (e.g., adding or removing edges based on global node similarities). In contrast, DeGLIF's reliance on data attribution via the influence function offers a much more natural and flexible pathway to inductive node classification.
>
> ---
>
> ### **Response to 2. (clarification on transferring Theorems 1 and 2 to test risk)**
>
> We thank the reviewer for highlighting this point and allowing us to clarify the statistical connection.
>
> Theorems 1 and 2 formally establish a reduction in the empirical risk evaluated on the clean subset $D_c$. The link to the test risk relies on the standard learning theory assumption that both $D_c$ and the test set are sampled i.i.d. from the identical underlying clean data distribution. Because $D_c$ is drawn from this clean distribution, the empirical risk $R(\theta, D_c)$ serves as an unbiased estimator of the true population risk. Consequently, an algorithmic step that is proven to reduce the empirical risk on $D_c$ is expected to correspondingly reduce the true expected risk, and by extension, the risk on the unseen test set. We see this theoretical expectation verified empirically by our results in Table 2, where DeGLIF consistently improves accuracy on the unseen test set.
>
> We agree that this distinction is important for theoretical rigor. We have revised the final sentence of Section 3 to be more mathematically precise, changing `can be directly translated to test risk' to 'serves as an unbiased estimator for the expected risk, thereby generalizing to the test risk'.
>
> ---
>
> *(Rebuttal responses continued below.)*

---

> > ### Author Response · Authors · 2026-05-14
> > **Response to 3. (regarding choice of single hidden layer)**
> >
> > ### **Response to 3. (regarding choice of single hidden layer)**
> >
> > First, we would like to clarify that the DeGLIF framework imposes no inherent restrictions on the number of hidden layers or the specific GNN architecture utilized. The framework is broadly applicable as long as the Hessian matrix of the loss function remains invertible.
> >
> > The choice to use a single hidden layer for the GCN was primarily driven by empirical observations rather than methodological limitations. During our initial experimental setup, a single hidden-layer architecture provided strong, stable performance for the baseline GCN. To ensure a fair and consistent evaluation, we adapted this exact same architecture across all our relevant experiments.
> >
> > Furthermore, this architectural choice aligns directly with established practices in the field of graph learning under label noise. Recent comprehensive benchmarking studies, specifically NoisyGL (Wang et al., 2024), have similarly found that a single hidden-layer architecture is highly effective for these tasks and have adopted it as their standard evaluation setting. Using a similar architecture makes comparison more robust and fair.
> >
> > ----
> >
> > We again thank the reviewer for their time and constructive feedback.

---

### Review · Reviewer_CKQA · 2026-04-29

**Summary Of Contributions:**

This paper introduces a denoising framework for robust node classification against label noise on graphs. The method assumes access to a small clean validation set alongside a noisy training set, and extends the leave-one-out influence function for GNNs from previous work to estimate the change in validation loss when a training node is removed. Intuitively, if the loss on the clean validation set drops after removing a training node, this training node is probably with label noise. Theoretical results argue that, under additivity assumptions, removing/relabeling the detected nodes does not increase risk on the clean validation set.

**Additional Comments:**

1. How is the invertibility of the Hessian $H_\theta$ ensured in practice? Cross-entropy with a softmax output and a non-trivial GNN architecture is not generally strictly convex, so the empirical Hessian may be singular or near-singular. Is a damping/regularization term (e.g., $H_\theta + \lambda I$) added before inversion? If so, what is its value and how was it chosen?
2. The theory and Algorithm 1 are stated for an L2-regularized convex objective, but Model-1 is a GCN, which is non-convex. The optimum $\hat\theta$ in Theorem 1 should be a stationary point rather than a global minimum. Could the authors comment on whether their analysis goes through under this relaxation, and whether the empirical optimum (a local minimum from Adam/SGD) is close enough to a stationary point for the influence approximation to be meaningful?
3. The successive-applications experiment shows that the noise fraction saturates rather than going to zero. Is the residual noise concentrated on specific node types (e.g., low-degree, near class boundaries)? Understanding which nodes DeGLIF cannot fix would clarify its limitations.
4. If I understand correctly, Theorem 2’s relabeling guarantee compares against $\hat\theta_{-D_n}$, i.e., training with $D_n$ removed. But the algorithm trains Model-2 on the relabeled data, not on $D \setminus D_n$. The comparison the practitioner cares about is DeGLIF (relabel) vs. baseline (train on $D$). Could the authors clarify the chain of inequalities tying these together?

**Audience:**

Yes

**Audience Explanation:**

Yes. The paper sits at a useful intersection: data-centric robustness for GNNs combined with influence-function-based attribution.

**Broader Impact Concerns:**

I do not identify any major ethical concerns requiring additional discussion.

**Claims And Evidence:**

No

**Claims Explanation:**

The empirical evidence broadly supports the central claim that DeGLIF improves robustness at moderate-to-high noise levels across the considered datasets.

Several aspects weaken the strength of the evidence.

(1) On Cora and Citeseer, the gains over the strongest baseline are within 1–2 percentage points at many noise levels, and the standard deviations frequently overlap, no statistical significance test is reported. What is more, in Amazon Photo, the proposed method is worse than the baseline CP.

(2) The theoretical results rely on a first-order Taylor approximation and an additivity assumption when multiple training nodes are removed. The additivity assumption is particularly strong for graphs, since removing multiple nodes can interact non-additively through message passing (e.g., when removed nodes share neighbors). The paper inherits this assumption from previous work, but those works study i.i.d. data.

(3) Theorems 1–3 establish that the approximate change in risk is non-negative; whether the approximation error swamps this guarantee at realistic problem sizes is not investigated.

(4) The translation from $R(\hat\theta_{-D_n}, D_c)$ to test risk is asserted from $D_c$ being sampled from the clean distribution, but this only holds in expectation, and $|D_c|=50$ is quite small.

**Requested Changes:**

1. The additivity assumption underlying Theorems 1 and 2 deserves explicit discussion in the main text (not just citation). For graph data, when two removed nodes are graph-neighbors or share neighbors, the second-order interaction term $I(-z_1, -z_2) - I(-z_1) - I(-z_2)$ is unlikely to be negligible. An empirical check, e.g., comparing the influence-predicted change in $D_c$ loss against the actual change after retraining without the detected nodes would substantially strengthen the theoretical claims.
2. Several baselines appear to perform unusually badly on Amazon Photo, with very large standard deviations (e.g., 12.3±17.7). This suggests possible reproduction issues. A short note clarifying whether default hyperparameters were used or whether they were re-tuned for this dataset would address concerns about whether DeGLIF’s edge over these baselines reflects the method or the tuning effort.
3. Statistical significance: with 5 seeds and overlapping standard deviations in many cells of Table 2, claims like “outperforms by up to 17.8%” are driven by individual cells. Reporting average rank across all (dataset, noise-level) cells, or a Wilcoxon signed-rank test against the strongest baseline, would give a fairer summary.
4. The citation format in Section 6 COMPUTATIONAL COMPLEXITY is incorrect:  "(Koh et al., 2017; [2])" and "i.i.d. data Koh et al. (2024); Lyu et al. (2023)". Section 4.1.1 contains an unresolved reference (Appendix ??).

Minor:

1. The notation in Algorithm 1 mixes $z_i$ and $x_i$ in step 5 without re-introducing $x_i$.
2. In Theorem 2, the bound $0 \le \varphi_k(z) \le 1$ is stated but not justified. (For reviewers: it does hold because $f_k + f_m \le 1$ implies $1-f_m \ge f_k$, hence $\log_{f_k}(1-f_m) \le 1$.) A one-line justification would help readers verify the claim.
3. Figure 1 uses a different visual style than the rest of the paper; this is cosmetic but noticeable.
4. In Section 3.1, the description of $A_z$ and the indicator notation $1_{A_z}(v_i)$ is harder to read than necessary—just stating “we count the validation points with negative influence” would be clearer.
5. The discussion in Section 6 cites “Koh et al., 2017” and a bracketed “[2]” without a clear referent.
6. The PubMed result for RNCGLN is reported as “NA” due to OOM. This is fine, but it should be explicitly noted as OOM in the table caption rather than NA, which is ambiguous.

---

> ### Author Response · Authors · 2026-05-14
> **Response to Reviewer CKQA**
>
> We thank the reviewer for the time devoted to carefully reading our paper and for the many insightful suggestions. We have structured our response to the points raised in the following order. First, we address the requested changes (both major and minor). Next, we address the comments, justifying the reviewer's assessment of our paper's accuracy. Finally, we address all additional comments.
>
> ---
>
> ## **Response to requested changes: Major**
>
> ### **Response to 1. (regarding additivity assumption)**
>
> We thank the reviewer for this insightful observation. We partially agree with the reviewer's intuition.
> To strengthen our additivity claims, we designed an experiment following the reviewer's suggestion. Specifically, we compared the actual change in validation loss against the change predicted by our influence metric when removing a group of nodes.
>
> Experimental Setup and Result:
>
> To evaluate the impact of removing groups of size $k$, we randomly sampled 100 distinct subsets of training nodes, each containing $k$ nodes. For each subset, the predicted change in validation loss was computed as the sum of the individual node influences $(I(-z_1, \dots, -z_k) = I(-z_1) + \dots + I(-z_k))$. To determine the actual ground-truth change in validation loss, we removed the nodes and their associated edges in each subset and retrained the network from scratch. Finally, we calculated the Pearson correlation between the predicted and actual changes in loss. We repeated this process for varying group sizes, specifically $k \in \lbrace 1, 2, 25, 50,200 \rbrace$. The resulting Pearson correlations are as follows (Computations are available at: <https://anonymous.4open.science/r/DeGLIF-0487/group_effect.ipynb>):
>
> | Group Size ($k$) | Pearson Correlation |
> | :--- | :--- |
> | 1 | 0.8285 |
> | 2 | 0.7707 |
> | 25 | 0.7500 |
> | 50 | 0.7670 |
> | 200 | 0.7642 |
>
>
>  As anticipated, moving from individual node removal ($k=1$) to group removal ($k>1$) results in a slight drop in correlation, which confirms the presence of unmodeled second-order interactions. However, the correlation stabilizes quickly and remains robustly high (consistently above 0.75) even as the group size scales up to $k=200$. This demonstrates that while the strict additivity assumption is indeed an approximation, the error introduced by ignoring interaction terms is not prohibitive for practical applications. The aggregated individual influences serve as a highly reliable and scalable directional proxy for group influence.
>
>  We have added a dedicated remark just after Theorem 1 statement, which explicitly discusses the non-trivial nature of second-order interactions in graph data and summarizes our empirical findings. The full experimental setup and tabular results have been detailed in Appendix D.1 to support this main-text discussion.
>
> ---
>
> ### **Response to 2. (regarding high variance for some baselines on the Amazon Photo dataset)**
>
> We agree with the reviewer's observation about high standard deviation and appreciate the careful attention to detail. Indeed, we also had noticed this unusually poor performance and high variance (e.g., 12.3 $\pm$ 17.7) for certain baselines like CRGNN on the Amazon Photo dataset during our own experimentation phase.
>
> *To address the reproduction and tuning concerns:* the official implementation for these algorithms is not publicly available, nor did the original authors report results on the Amazon Photo dataset. This inherently makes direct reproduction challenging. As already noted in the implementation section of our manuscript, to ensure a fair and rigorous comparison, we relied on the standardized implementations provided by the NoisyGL benchmark  (Wang et al., 2024, NeurIPS 2024).
>
> We can confirm that these baselines were properly tuned and not simply run with default hyperparameters. We relied on the extensive hyperparameter tuning sweep conducted by NoisyGL, which is specifically detailed in their Table A8. For instance:
> * For CRGNN, the hyperparameters $\alpha$ and $\beta$ were tuned over the set {0.1, 0.2, ..., 1}.
> * For CGNN, the parameters $\gamma$ and $\omega$ were tuned over the set {0.6, 0.7, 0.8, 0.9, 0.95}.
>
> Furthermore, alongside these model-specific parameters, general network settings such as learning rate, weight decay, layer number, and hidden size were also systematically tuned during the search process.
>
> Importantly, the NoisyGL benchmark (Wang et al., 2024) also reports similar instability and poor performance for both CRGNN and CGNN on the Amazon Photo dataset. Therefore, the observed gap in performance is not due to a lack of tuning or a reproduction error on our end, but rather reflects an inherent limitation of these specific baseline methods when applied to this particular dataset.
>
> ___

---

> ### Author Response · Authors · 2026-05-14
> **Response to requested changes: Major (continued)**
>
> ### **Response to 3. (Regarding statistical significance of results)**
>
> We sincerely thank the reviewer for this constructive suggestion. We agree that, when evaluating performance across different experimental conditions, a rigorous statistical test provides a much fairer and clearer summary of the results.
>
> Following the reviewer's recommendation, we computed the average rank for all algorithms across all experimental configurations (datasets, noise types, and noise levels). We then conducted Wilcoxon signed-rank tests to compare both DeGLIF variants against the two strongest performing baselines, TSS and RTGNN (strongest based on average ranked test).
>
> The results conclusively demonstrate that DeGLIF is statistically significantly better than the compared baseline, and claims of better performance are not driven by a single cell.
>
> * DeGLIF-sum: Achieved an average rank of 1.27, compared to 4.43 for the strongest baseline in this evaluation (TSS). The Wilcoxon signed-rank test confirms this performance gap is highly statistically significant against both TSS ($p = 8.09 \times 10^{-12}$) and RTGNN ($p = 3.37 \times 10^{-11}$).
>
> * DeGLIF-mv: Achieved an average rank of 2.03, compared to 4.30 for the strongest baseline in this evaluation (RTGNN). This improvement is also highly statistically significant against both RTGNN ($p = 8.15 \times 10^{-6}$) and TSS ($p = 4.59 \times 10^{-9}$).
>
> Computations are available at: <https://anonymous.4open.science/r/DeGLIF-0487/wilcoxon_test.ipynb>
>
>
>  *Furthermore, we wish to emphasize that while certain baseline algorithms achieve comparable results on specific datasets (such as CP on Amazon Photo), DeGLIF is unique in its consistent robustness. Unlike other methods, our framework maintains strong performance across all evaluated scenarios, including variations in average node degree and training set sizes. We attribute this broad generalization to several key factors, which we explain below.*
>
> Conceptually, the majority of recent robust GNN methods, such as RTGNN, CGNN, CP, and NRGNN, etc., can be broadly grouped into two categories based on their mechanisms. The first category includes methods like NRGNN and RTGNN, which attempt to mitigate noise by adding edges between unlabelled and labeled nodes based on feature similarity. While effective on sparse networks, this edge-adding strategy offers little benefit on already dense graphs like Amazon Photos (in contrast, it degrade performance). The second category, which includes methods like CP and CGNN, relies heavily on label smoothness and strong community structures. Consequently, these algorithms tend to underperform on sparse citation networks like Cora and Citeseer, where community boundaries are weak or overlapping.
>
> *Because DeGLIF does not rely on these topological modifications or smoothness assumptions, it maintains robust performance across both sparse and dense graphs.* Instead, DeGLIF uses a leave-one-out influence function to estimate the impact of removing a training node on a small clean validation set.
>
> Compared with approaches like TSS, DeGLIF fundamentally differs in how it handles corrupted data. TSS relies on topological sample selection, progressively selecting clean nodes for training while discarding noisy ones. DeGLIF, conversely, actively identifies noisy data and uses a theoretically motivated relabeling function to correct it. This capability allows DeGLIF to work exceptionally well even when the training set is small (such as in the Amazon Photo dataset with 5\% nodes as the training set), a scenario where discarding data via sample selection severely degrades performance.
>
> Furthermore, as highlighted in Section 5.4 of our manuscript, DeGLIF is not solely designed to compete with these algorithms, but rather to assist them. Because DeGLIF acts as a denoising framework, the resulting cleaned dataset can be fed into any other end-to-end noise-robust algorithm. Our experiments ( in Section 5.4) demonstrate that a hybrid model (combining DeGLIF's denoising with TSS) performs better than either method alone, particularly under high noise conditions.
>
> ---
>
> ### **Response to 4. (Regarding typos in citation format)**
>
> We thank the reviewer for pointing this out. We have fixed the issues with citation format throughout the paper.

---

> > ### Author Response · Authors · 2026-05-14
> > **Response to requested changes: Minor**
> >
> > ## **Response to requested changes: Minor**
> >
> > ### **Response to 1. (Regarding notation in Algorithm 1)**
> >
> > The notation used in Algorithm 1 is deliberate. In our framework, a training node is defined as the tuple $z_i = (x_i, y_i)$, where $x_i$ represents the node features and $y_i$ represents the label. In Step 5, $f$ denotes the forward pass of the GNN model. Because the model must generate predictions based solely on the node features (without seeing the label), it evaluates $f(x_i)$ rather than $f(z_i)$. We agree that we had used $f(z)$ in Section 3.2, which is now changed to $f(x)$.
> >
> > However, we completely agree that switching between $z_i$ and $x_i$ without an immediate refresher can cause unnecessary confusion for the reader. To resolve this and improve the paper's readability, we have added a dedicated Notations paragraph at the end of the Introduction. This section explicitly defines the relationship between $z_i$, $x_i$, $y_i$, and the GNN model $f$, ensuring all terms are cleanly introduced before they appear in the algorithms and mathematical derivations.
> >
> > ---
> >
> > ### **Response to 2. (Regarding justification in Theorem 2)**
> >
> >  We sincerely thank the reviewer for pointing this out and for providing the exact mathematical reasoning for other reviewers. We completely agree that including this saves the reader from having to derive it themselves. Following your suggestion, we have incorporated this one-line justification directly into Section 3.2 of our revised manuscript to easily verify the claim.
> >
> > ---
> >
> > ### **Response to 3. (Regarding the visual style of Figure 1)**
> > We thank the reviewer for this observation and will update the visual style of Figure 1 to align with the rest of the paper in the camera-ready version.
> >
> > ---
> >
> > ### **Response to 4,5,6**
> >
> > We thank the reviewer for pointing these out. We have fixed all of them.
> >
> > ----
> >
> > We now address the comments made in justification of the reviewer's assessment of our paper's accuracy.

---

> ### Author Response · Authors · 2026-05-14
> **Response to reviewer's comments regarding the technical accuracy of results**
>
> ## **Response to reviewer's comments regarding the technical accuracy of results**
>
> ### **Response to (1) (Regarding statistical significance)**
>
> Answered in requested changes 2.
>
> ---
>
> ### **Response to (2) (Regarding additivity assumption)**
> Answered in Requested changes 1.
>
> ---
>
> ### **Response to (3) (Regarding approximation in statement of Theorem 1-3)**
>
> We thank the reviewer for raising this insightful point regarding the approximation error and its potential impact on the guarantees in Theorems 1-3.
>
> To clarify, the approximation error in our framework stems strictly from the use of the first-order Taylor series expansion in calculating the influence function. Because DeGLIF directly extends the graph influence formulation proposed by Chen et al. (2023), the theoretical bounds of this approximation error are governed by their foundational analysis.
>
> Chen et al. have already conducted a rigorous theoretical investigation into this specific approximation error for graph data. In their work, they mathematically define the error bounds for removing graph elements (Theorem 4.1 and Corollary A.1). They also investigate the accuracy of Influence values in practical scenarios. Their empirical validation shows that even when the exact numerical estimates diverge slightly, the relative ranking of the influence values remains highly accurate and strongly correlated with the true change in risk.
>
> This distinction is crucial for DeGLIF. Our framework relies primarily on the sign and relative ranking of the influence values to threshold and identify noisy nodes (via DeGLIF-mv and DeGLIF-sum). Because the rank correlation remains highly consistent even at realistic problem sizes, the approximation error is very unlikely to affect the practical utility of our guarantees.
>
> Finally, we emphasize that relying on a first-order Taylor series approximation is standard practice when using influence functions for both graph and non-graph data. This approach is widely adopted in the machine learning literature and consistently performs well in practical settings. Furthermore, even a first-order approximation requires computing the Hessian, which is already a major computational bottleneck. Pursuing higher-order approximations would introduce significantly more complexity and would require further research from the community to become computationally feasible.
>
> ---
>
> ### **Response to (4) (Regarding Relation between risk on D_c and actual test risk)**
> We sincerely thank the reviewer for rigorous attention to this detail, and we completely agree with the mathematical assessment: the translation from the empirical risk on $D_c$ to the true test risk holds strictly in expectation, and the finite (and small) size of $D_c$ naturally introduces variance.
>
> Mathematically, the statements and proofs of Theorems 1 and 2 would hold exactly as written if $D_c$ were replaced directly by the test set. However, using the test set to compute influence and identify noisy training nodes would constitute data leakage and violate the fundamental principle of evaluating on unseen data. Therefore, we utilize the independent subset $D_c$ as a proxy.
>
> In this capacity, $R(\hat{\theta}_{-D_n}, D_c)$ serves as an unbiased estimator of the expected test risk, given the standard assumption that $D_c$ and the test set are drawn i.i.d. from the identical underlying clean distribution. This theoretical positioning directly follows established influence-based literature (e.g., Kong et al., ICLR 2022; Wang et al., 2018; Ting \& Brochu, 2018; Ren et al., 2020;
> Wang et al., 2020), which similarly utilize small, unbiased validation sets as proxies for true expected risk.
>
> From a practical standpoint, the size of $D_c$ was deliberately kept small ($|D_c| = 50$) to ensure DeGLIF avoids becoming a hindrance in real-world applications. In most practical domains, curating a massive dataset of perfectly clean, expert-verified labels is prohibitively expensive. By explicitly restricting our method to a very small $D_c$, we designed DeGLIF to be viable in realistic settings where clean data is highly scarce.
>
> Furthermore, we validate this assumption empirically in Section 5.2 (Table 3) of our manuscript, where we investigate the impact of varying the size of $D_c$. The results confirm the reviewer's statistical intuition: as $|D_c|$ increases, it more accurately reflects the true population distribution, tightening the bounds and yielding higher accuracy. Yet, our experiments simultaneously demonstrate that even with a highly constrained $|D_c| = 50$, the unbiased estimator provides a sufficiently strong and accurate signal to successfully correct noisy nodes.
>
> *Answer continued below*

---

> > ### Author Response · Authors · 2026-05-14
> > **Response to (4) (Regarding Relation between risk on D_c and actual test risk): Continued**
> >
> > ### **Response to (4) (Regarding Relation between risk on D_c and actual test risk): Continued**
> >
> > We fully agree that the theoretical distinction is important. To ensure mathematical precision, we have revised the end of Section 3 to explicitly state: `It is worth mentioning that, because $D_c$ is
> > sampled i.i.d. from the underlying clean distribution, the empirical risk reductions established in Theorems
> > 1 and 2 serve as unbiased estimators for the true expected risk, thereby generalizing to the unseen test risk'.
> >
> > * Tianyang Wang, Jun Huan, and Bo Li. Data dropout: Optimizing training data for convolutional
> > neural networks. ICTAI, IEEE, 2018.
> > * Daniel Ting and Eric Brochu. Optimal subsampling with influence functions. NeurIPS, 2018.
> > * Zhongzheng Ren, Raymond A. Yeh, and Alexander G. Schwing. Not all unlabeled data are equal:
> > Learning to weight data in semi-supervised learning. NeurIPS, 2020.
> > *Zifeng Wang, Hong Zhu, Zhenhua Dong, Xiuqiang He, and Shao-Lun Huang. Less is better: Unweighted data subsampling via influence function. AAAI,  2020
> >
> > ---
> >
> > We next address additional comments given by the reviewer

---

> ### Author Response · Authors · 2026-05-14
> **Response to Additional Comments**
>
> ## **Response to additional comments**
>
> ### **Response to 1. (Regarding damping/regularization term)**
>
> We thank the reviewer for this insightful question. The reviewer is absolutely correct that the empirical Hessian for a GNN trained with standard cross-entropy is often singular or near-singular.While we do not manually add a separate damping term strictly for the inversion step, we employ $L_2$ regularization (weight decay) during the initial training of the GCN. Mathematically, adding an $L_2$ penalty of $\frac{\lambda}{2} \|\theta\|\_2^2$ to the cross-entropy objective naturally results in a Hessian of $H\_{CE} + \lambda I$. This inherently provides the exact damping effect the reviewer describes.
>
> For our experiments, the regularization parameter ($\lambda$) was set to $1e-4$ for the Cora, Citeseer, and PubMed datasets, and $1e-5$ for the Amazon Photo dataset. These values were chosen because they yielded the best performance on the validation set for the base GCN (Model-1). Furthermore, these values are standard within the graph learning community and are well-known to provide stable, optimal performance for these specific benchmark datasets in both general node classification and label-noise robust learning scenarios. We have explicitly stated in the revised manuscript.
>
> ---
>
> ### **Response to 3. (Regarding the behavior of residual noisy nodes)**
>
> We thank the reviewer for this insightful suggestion. We investigated whether the residual noise is concentrated on specific node types by analyzing several node properties: degree, node homophily (as a proxy for class boundaries), training loss, and PageRank centrality. However, we did not observe any distinguishing characteristics or clear patterns among the remaining noisy nodes.
>
> ---
>
> ### **Response to 4, (clarity on chain of inequalities)**
>
> The guarantee for the practitioner (DeGLIF relabeling vs. the baseline) is established through the transitive relationship between Theorem 1 and Theorem 2. Let $R(\theta, D_c)$ denote the empirical risk on the clean validation set.
>
> 1. Theorem 1 establishes that dropping the identified noisy nodes ($D_n$) results in a lower or equal risk compared to the baseline trained on the fully noisy dataset ($D$):$$R(\hat{\theta}, D_c) \ge R(\hat{\theta}_{-D_n}, D_c)$$
>
> 2. Theorem 2 establishes that training on the relabeled data yields a lower or equal risk compared to simply dropping those nodes: $$R(\hat{\theta}_{-D_n}, D_c) \ge R(\hat{\theta}_r, D_c)$$
>
> By combining these two theorems, we arrive at the final chain of inequalities:
> $$R(\hat{\theta}, D_c) \ge R(\hat{\theta}_{-D_n}, D_c) \ge R(\hat{\theta}_r, D_c)$$
>
> This chain directly demonstrates that DeGLIF's relabeling strategy ($R(\hat{\theta}_r, D_c)$) theoretically achieves a lower risk than the baseline trained on the original noisy data ($R(\hat{\theta}, D_c)$).
>
> ---
>
> *Response to 2. is included in the next block.*

---

> > ### Author Response · Authors · 2026-05-14
> > **Response to additional comments (Continued)**
> >
> > ### **Response to 2. (Regarding convergence of empirical optimum to a stationary point)**
> >
> > We thank the reviewer for this insightful observation. The reviewer is correct that for a non-convex GCN, the optimization process mostly converges to a local stationary point rather than a strict global minimum.
> >
> > We formulated the theoretical foundation under the assumption of a global minimum primarily to ensure theoretical smoothness, as a strictly convex objective naturally guarantees that the Hessian matrix is positive definite and easily invertible. However, the core mathematical mechanics of our analysis do indeed carry through to a local stationary point. The first-order Taylor expansion used to derive the influence function in Section C.1 relies on the condition that the gradient of the empirical risk is zero ($\nabla_{\theta}R(\hat{\theta}) = 0$). Because this property holds at any local stationary point, the derivation of the parameter change approximation remains mathematically valid. The only strict requirement for this relaxation to hold in practice is that the local Hessian matrix at that specific stationary point must be invertible.
> >
> > We explicitly accounted for this practical reality in Section 3 of the paper, where we state that DeGLIF is architecture-agnostic and one can choose any model for Model-1 "as long as the Hessian obtained is invertible." In our implementation, the inclusion of standard L2 regularization (weight decay) serves as a damping term that adds a positive constant to the diagonal of the Hessian, ensuring the Hessian is invertible at the empirical stationary point and, hence, allowing Theorems 1 through 3 to hold locally.
> >
> > To address the second part of the reviewer's question, we empirically examined our training dynamics. In our standard experimental setup, we observed that the gradient norm does not vanish completely at convergence, typically settling in the range of $[0.01, 0.02]$ due to the momentum dynamics of the Adam optimizer. However, as discussed by Koh \& Liang (2017, Section 4.2), influence function approximations can still yield highly accurate estimates even when non-convex models (such as CNNs) do not strictly converge to a point where the gradient is exactly zero.
> >
> > To verify if this holds for our GCN (Model-1), we conducted a ground-truth correlation experiment following a similar methodology. Specifically, we measured the Pearson correlation between the predicted change in validation loss (estimated efficiently via DeGLIF) and the actual change in validation loss (obtained by physically removing a training node, stripping its edges, and completely retraining the model from scratch). We repeat this process for 50 randomly sampled nodes from the training set. To train GCNU, we use hyperparameters used in our standard experiments (learning rate $= 0.005$, weight decay $\lambda = 1e^{-4}$, and $|D_c|=50$). We consistently observed strong Pearson correlation coefficients ranging from $0.70$ to $0.86$ (across runs). The variance across different runs is primarily attributable to the statistical noise of randomly sampling these 50 training nodes, alongside the inherent optimization variance of Adam.
> >
> > To isolate the mathematical validity of our approximation from this sampling and optimization noise, we modified the evaluation setup. First, to eliminate sampling bias, we expanded the leave-one-out actual loss evaluation from 50 random training nodes to the entire training set (while keeping the standard 50-node validation set constant). Furthermore, slightly increased the damping factor ($\lambda = 1e^{-2}$) to perfectly condition the local Hessian matrix. Under these stabilized conditions, the Pearson correlation reaches $0.999$. Another choice of damping factor ($\lambda=1e^{-3}$ gives Pearson correlation $0.8284.$
> >
> > These empirical results carry two important implications. First, the perfect correlation in the stabilized setup mathematically validates that our derivation holds when the model reaches a strict local stationary point. Second, the strong correlation ($0.70$ to $0.86$) achieved under our standard training setup demonstrates that the empirical optimum reached by Adam is practically good enough. This confirms that the influence approximation remains highly meaningful without requiring perfect convergence, allowing DeGLIF to reliably identify harmful nodes for practical denoising applications.
> >
> > We have added a summary of this discussion as a remark after Theorem 1.
> >
> > ---
> >
> > We thank the reviewer again. We have also updated the PDF.

---

### Decision · Action_Editor_eruo · 2026-07-06

**Recommendation:** Accept as is

**Audience:**

Yes

**Audience Explanation:**

At least some TMLR readers are likely to be interested in this work. The paper’s contribution is not primarily in developing new influence-function theory, but in adapting and applying influence-based denoising to label-noise-robust node classification and demonstrating that this can be effective empirically. The problem of robust learning with noisy labels on graphs is relevant to graph machine learning, data-centric robustness, and practical weakly supervised graph applications.

**Claims And Evidence:**

Yes

**Claims Explanation:**

The major claims are sufficently supported.

The reviewers converged toward the view that the main empirical claims are sufficiently supported after the rebuttal and revision cycle. The paper targets on label-noise-robust node classification using influence-function-based denoising. The empirical results support the claim that DeGLIF improves robustness across several graph benchmarks and noise settings.

Reviewer 4cXi found that the empirical results generally support the robustness claims and noted that the paper includes useful ablations on clean-set size, relabeling, repeated denoising, and hyperparameter sensitivity. Reviewer 6D3i also judged the evidence to be positive, while noting that gains are more convincing at higher noise levels and can be marginal at lower noise levels. Reviewer CKQA initially raised substantial concerns about statistical significance, overlapping standard deviations, the additivity assumption in graph settings, the use of first-order influence approximations, and the link between clean-validation risk and test risk. The authors have added addtional experiments and metrics which addresses the concern.

The theoretical claims remain more limited than the empirical contribution. They rely on influence-function approximations, local assumptions, and a small clean validation set. The authors also acknowledged these limitations and revised the discussion.